# Genome-wide association study meta-analysis provides insights into the etiology of heart failure and its subtypes

Heart failure (HF) is a major contributor to global morbidity and mortality. While distinct clinical subtypes, defined by etiology and left ventricular ejection fraction, are well recognized, their genetic determinants remain inadequately understood. In this study, we report a genome-wide association study of HF and its subtypes in a sample of 1.9 million individuals. A total of 153,174 individuals had HF, of whom 44,012 had a nonischemic etiology (ni-HF). A subset of patients with ni-HF were stratified based on left ventricular systolic function, where data were available, identifying 5,406 individuals with reduced ejection fraction and 3,841 with preserved ejection fraction. We identify 66 genetic loci associated with HF and its subtypes, 37 of which have not previously been reported. Using functionally informed gene prioritization methods, we predict effector genes for each identified locus, and map these to etiologic disease clusters through phenome-wide association analysis, network analysis and colocalization. Through heritability enrichment analysis, we highlight the role of extracardiac tissues in disease etiology. We then examine the differential associations of upstream risk factors with HF subtypes using Mendelian randomization. These findings extend our understanding of the mechanisms underlying HF etiology and may inform future approaches to prevention and treatment.

Genome-wide association studies (GWAS) of HF have provided valuable translational insights, yet existing studies are often limited by phenotypic heterogeneity and a lack of stratification by etiology[1–3]. HF often develops following significant risk factor exposures, such as myocardial infarction and valvular heart disease[4]. We hypothesized that a study of HF occurring without major antecedent risk factors (specifically, myocardial infarction, coronary revascularization or marked structural or valvular heart diseases) could better capture genetic factors directly influencing HF susceptibility[5] (Extended Data Fig. 1 and Supplementary Fig. 1). We refer to this form of HF as ni-HF. Where data permitted, ni-HF cases were further stratified by left ventricular systolic dysfunction, defining those with left ventricular ejection fraction (LVEF) above or below 50% as ni-HFpEF or ni-HFrEF, respectively.

We performed GWAS meta-analyses to identify genomic loci, prioritize putative effector genes and characterize the key pathways, tissues and cell types involved in etiology. We further investigated etiologic clusters through a pleiotropy scan and used Mendelian randomization (MR) to assess the genetic evidence supporting risk factor associations. Our study advances the understanding of the genetic architecture of HF and its subtypes.

## Results
### GWAS meta-analysis
We conducted a GWAS meta-analysis of HF and its subtypes across 42 studies participating in the Heart Failure Molecular Epidemiology for Therapeutic Targets (HERMES) consortium. The study population

✉e-mail: t.lumbers@ucl.ac.uk

comprised 1,946,349 individuals, including 153,174 diagnosed cases of HF (hereafter termed $HF_{all}$), of which 44,012 had a nonischemic etiology (ni-HF). Among these, 5,406 were classified as ni-HFrEF (LVEF < 50%) and 3,841 were classified as ni-HFpEF (LVEF ≥ 50%; Supplementary Figs. 1 and 2). Antecedent myocardial infarction was reported for 29% of the $HF_{all}$ cases and <1% of the ni-HF phenotypes (Supplementary Table 1). The identified HF cases encompassed five major ancestry groups: 139,533 (91%) European; 9,413 (6.2%) East Asian; 3,292 (2.2%) African; 779 (0.5%) South Asian; and 157 (0.1%) admixed American (Supplementary Table 2).

We performed a cross-ancestry, fixed-effect inverse-variance-weighted (IVW) meta-analysis to investigate the associations of 10,199,961 genetic variants (minor allele frequency (MAF) >1%) with the risk of HF and its subtypes. We identified 59 conditionally independent (sentinel) genetic variants associated with $HF_{all}$ at $P < 5 \times 10^{-8}$, at 56 nonoverlapping genomic loci (distance > 500 kb; Fig. 1 and Supplementary Table 3). We did not observe heterogeneity of effects between ancestry groups, except for one sentinel variant at the *LPA* locus ($P_{het-ancestry} < 0.05/59$; Supplementary Fig. 3 and Supplementary Table 4). Ten additional variants were associated with ni-HF subtypes but not with $HF_{all}$. In total, 66 independent genomic loci were identified across the HF phenotypes at $P < 5 \times 10^{-8}$, of which 46 (70%) loci passed a multiplicity-adjusted threshold of $P < 5 \times 10^{-8}/4$ phenotypes. Of the 66 identified loci, 46 (70%) were associated with at least one nonischemic subtype at adjusted $P < 0.05/66$ (hereafter termed ni-HF loci). The remaining 20 loci were associated with $HF_{all}$ but not with the nonischemic subtypes (hereafter termed other HF loci; Fig. 2c). We identified 37 variants not previously reported and successfully replicated 76 of 87 (87%) previously reported HF variants available in our study ($P < 0.05/87$; Supplementary Fig. 4 and Supplementary Table 5)[1–3,6,7].

To investigate sex as a potential modifier of variant effects, we conducted a meta-regression analysis of sex ratio (proportion of male sex) and additive effects of risk-increasing alleles for 66 identified HF sentinel variants across the participating studies. Although limited by statistical power, we observed nominally significant evidence of effect modification by sex at 3 of the 66 loci ($P < 0.05$; Supplementary Table 6).

## Genetic architecture and heritability

The genetic architecture of HF was found to be highly polygenic, evidenced by an elevated genomic inflation factor ($\lambda_{GC} = 1.22$) in the absence of population stratification (linkage disequilibrium score (LDSC) regression intercept = 1.01; Supplementary Figs. 5 and 6 and Supplementary Table 7). We observed an exponential relationship between allele frequency and effect size for the associated variants (Extended Data Fig. 2). The estimated proportion of variance in disease liability explained by common genetic variants under an additive model, or SNP-based heritability ($h^2_g$), was 5.4 ± 0.2% for $HF_{all}$, 6.1 ± 0.5% for ni-HF, 11.8 ± 2.6% for ni-HFrEF and 1.8 ± 1.3% for ni-HFpEF (Extended Data Fig. 3)[8]. Positive genetic correlations were observed across between all HF subtype pairs, with estimates ranging from 0.42 (s.e. = 0.18) between ni-HFrEF and ni-HFpEF to 0.93 (s.e. = 0.15) between ni-HF and ni-HFpEF (Supplementary Fig. 7). We derived a polygenic score ($PGS_{HF}$) from the $HF_{all}$ GWAS, excluding UK Biobank (UKB) participants, and evaluated its association with HF in UKB. Among 347,235 UKB European participants (13,793 $HF_{all}$ cases), the $PGS_{HF}$ was associated with $HF_{all}$ (odds ratio (OR) per PGS s.d. = 1.37 (95% confidence interval (CI) = 1.35–1.39), $P < 2 \times 10^{-16}$), after adjusting for sex, age and first ten genetic principal components (PCs). At a 5% false positive rate, 19% of the cases were identified. Individuals in the top decile of $PGS_{HF}$ had 1.70-fold higher odds of developing heart failure compared to those in the fifth decile (OR = 1.70, 95% CI = 1.59–1.82, $P = 1.7 \times 10^{-142}$) and 2.89-fold compared to those in the bottom decile (OR = 2.89, 95% CI = 2.66–3.14, $P = 2 \times 10^{-51}$; Supplementary Table 8 and Extended Data Fig. 4).

## Prioritization of effector variants, genes and pathways

We then examined the functional properties of variants and genes within each GWAS locus. Through functionally informed fine mapping, we identified 70 credible sets containing 547 putative causal variants at 47/66 HF loci (cumulative posterior inclusion probability (PIP) >0.95)[9]. Eleven fine-mapped variants, including exonic variants in established dilated cardiomyopathy (DCM) genes *FLNC*, *BAG3* and *HSPB7*, had high predicted deleteriousness (Combined Annotation-Dependent Depletion (CADD) Phred score > 20)[10–13] (Extended Data Fig. 5 and Supplementary Table 9). Next, we used a two-stage approach to identify potential effector genes by evaluating 758 protein-coding genes located within heart failure loci. In the first stage, we identified candidate genes ranked highest by one or more of the following gene prioritization methods: (1) variant-to-gene (V2G) scoring of fine-mapped variants[14], (2) gene-level polygenic priority scores (PoPS)[15] and (3) association of predicted gene expression with $HF_{all}$ in multitissue transcriptome-wide association study (TWAS)[16]. In the second stage, we nominated a single best-prioritized gene for each of the 66 loci based on the following: (1) predicted enhancer-gene activity using the activity-by-contact (ABC) model[17,18], (2) colocalization of gene expression in a relevant tissue[19] and (3) association with a phenotypically relevant Mendelian disorder[20] (Fig. 2 and Supplementary Table 10). Results for regional genetic associations, gene prioritization scores, cross-trait association and study-level estimates are provided in Supplementary Data 1.

The prioritized genes in ni-HF loci included those implicated in cardiac development and cardiomyopathy (*ACTN2*, *BAG3*, *CAMKD2* (ref. 21), *CAND2* (ref. 22), *FLNC*, *ACTN2* and *HSPB7* (refs. 11–13,23), *CAMKD2* (ref. 21), *NKX2-5* and *STRN*[24]), cardiac hypertrophy (*CAMKD2* (ref. 21) and *CAND2* (ref. 22)) and cardiac arrhythmia (*PITX2*, *KLF12* and *ATP1B1* (refs. 25–27)). Genes with an established role in atherosclerosis (*LDLR*[28], *LPL*[29], *ABCG5* (refs. 30,31) and *LPA*[32]) were notable among loci associated with $HF_{all}$ but not with ni-HF.

We also performed an exploratory enrichment analysis of the candidate gene sets to identify additional pathways and cellular components associated with $HF_{all}$ and ni-HF (insufficient genes were identified for ni-HFrEF and ni-HFrEF; Extended Data Fig. 6 and Supplementary Table 11). Cellular growth and senescence pathways were highly enriched for both $HF_{all}$ and ni-HF. Notably, a key senescence-associated secretory phenotype (SASP) gene, insulin-like growth factor binding protein 7 (*IGFBP7*), was prioritized at one of the ni-HFpEF loci. *IGFBP7* encodes a circulating anti-angiogenic factor produced by the vasculature, which has recently been implicated in cardiomyocyte senescence and cardiac remodeling[33]. The glucocorticoid pathway was enriched across all HF phenotypes, and genes associated with the sarcomeric Z-disk were overrepresented in both $HF_{all}$ and ni-HF.

## Identifying organs, tissues and cells involved in etiology

To identify the biological components contributing to HF etiology, we performed a heritability enrichment analysis across 206 tissues and cell types, representing 12 organs and systems (Fig. 3 and Supplementary Table 12). We observed enrichment of 46 unique tissues and cell types (one-sided $P < 0.05$), highlighting the diverse range of organ systems involved in HF etiology. Overall, cardiac tissues were the most frequently enriched (15 enrichments), followed by musculoskeletal/connective tissues (12 enrichments) and nervous system tissues (8 enrichments). Cardiac tissues showed the highest enrichment for $HF_{all}$, ni-HF and ni-HFrEF, whereas for niHFpEF, the kidney and pancreas showed the highest enrichment. We then investigated the contribution of 14 cardiac cell populations using single-nucleus transcriptomics derived from 185,185 nuclei from 16 nonfailing human hearts. Heritability enrichment and gene-based association enrichment using MAGMA[34] revealed cardiomyocytes as the major effector cell type for ni-HF. Additionally, vascular smooth muscle cells were also enriched for $HF_{all}$ but not for ni-HF (Supplementary Fig. 8 and Supplementary Table 13).

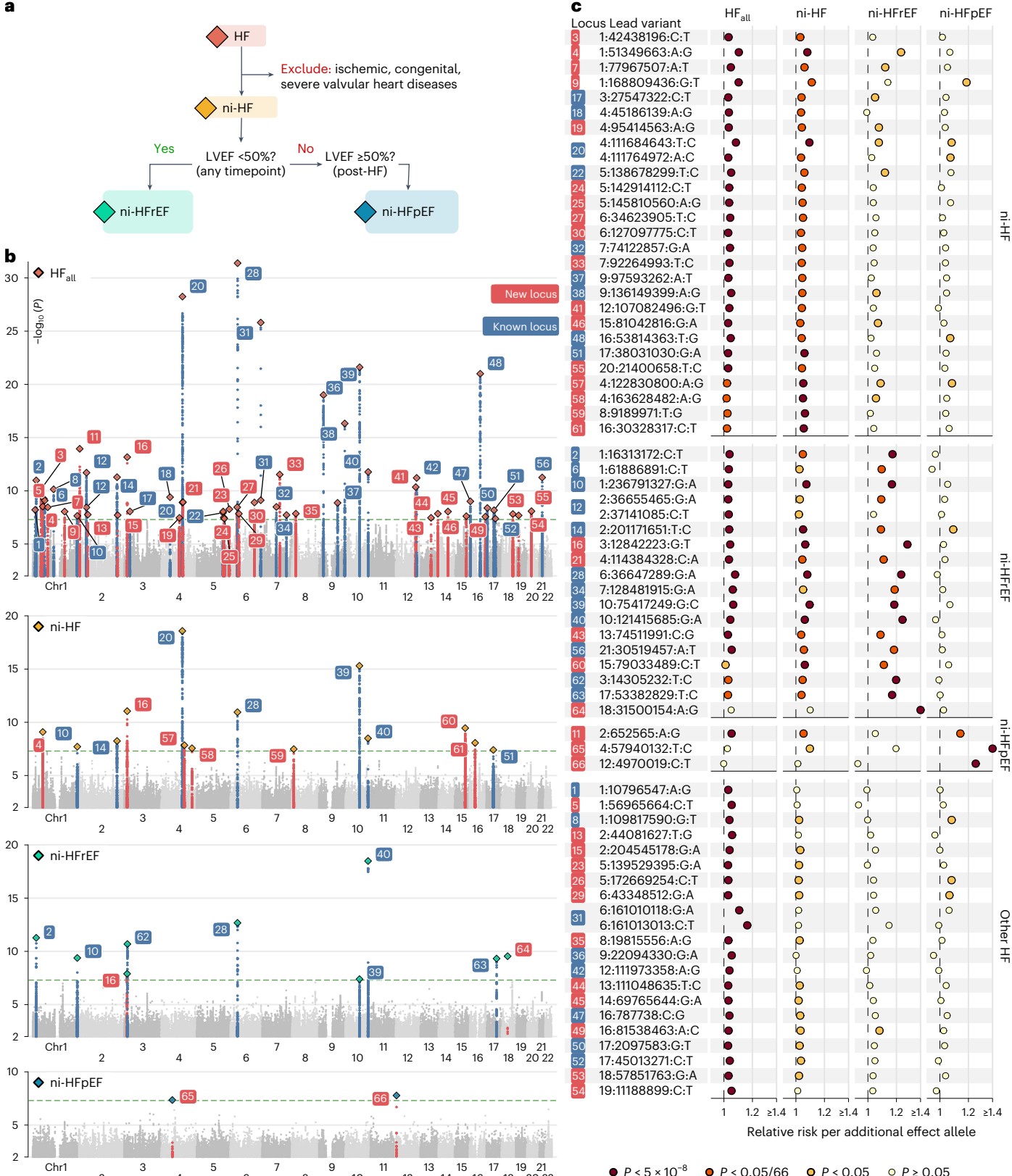

**Fig. 1 | GWAS meta-analysis results across heart failure phenotypes.**
**a**, Phenotyping schema. **b**, Manhattan plots of four HF subtypes showing
$-\log_{10}(P)$ for genetic associations from GWAS meta-analysis ($y$ axis) across
genetic variants ordered by chromosome and base pair positions ($x$ axis).
**c**, Summary of conditionally independent lead variants effect across HF
phenotypes. Lead variants are denoted using chromosome, base pair position

according to the GRCh37 assembly, risk-increasing allele in HF phenotype with
the lowest $P$ value for association and the other allele. Loci are categorized by the
strength of genetic associations across HF phenotypes, with labels on the right
edge of the plot. The presented $P$ values are derived from two-sided association
tests as described in Methods. Chr1, chromosome 1.

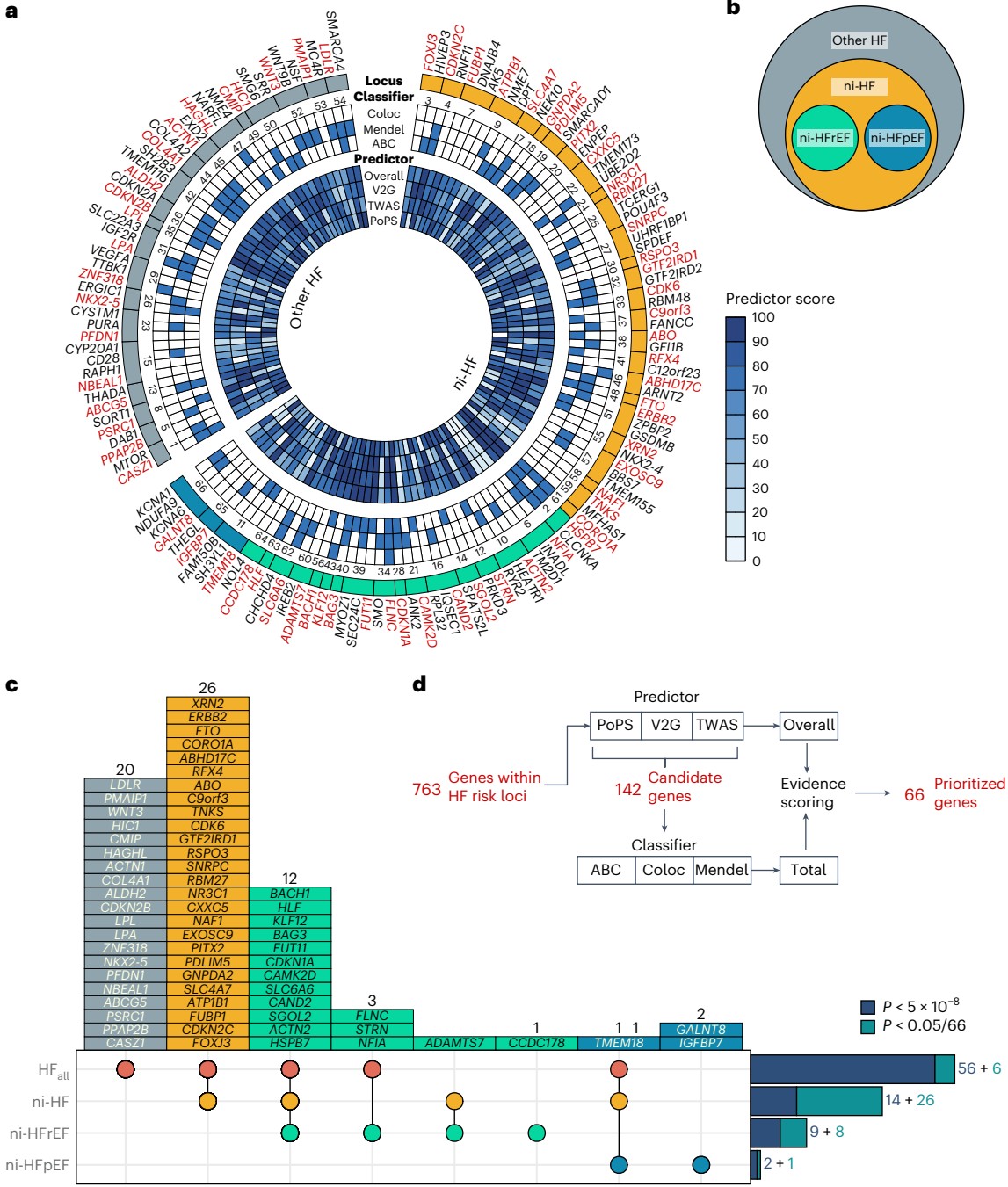

**Fig. 2 | Prioritization of effector genes across heart failure genetic susceptibility loci. a**, Circular heatmap showing predictor score and Boolean classifier scores for 142 candidate genes across HF susceptibility loci. Blue tiles indicate a 'true' value and prioritized genes within each locus are highlighted in red. Coloc refers to colocalization evidence due to shared causal variants with gene expression level in the most relevant tissue; Mendel indicates overlap with Mendelian disease genes; ABC refers to the ABC measure for enhancer-gene activity of overlapping lead variants and their proxies or fine-mapped variants; overall refers to the weighted average of PoPS, TWAS statistics and V2G scaled predictor score (with 2:1:2 weight ratio). **b**, Schematic diagram of locus categorization by genetic associations across HF subtypes. **c**, Upset plot showing number of loci per phenotype category. **d**, Schematic diagram of effector gene prioritization.

To investigate whether genes prioritized at the identified HF loci were differentially regulated in failing hearts, we compared the cell-type transcriptomic profile of the nonfailing heart samples with those from 28 DCM and hypertrophic cardiomyopathy (HCM) patients (*n* nuclei = 344,797). We found that 30/95 (30%) of ni-HF genes were differentially regulated, predominantly within fibroblasts and cardiomyocytes (Extended Data Fig. 7 and Supplementary Table 14). Notably, nuclear receptor subfamily 3 group C member 1 (*NR3C1*),

which encodes glucocorticoid receptors, was upregulated in 12 of 14 cardiac cell types from failing hearts. This upregulation aligns with the glucocorticoid pathway enrichment among the prioritized genes identified at HF GWAS loci. These findings suggest that glucocorticoid signaling may have a cardioprotective role, potentially through the inhibition of pathological mineralocorticoid receptor signaling[35], which could help explain, in part, the beneficial effects of mineralocorticoid receptor antagonist drugs in HF[36]. Another pro-survival HF gene, *KLF12*,

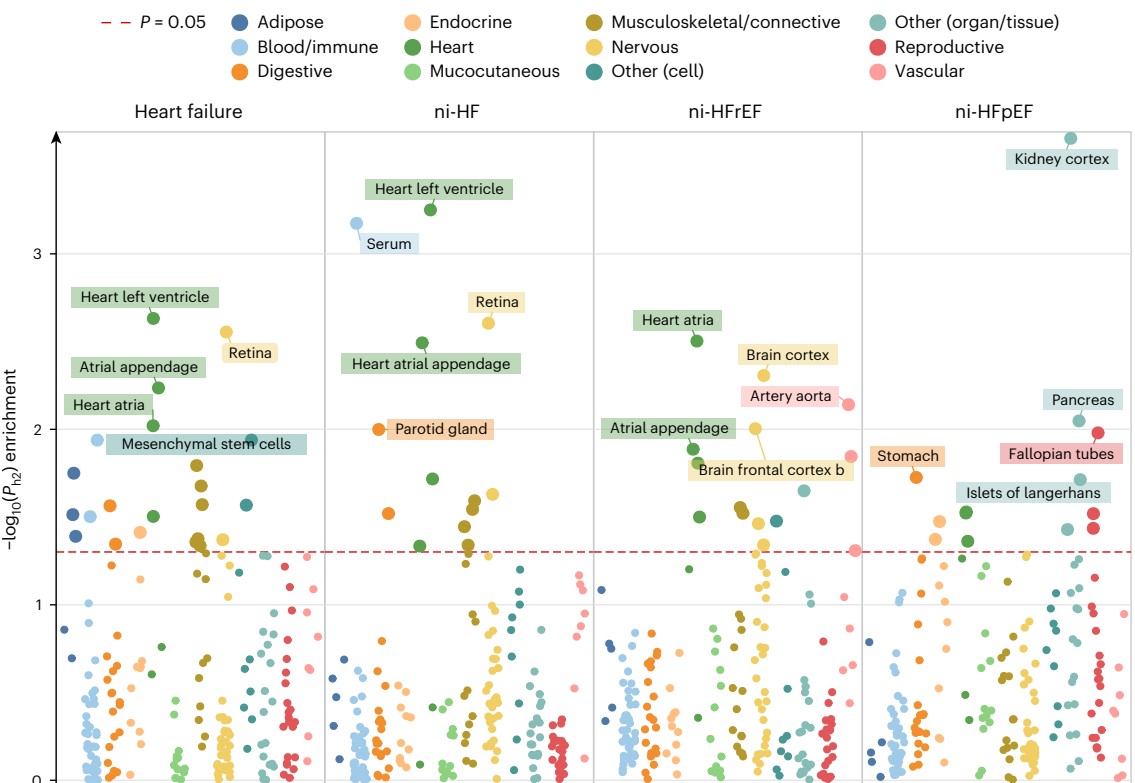

**Fig. 3 | Heritability enrichment of tissues and cell types across heart failure phenotypes.** Estimates for heritability enrichment of 206 tissues and cell types in 12 system/organ categories across four HF phenotypes are presented. Top five tissues/cell types per phenotype are highlighted with labels. The presented $-\log_{10}(P)$ is derived from a one-sided statistical test of heritability enrichment using LDSC-SEG as described in Methods.

was upregulated in both cardiomyocytes and cardiac fibroblasts. KLF12 is a transcription factor that upregulates p53, which in turn leads to upregulation of p21 (*CDKN1A*)[37]. Activation of p53 has been implicated as an important mechanism in *LMNA* cardiomyopathy[38]. In contrast to prior reports, we did not observe differential expression of *IGFBP7* in cardiomyocytes from failing hearts[33].

**Mapping genetic loci to etiologic clusters**

To explore potential mechanisms by which HF genetic loci influence disease risk, we conducted a pleiotropy scan to examine associations between the identified loci and 294 clinical traits and diseases systematically ascertained from UKB[39]. We identified 207 pleiotropic associations at false discovery rate (FDR) < 1% (corresponding to *P* < 0.001) between 46/66 (70%) HF loci and 79 (27%) of the phenotypes tested (Extended Data Fig. 8, Supplementary Fig. 9 and Supplementary Table 15). To visualize these associations, we constructed a pleiotropy graph network by representing loci and diseases as nodes and locus–phenotype associations as edges (Extended Data Fig. 9a). Using a community detection algorithm, we identified 18 etiological clusters of locus–disease associations (Fig. 4, Extended Data Fig. 9b and Supplementary Table 16). The largest cluster, cluster 1, included traits and diseases related to atherosclerosis, with loci that were not associated with ni-HF subtypes. Cluster 2 encompassed cardiac arrhythmias and DCM and loci associated with ni-HF. Cluster 3 centered on the pleiotropic effects of a locus prioritizing *ALDH2*, which encodes an enzyme involved in removing toxic acetaldehydes, with deficiencies in ALDH2 being linked to increased risks of cancer and cardiovascular diseases[40]. The sentinel variant at the corresponding locus has been associated with reduced *ALDH2* expression in blood[41]. Cluster 4 was centered around hypertension, a major risk factor for all HF subtypes. Finally, cluster 5 comprised ni-HF and ni-HFpEF loci associated with adiposity,

diabetes and carpal tunnel syndrome, which are notable clinical risk factors for HFpEF.

We then investigated the genetic associations of sentinel variants across 66 HF loci with 24 related diseases, lifestyle factors and quantitative traits related to cardiovascular physiology and cardiac structure and function. This analysis revealed 207 pleiotropic associations at FDR < 1% (Extended Data Fig. 10 and Supplementary Table 17). Notably, 24 of the 66 HF loci (36%) were not associated with any of the traits investigated ($P \geq 5 \times 10^{-8}$). Cross-trait colocalization analysis revealed evidence for shared causal variants for 105 locus–phenotype pairs (posterior probability of shared causal variants > 0.8; Fig. 5 and Supplementary Table 18). For example, the BTB domain and CNC homolog 1 (*BACH1*) ni-HFrEF locus colocalized with both coronary artery disease (CAD) and DCM. BACH1, a transcription factor induced by mechanical stress in cardiac fibroblasts and endothelial cells, regulates Yes1-associated transcriptional regulator (*YAP*) expression by binding to its promoter[42]. The BACH1–YAP transcriptional network plays a crucial role in cardiac regeneration by modulating proliferative programs in cardiomyocytes[43]. Additionally, six HF loci colocalized with either estimated glomerular filtration rate (eGFR) or chronic kidney disease (CKD), highlighting the key role of the cardio–renal axis in HF. The prioritized gene at one of these loci, *PFDN1*, encodes prefoldin subunit 1, a molecular chaperone involved in protein folding. This gene has been associated with mortality and cardiovascular phenotypes in mouse knockout models[44,45].

**Estimation of risk factor effects on heart failure**

Previous genetic studies have identified certain exposures as potential causal factors for HF; however, these studies have been limited in the range of traits examined and characterization of their association with HF subtypes[1–3]. To address this gap, we investigated the potential

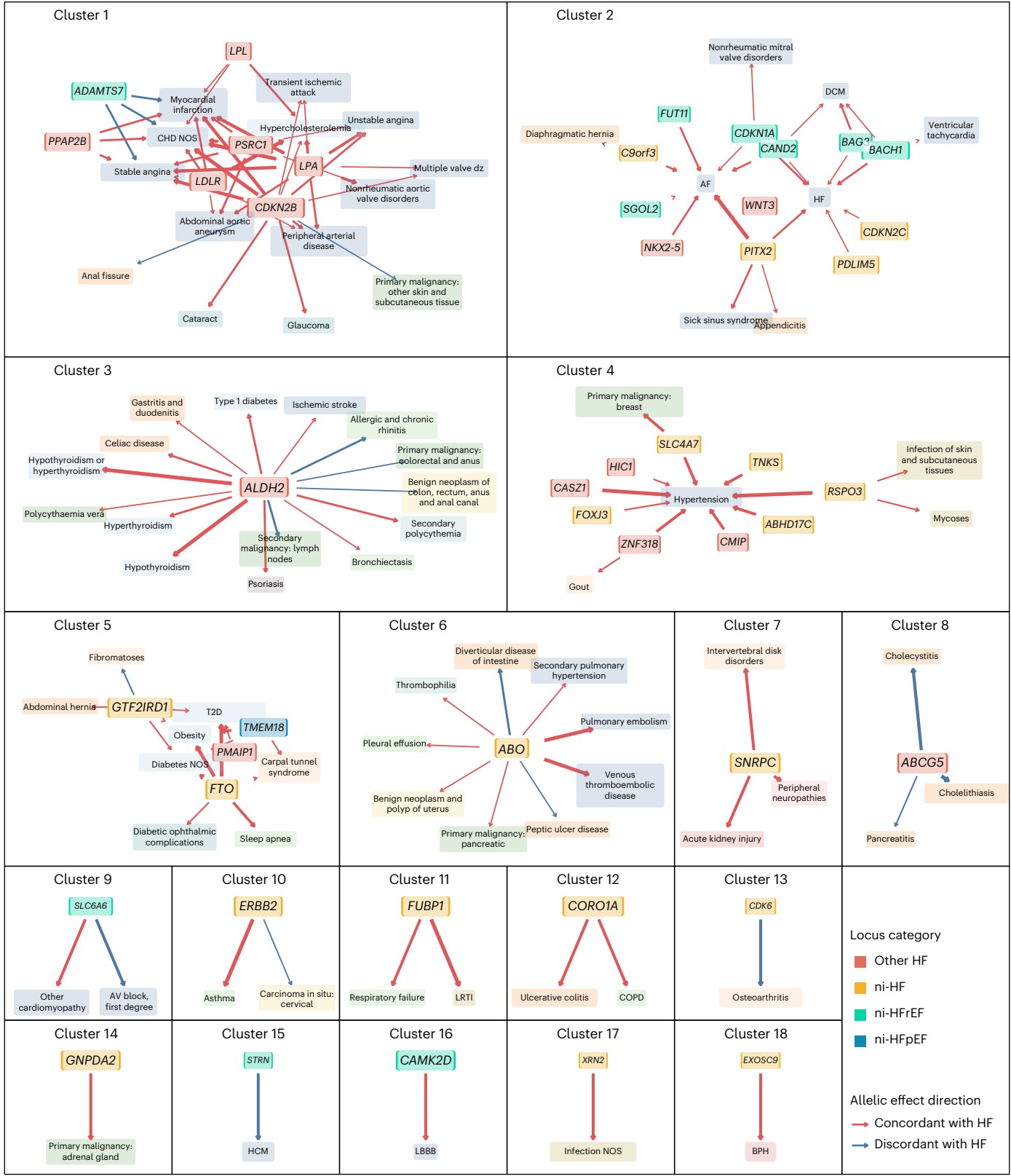

**Fig. 4 | Etiologic clusters of heart failure identified through pleiotropy network analysis.** The network is constructed from 207 genotype–phenotype associations across 79 unique diseases and 46 HF susceptibility loci identified from phenome-wide association (PheWAS) analysis in UKB at FDR <1%. Nodes represent genetic loci labeled by the prioritized gene (solid background with bold–italic label, colored by categorical association across HF phenotypes) and phenotypes (translucent background, colored by phenotype category), sized proportionally to centrality measure. Edges (arrows connecting locus nodes to phenotype nodes) represent the association, with thickness representing the strength of association

measured by absolute $z$ score. The full pleiotropy network and phenotype category color codes are presented in Extended Data Fig. 9. Phenotype abbreviations, phenotype categories and phenome-wide association results are presented in Supplementary Table 15 and Extended Data Fig. 8. Cluster membership and centrality measures of nodes are presented in Supplementary Table 26 and Supplementary Fig. 13. CHD, coronary heart disease; NOS, not-otherwise-specified (nonspecific cause), AF, atrial fibrillation; AV block, atrioventricular block; LRTI, lower respiratory tract infection; COPD, chronic obstructive pulmonary disease; LBBB, left bundle branch block; BPH, benign prostatic hyperplasia.

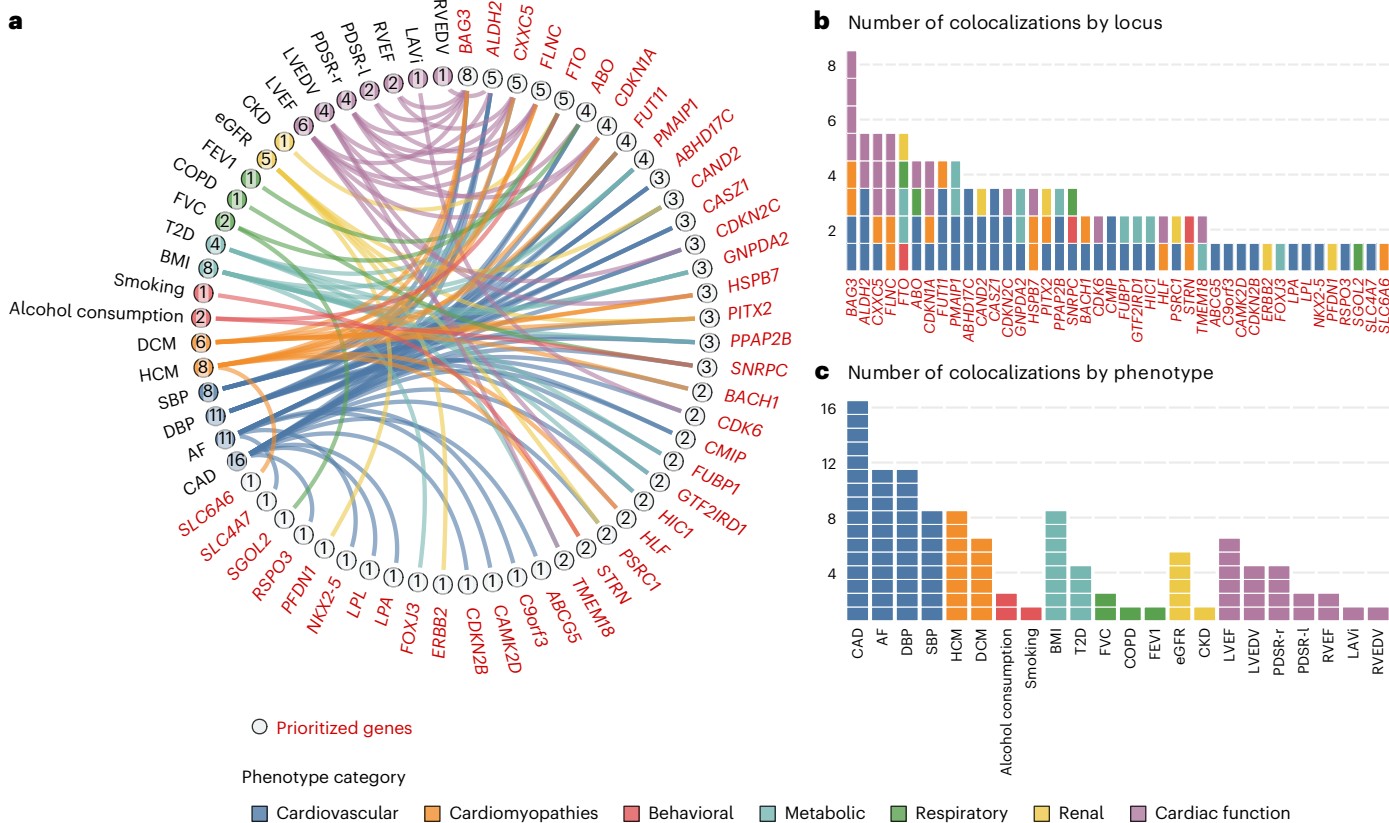

**Fig. 5 | Genetic colocalization between the overall heart failure phenotype and related phenotypes. a**, Chord diagram showing connections between 22 (of 24 tested) related phenotypes across 42 (of 66) HF susceptibility loci with posterior probability of shared causal variants ($PP_{colocH4}$) > 0.8. Each band connects a locus to a phenotype, representing the sharing of causal genetic variants (colocalization) between the tested phenotype and HF at the locus. **b**, Total number of colocalized phenotypes across HF susceptibility loci.

**c**, Total number of HF susceptibility loci with genetic colocalization across tested phenotypes. DBP, diastolic blood pressure; FVC, forced vital capacity; FEV₁, forced expiratory volume in 1 s; LVEDV, left ventricular end diastolic volume; PDSR-r, peak diastolic strain rate radial; PDSR-l, peak diastolic strain rate longitudinal; RVEF, right ventricular ejection fraction; RVEDV, right ventricular end diastolic volume; LAVi, left atrial volume index.

role of 24 related phenotypes (used in the etiologic cluster mapping above) in the etiology of HF and its subtypes. First, we estimated the genetic correlation ($r_g$) between the HF phenotypes and the exposures of interest using bivariate LDSC. We found evidence of shared additive genetic effects for 21/24 phenotypes, with $r_g$ values ranging from −0.25 to 0.67 (Fig. 6 and Supplementary Table 19). Next, we estimated the associations using MR with an IVW fixed effects model and performed sensitivity analyses using MR–Egger and the weighted median estimator (WME). Overall, 18 of the 24 of the exposures were associated with at least one HF subtype (FDR < 1% of IVW estimate, with directional concordance in sensitivity analyses; Fig. 6, Supplementary Figs. 10–12 and Supplementary Table 20).

The estimated effects of risk factors on HF_all and ni-HF outcomes were similar, with the exception of CAD. Consistent with prior studies, liability to CAD was associated with HF_all (MR OR per doubling prevalence with IVW estimator ($OR_{MR-IVW}$) = 1.20, 95% CIs = 1.18–1.22), but not with ni-HF ($OR_{MR-IVW}$ = 1.02, 1.00–1.04)[1]. We found weak evidence supporting a causal effect of liability to type 2 diabetes (T2D) on HF_all ($OR_{MR-IVW}$ = 1.04, 1.03–1.05 per doubling prevalence) as previously reported[1,46]; however, evidence of horizontal pleiotropy was observed ($P_{MR-Egger\ intercept} = 1 \times 10^{-6}$), and the association was attenuated when estimated using pleiotropy-robust methods ($OR_{MR-Egger}$ = 0.98, 0.96–1.00, $OR_{MR-WME}$ = 1.01, 1.00–1.02 per doubling prevalence). We did not find evidence supporting a causal T2D effect on ni-HF risk ($OR_{MR-IVW}$ = 1.01, 0.99–1.02 per doubling prevalence).

Despite the kidney being prioritized in heritability enrichment analyses and colocalization of HF phenotypes with renal traits,

neither lower eGFR or CKD were associated with ni-HFpEF in MR analysis (Figs. 3 and 5) and there was limited genetic correlation between CKD and HF_all ($r_g$ = 0.17, $P$ = 0.0002)[47,48]. In MR analysis, genetically predicted high systolic blood pressure (SBP) and higher body mass index (BMI) were associated with risk of all HF phenotypes, with the effect estimates differing between ni-HF subtypes stratified by LVEF; SBP had the largest effect on ni-HFrEF ($OR_{MR-IVW}$ = 1.92, 1.70–2.17 per s.d., $P_{interaction\ versus\ ni-HFpEF\ estimate}$ = 5.7 × 10⁻⁷), while BMI had the greatest magnitude of effect on ni-HFpEF ($OR_{MR-IVW}$ = 1.87, 1.70–2.06 per s.d., $P_{interaction\ versus\ ni-HFrEF\ estimate}$ = 0.002).

We identified potential causal effects of liability to chronic obstructive pulmonary disease ($OR_{MR-IVW}$ = 1.10, 95% CI = 1.06–1.14 per doubling prevalence) and lower forced expiratory volume on risk of HF ($OR_{MR-IVW}$ = 0.91, 95% CI = 0.88–0.95 per s.d.) and risk of ni-HF ($OR_{MR-IVW}$ = 0.91, 95% CI = 0.88–0.95 per s.d.). We also found evidence for effects of smoking behavior on HF ($OR_{MR-IVW}$ = 1.26, 95% CI = 1.19–1.33 per s.d. pack-years of smoking). Given the prioritization of *ALDH2* at an identified GWAS locus for HF and ni-HF, we estimated the effects of higher alcohol consumption on HF risk. We found evidence of a positive association between higher alcohol consumption and risk of both HF ($OR_{MR-IVW}$ = 1.17, 95% CI = 1.06–1.29 per s.d. drinks per week) and ni-HFrEF ($OR_{MR-IVW}$ = 1.87, 95% CI = 1.37–2.55 per s.d. drinks per week), consistent with clinical studies that identify alcohol as a risk factor for dilated cariomyopathy[49].

In MR analyses, lower LVEF, a trait linked to impaired myocardial contractility, was associated with a higher risk of all HF phenotypes except for ni-HFpEF (Fig. 6 and Supplementary Fig. 12). The strongest

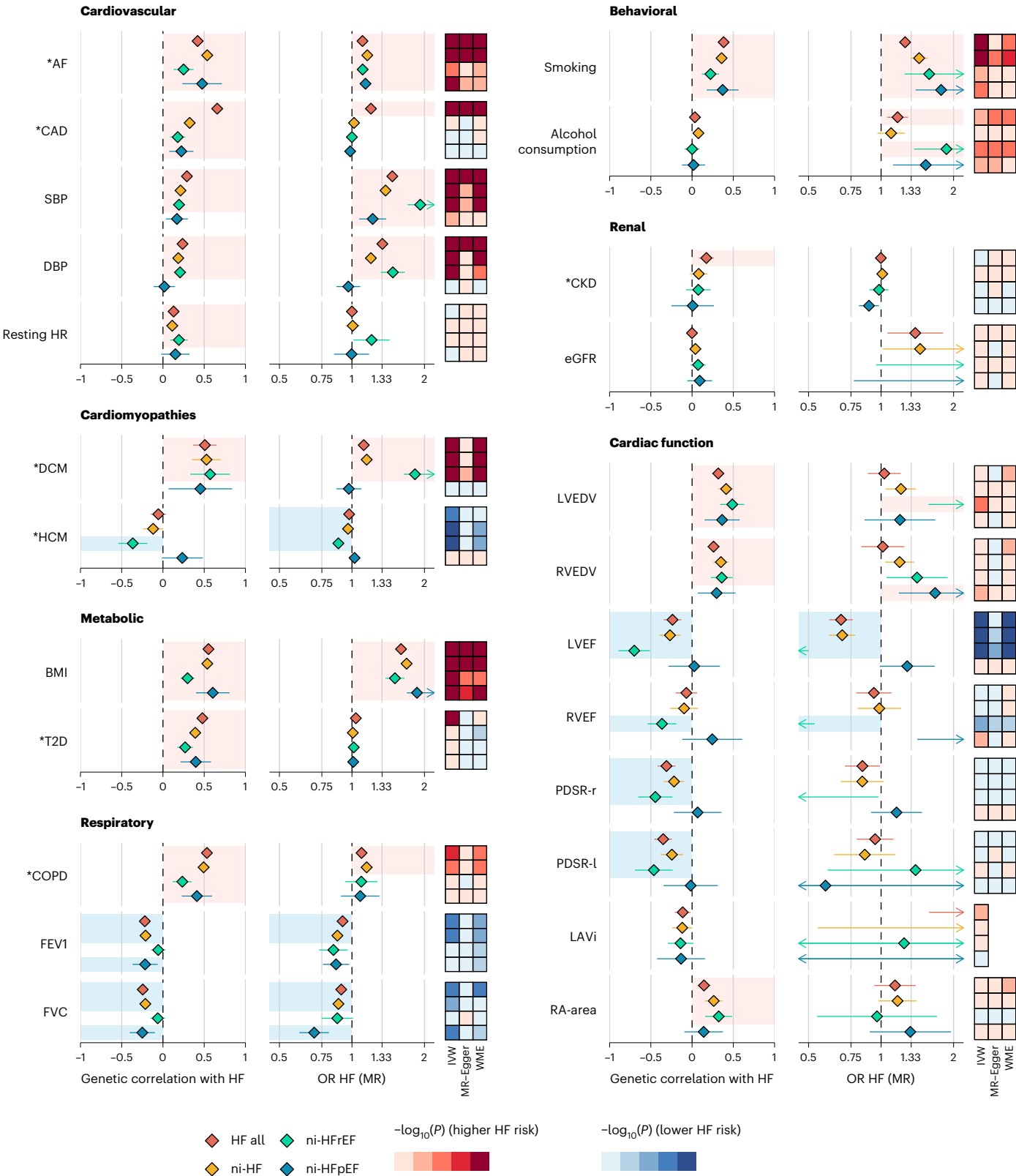

**Fig. 6 | Genetic correlation ($r_g$) and MR estimates across 24 traits and four heart failure phenotypes.** Asterisks (*) indicate binary traits. MR effect estimates are reported as OR (OR$_{MR}$) per doubling prevalence for binary traits or per s.d. increase for quantitative traits. Estimates that were robust to multiple-testing adjustment at FDR <1% and sensitivity analyses were indicated by light blue shade (for $r_g$ < 0 and OR$_{MR}$ < 1) or light red shade (for $r_g$ > 0 and OR$_{MR}$ > 1). The heatmaps represent two-sided $P$ values for associations from different MR models, color-coded with the direction of MR estimates and strength of associations. Diamonds and whiskers on the forest plots represent point estimates and 95% CIs for $r_g$ (left) and OR$_{MR}$ (right). Missing diamonds/point whiskers with arrows represent point estimates/95% CIs outside the scale. HR, heart rate; RA-area, right atrial area.

association was observed with ni-HFrEF ($OR_{MR-IVW}$ = 0.18, 0.12–0.27 per s.d. LVEF; $OR_{MR-IVW}$ = 1.83, 1.64–2.03 per doubling prevalence of DCM). Conversely, a higher liability to hypertrophic cardiomyopathy, a condition linked to increased myocardial contractility, was protective against ni-HFrEF ($OR_{MR-IVW}$ = 0.88, 0.85–0.91 per doubling prevalence of HCM). These findings are consistent with the reported opposing genetic associations between dilated and HCM and with observational studies reporting divergent association between lower and higher LVEF and the risk of HFrEF and HFpEF, respectively[50,51]. Despite the widely held hypothesis that diastolic dysfunction is the key driver for HFpEF, we did not find evidence supporting a causal association ($OR_{MR-IVW}$ for ni-HFpEF = 1.16, 0.91–1.48 per s.d. peak diastolic strain rate radial), although the limited set of diastolic parameters investigated are not established as clinical indices of diastolic dysfunction[52].

## Discussion

We report a GWAS of heart failure in 1.9 million individuals, including over 150,000 cases, with a focus on HF subtypes defined by etiology and LVEF. Our analysis identifies 66 distinct genetic loci associated with HF, including 37 new loci, with specific associations identified for ni-HFpEF. Through a comprehensive framework integrating functional genomic data at the variant, locus and genome-wide levels, we uncover new genes and pathways underlying HF and its subtypes. We leverage genetic association data to map the organs, tissues and cells contributing to HF pathogenesis and estimate subtype-specific effects of key risk factors.

Our study focusses on HF occuring in the absence of major secondary causes, specifically, previous myocardial infarction, revascularization, congenital heart disease and significant valvular heart disease. By focusing on cases without these common upstream drivers, our findings highlight mechanisms intrinsic to HF, including cardio–renal mechanisms and primary cardiomyopathies[53]. The ni-HFrEF phenotype captures both HF with mildly reduced (LVEF = 41–49%) and reduced (LVEF < 40%)[54] ejection fraction and is enriched for DCM. Among the subtypes analyzed, this phenotype showed the highest heritability. In contrast, ni-HFpEF had low heritability but significant enrichment for renal tissues, reflecting a more heterogeneous phenotype consistent with a multisystem disorder.

Our comprehensive appraisal of the genetic and phenotypic etiology of HF and its subtypes offers key insights with tangible translational implications. First, by integrating functional genomic data to identify likely causal variants, genes and pathways, we identify new loci and genes, as well as prioritizing alternative candidate genes at previously reported HF loci. Molecular mechanisms regulating growth emerged as potentially important, including modulation of *BACH1–YAP*[42] and *KLF12–CDKN1A*[37] activity. Notably, we prioritized *ALDH2* at a previously known HF locus, a gene that has been associated with increased cardiovascular risk via its effect on alcohol metabolism. We also found enrichment of the SASP pathway and identified *IGFBP7* as a potential HFpEF gene. IGFBP7 is a well-established clinical biomarker of acute renal failure and a driver of cell cycle arrest, fibrosis and vascular rarefaction, presenting a potentially promising target for new therapeutics[33,55,56]. Second, our findings underscore the significance of noncardiac tissues in HF etiology. In particular, the kidney, vascular system and metabolic tissues were identified as important for HFpEF, although we did not find evidence of associations between CKD and eGFR in our MR analysis[57]. Third, our appraisal of cardiovascular risk factor effects across HF subtypes yielded several findings with potential implications for preventive strategies. For example, the weak evidence of the causal effect of T2D on HF suggests that observational associations between these conditions might partially stem from shared upstream mechanisms. Notably, certain diabetes treatments, such as SGLT2 inhibitors, have demonstrated substantial benefits in patients with HF without diabetes. This observation may reflect the drugs' effects on common upstream pathways influencing both

HF and diabetes[58]. We also found no evidence supporting the causal effects of CAD on liability to HF in the absence of previous myocardial infarction or revascularization (ni-HF). Finally, we provide new evidence implicating modifiable lifestyle factors, including smoking and alcohol consumption, in HF risk, highlighting potentially actionable targets for prevention[59,60].

Our study has important limitations that suggest directions for future research. Despite efforts to include data from diverse populations, participants of non-European ancestry were underrepresented in our analysis. Additionally, only a minority of HF cases could be stratified by LVEF due to limited data availability, and key measures of ventricular size and diastolic function were largely unavailable. Large prospective studies with detailed phenotypic measurements are particularly needed for HFpEF to enable a more precise characterization of the genetic basis underlying HF subtypes. The variants and genes identified in the discovery analysis require functional validation, and studies with larger sample sizes are needed to improve the precision of effect size estimates. Our meta-regression analysis of sex ratio likely underestimates genetic effect modification by sex due to the limited statistical power and potential confounding by sex-related factors. To address this, sex-stratified genetic association studies are needed to uncover sex-specific effects. In our cross-trait genetic analyses, we note a minor sample overlap between samples in the current GWAS and those used in GWAS of CAD and atrial fibrillation; however, this overlap is unlikely to have significantly biased the MR estimates due to the limited scope of the overlap and the strength of the exposure instruments. Nevertheless, further studies are needed to model relationships between individual risk factors more effectively, incorporating multivariable and mediation analysis.

In summary, we used human genetic analysis to map the etiologies underlying HF and its subtypes, classified by etiology and LVEF. Our findings provide new evidence supporting the concept of HF as a multisystem clinical syndrome and provide insights into the underlying biological mechanisms. These insights may guide the development of more effective strategies for prevention and treatment.

## Online content

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

Albert Henry [1,2], Xiaodong Mo[3], Chris Finan [1], Mark D. Chaffin [4], Doug Speed [5], Hanane Issa[2], Spiros Denaxas[2,6,7,8], James S. Ware [4,9,10,11,12], Sean L. Zheng [9,11], Anders Malarstig[13,14], Jasmine Gratton[1], Isabelle Bond[1], Carolina Roselli [4,15], David Miller[16], Sandesh Chopade[1], A. Floriaan Schmidt [1,17,18], Erik Abner [19], Lance Adams[20], Charlotte Andersson[21,22], Krishna G. Aragam [4,23,24], Johan Ärnlöv[25,26], Geraldine Asselin [27], Anna Axelsson Raja[28], Joshua D. Backman [29], Traci M. Bartz[30], Kiran J. Biddinger[4,23], Mary L. Biggs[30,31], Heather L. Bloom[32], Eric Boersma[33], Jeffrey Brandimarto[34], Michael R. Brown [35], Søren Brunak [36], Mie Topholm Bruun [37], Leonard Buckbinder[14], Henning Bundgaard[28], David J. Carey[38], Daniel I. Chasman[39,40], Xing Chen[14], James P. Cook[41], Tomasz Czuba[42], Simon de Denus[27,43], Abbas Dehghan [44], Graciela E. Delgado[45], Alexander S. Doney[46], Marcus Dörr [47,48], Joseph Dowsett [49], Samuel C. Dudley [50], Gunnar Engström[51], Christian Erikstrup [52,53], Tõnu Esko[4,19], Eric H. Farber-Eger [54], Stephan B. Felix[47,48], Sarah Finer [55], Ian Ford [56], Mohsen Ghanbari [57], Sahar Ghasemi[48,58,59], Jonas Ghouse[60], Vilmantas Giedraitis [61], Franco Giulianini[39], John S. Gottdiener[62], Stefan Gross [47,48], Daníel F. Guðbjartsson [63,64], Hongsheng Gui [65], Rebecca Gutmann[66], Sara Hägg [67], Christopher M. Haggerty[38], Åsa K. Hedman [13], Anna Helgadottir [63], Harry Hemingway [2,6], Hans Hillege[15], Craig L. Hyde [14], Bitten Aagaard Jensen [68], J. Wouter Jukema [69,70], Isabella Kardys[33], Ravi Karra [71,72], Maryam Kavousi [57], Jorge R. Kizer[73], Marcus E. Kleber [45], Lars Køber[74], Andrea Koekemoer [75], Karoline Kuchenbaecker [76,77], Yi-Pin Lai[14], David Lanfear[65,78], Claudia Langenberg [79,80,81], Honghuang Lin [22,82], Lars Lind[83], Cecilia M. Lindgren[4,84,85], Peter P. Liu[86,87,88], Barry London[89], Brandon D. Lowery [54], Jian'an Luan [81], Steven A. Lubitz [4,90], Patrik Magnusson [67], Kenneth B. Margulies[34], Nicholas A. Marston [91], Hilary Martin [92], Winfried März[45,93,94], Olle Melander[95], Ify R. Mordi [46], Michael P. Morley[34], Andrew P. Morris[41,85], Alanna C. Morrison [35], Lori Morton[96], Michael W. Nagle [14], Christopher P. Nelson [75], Alexander Niessner[97], Teemu Niiranen [98,99], Raymond Noordam [100], Christoph Nowak [25], Michelle L. O'Donoghue[91], Sisse Rye Ostrowski [49,101], Anjali T. Owens[34], Colin N. A. Palmer [102], Guillaume Paré [103,104,105], Ole Birger Pedersen [101,106], Markus Perola[99], Marie Pigeyre [105,107], Bruce M. Psaty [108,109], Kenneth M. Rice [30], Paul M. Ridker[39,40], Simon P. R. Romaine[75], Jerome I. Rotter[110,111,112], Christian T. Ruff[91], Marc S. Sabatine[91], Neneh Sallah [2], Veikko Salomaa [99], Naveed Sattar [113], Alaa A. Shalaby[114], Akshay Shekhar[96], Diane T. Smelser[38], Nicholas L. Smith [109,115,116], Erik Sørensen [49], Sundararajan Srinivasan [102], Kari Stefansson[63,117], Garðar Sveinbjörnsson [63], Per Svensson [118,119], Mari-Liis Tammesoo[19], Jean-Claude Tardif [27,120], Maris Teder-Laving [19], Alexander Teumer [48,59,121], Guðmundur Thorgeirsson [63,117,122], Unnur Thorsteinsdottir[63,117], Christian Torp-Pedersen [123], Vinicius Tragante [63], Stella Trompet [69,100], Andre G. Uitterlinden [57,124], Henrik Ullum[125], Pim van der Harst[15,17], David van Heel [126], Jessica van Setten [17], Marion van Vugt [17], Abirami Veluchamy[46,102], Monique Verschuuren [127,128], Niek Verweij[15], Christoffer Rasmus Vissing [28], Uwe Völker [48,129], Adriaan A. Voors[15], Lars Wallentin[130],

Yunzhang Wang [67], Peter E. Weeke[60], Kerri L. Wiggins [31], L. Keoki Williams[65], Yifan Yang[34], Bing Yu [35], Faiez Zannad [131], Chaoqun Zheng [28], Genes & Health Research Team*, Estonian Biobank Research Team*, DBDS Genomic Consortium*, Folkert W. Asselbergs[2,8,18], Thomas P. Cappola [34], Marie-Pierre Dubé [27,120], Michael E. Dunn[96], Chim C. Lang[46], Nilesh J. Samani[75], Svati Shah[71,132,133], Ramachandran S. Vasan [22,134,140], J. Gustav Smith[42,135,136,140], Hilma Holm [63,140], Sonia Shah [3,140], Patrick T. Ellinor [4,90,137,138,140], Aroon D. Hingorani [1,140], Quinn Wells[139,140], R. Thomas Lumbers [2,6,8,140] ✉ & HERMES Consortium*

[1]Institute of Cardiovascular Science, University College London, London, UK. [2]Institute of Health Informatics, University College London, London, UK. [3]Institute for Molecular Bioscience, The University of Queensland, Brisbane, Queensland, Australia. [4]Program in Medical and Population Genetics, The Broad Institute of MIT and Harvard, Cambridge, MA, USA. [5]Center for Quantitative Genetics and Genomics, Aarhus University, Aarhus, Denmark. [6]Health Data Research UK, London, UK. [7]British Heart Foundation Data Science Centre, London, UK. [8]The National Institute for Health Research University College London Hospitals Biomedical Research Centre, University College London, London, UK. [9]National Heart & Lung Institute, Imperial College London, London, UK. [10]MRC London Institute of Medical Sciences, Imperial College London, London, UK. [11]Royal Brompton & Harefield Hospitals, Guy's and St. Thomas' NHS Foundation Trust, London, UK. [12]Hammersmith Hospital, Imperial College Hospitals NHS Trust, London, UK. [13]Cardiovascular Medicine Unit, Department of Medicine Solna, Karolinska Institute, Stockholm, Sweden. [14]Pfizer Worldwide Research & Development, Cambridge, MA, USA. [15]Department of Cardiology, University Medical Center Groningen, University of Groningen, Groningen, the Netherlands. [16]Division of Biosciences, University College London, London, UK. [17]Department of Cardiology, Division of Heart and Lungs, University Medical Center Utrecht, Utrecht, the Netherlands. [18]Department of Cardiology, Amsterdam Cardiovascular Sciences, Amsterdam University Medical Centers, Amsterdam, the Netherlands. [19]Estonian Genome Center, Institute of Genomics, University of Tartu, Tartu, Estonia. [20]Geisinger Health System, Danville, PA, USA. [21]Department of Cardiology, Herlev Gentofte Hospital, Herlev, Denmark. [22]National Heart, Lung and Blood Institute's and Boston University's Framingham Heart Study, Framingham, MA, USA. [23]Cardiovascular Research Center, Massachusetts General Hospital, Boston, MA, USA. [24]Center for Genomic Medicine, Massachusetts General Hospital, Boston, MA, USA. [25]Department of Neurobiology, Care Sciences and Society/Section of Family Medicine and Primary Care, Karolinska Institutet, Stockholm, Sweden. [26]School of Health and Social Sciences, Dalarna University, Falun, Sweden. [27]Montreal Heart Institute, Montreal, Quebec, Canada. [28]Department of Cardiology, The Heart Centre, Copenhagen University Hospital, Rigshospitalet, Copenhagen, Denmark. [29]Analytical Genetics, Regeneron Genetics Center, Tarrytown, NY, USA. [30]Department of Biostatistics, University of Washington, Seattle, WA, USA. [31]Department of Medicine, University of Washington, Seattle, WA, USA. [32]Department of Medicine, Division of Cardiology, Emory University Medical Center, Atlanta, GA, USA. [33]Department of Cardiology, Erasmus University Medical Center, Rotterdam, the Netherlands. [34]Penn Cardiovascular Institute, Perelman School of Medicine, University of Pennsylvania, Philadelphia, PA, USA. [35]Department of Epidemiology, Human Genetics, and Environmental Sciences, The University of Texas School of Public Health, Houston, TX, USA. [36]Novo Nordisk Foundation Center for Protein Research, Faculty of Health and Medical Sciences, University of Copenhagen, Copenhagen, Denmark. [37]Department of Clinical Immunology, Odense University Hospital, Odense, Denmark. [38]Department of Molecular and Functional Genomics, Geisinger, Danville, PA, USA. [39]Division of Preventive Medicine, Brigham and Women's Hospital, Boston, MA, USA. [40]Harvard Medical School, Boston, MA, USA. [41]Department of Biostatistics, University of Liverpool, Liverpool, UK. [42]Department of Molecular and Clinical Medicine, Institute of Medicine, Gothenburg University and Sahlgrenska University Hospital, Gothenburg, Sweden. [43]Faculty of Pharmacy, Université de Montréal, Montreal, Quebec, Canada. [44]MRC-PHE Centre for Environment and Health, Department of Epidemiology and Biostatistics, Imperial College London, London, UK. [45]Vth Department of Medicine (Nephrology, Hypertensiology, Endocrinology, Diabetology, Rheumatology), Medical Faculty of Mannheim, University of Heidelberg, Heidelberg, Germany. [46]Division of Molecular & Clinical Medicine, University of Dundee, Ninewells Hospital and Medical School, Dundee, UK. [47]Department of Internal Medicine B, University Medicine Greifswald, Greifswald, Germany. [48]DZHK (German Center for Cardiovascular Research), Partner Site Greifswald, Greifswald, Germany. [49]Department of Clinical Immunology, Copenhagen University Hospital, Rigshospitalet, Copenhagen, Denmark. [50]Department of Medicine, Cardiovascular Division, University of Minnesota, Minneapolis, MN, USA. [51]Department of Clinical Sciences, Lund University, Malmö, Sweden. [52]Department of Clinical Immunology, Aarhus University Hospital, Aarhus, Denmark. [53]Deparment of Clinical Medicine, Health, Aarhus University, Aarhus, Denmark. [54]Vanderbilt Institute for Clinical and Translational Research, Vanderbilt University Medical Center, Nashville, TN, USA. [55]Centre for Primary Care and Public Health, Wolfson Institute of Population Health, Queen Mary University of London, London, UK. [56]Robertson Center for Biostatistics, Institute of Health and Wellbeing, University of Glasgow, Glasgow, UK. [57]Department of Epidemiology, Erasmus University Medical Center, Rotterdam, the Netherlands. [58]Institute of Genetic Epidemiology, Faculty of Medicine and Medical Center, University of Freiburg, Freiburg, Germany. [59]Department of Psychiatry and Psychotherapy, University Medicine Greifswald, Greifswald, Germany. [60]Department of Cardiology, Rigshospitalet, Copenhagen University Hospital, Copenhagen, Denmark. [61]Department of Public Health and Caring Sciences, Geriatrics, Uppsala, Sweden. [62]Department of Medicine, Division of Cardiology, University of Maryland School of Medicine, Baltimore, MD, USA. [63]deCODE genetics/Amgen Inc., Reykjavik, Iceland. [64]School of Engineering and Natural Sciences, University of Iceland, Reykjavik, Iceland. [65]Center for Individualized and Genomic Medicine Research, Department of Internal Medicine, Henry Ford Hospital, Detroit, MI, USA. [66]Division of Cardiovascular Medicine, University of Iowa Carver College of Medicine, Iowa City, IA, USA. [67]Department of Medical Epidemiology and Biostatistics, Karolinska Institutet, Stockholm, Sweden. [68]Department of Clinical Immunology, Aalborg University Hospital, Aalborg, Denmark. [69]Department of Cardiology, Leiden University Medical Center, Leiden, the Netherlands. [70]Einthoven Laboratory for Experimental Vascular Medicine, LUMC, Leiden, the Netherlands. [71]Department of Medicine, Division of Cardiology, Duke University Medical Center, Durham, NC, USA. [72]Department of Pathology, Duke University Medical Center, Durham, NC, USA. [73]Cardiology Section, San Francisco Veterans Affairs Health System, and Departments of Medicine, Epidemiology and Biostatistics, University of California San Francisco, San Francisco, CA, USA. [74]Department of Cardiology, Nordsjaellands Hospital, Copenhagen, Denmark. [75]Department of Cardiovascular Sciences, University of Leicester and NIHR Leicester Biomedical Research Centre, Glenfield Hospital, Leicester, UK. [76]Division of Psychiatry, University College London, London, UK. [77]UCL Genetics Institute, University College London, London, UK. [78]Heart and Vascular Institute, Henry Ford Hospital, Detroit, MI, USA. [79]Precision Healthcare University Research Institute, Queen Mary University of London, London, UK. [80]Computational Medicine, Berlin Institute of Health (BIH) at Charité—Universitätsmedizin Berlin, Berlin, Germany. [81]MRC Epidemiology Unit, Institute of Metabolic Science, University of Cambridge, Cambridge, UK. [82]Department of Medicine, University of Massachusetts Chan Medical School, Worcester, MA, USA. [83]Department of Medical Sciences, Uppsala University, Uppsala, Sweden. [84]Big Data Institute at the Li Ka Shing Centre for Health Information and Discovery, University of Oxford, Oxford, UK. [85]Wellcome Trust Centre for Human Genetics, University of Oxford, Oxford, UK. [86]University of Ottawa Heart Institute, Ottawa, Ontario, Canada. [87]Cellular and Molecular Medicine, University of Ottawa, Ottawa, Ontario, Canada.

[88]Department of Medicine, University of Ottawa, Ottawa, Ontario, Canada. [89]Division of Cardiovascular Medicine and Abboud Cardiovascular Research Center, University of Iowa, Iowa City, IA, USA. [90]Cardiac Arrhythmia Service and Cardiovascular Research Center, Massachusetts General Hospital, Boston, MA, USA. [91]TIMI Study Group, Division of Cardiovascular Medicine, Brigham and Women's Hospital, Harvard Medical School, Cambridge, MA, USA. [92]Wellcome Sanger Institute, Wellcome Genome Campus, Hinxton, UK. [93]Clinical Institute of Medical and Chemical Laboratory Diagnostics, Medical University of Graz, Graz, Austria. [94]Synlab Academy, Synlab Holding Deutschland GmbH, Mannheim, Germany. [95]Department of Internal Medicine, Clinical Sciences, Lund University and Skåne University Hospital, Malmö, Sweden. [96]Cardiovascular Research, Regeneron Pharmaceuticals, Tarrytown, NY, USA. [97]Department of Internal Medicine II, Division of Cardiology, Medical University of Vienna, Vienna, Austria. [98]Department of Medicine, Turku University Hospital and University of Turku, Turku, Finland. [99]National Institute for Health and Welfare, Helsinki, Finland. [100]Section of Gerontology and Geriatrics, Department of Internal Medicine, Leiden University Medical Center, Leiden, the Netherlands. [101]Department of Clinical Medicine, Faculty of Health and Medical Sciences, University of Copenhagen, Copenhagen, Denmark. [102]Division of Population Health and Genomics, University of Dundee, Ninewells Hospital and Medical School, Dundee, UK. [103]Department of Pathology and Molecular Medicine, McMaster University, Hamilton, Ontario, Canada. [104]Thrombosis and Atherosclerosis Research Institute, Hamilton, Ontario, Canada. [105]Population Health Research Institute, Hamilton, Ontario, Canada. [106]Department of Clinical Immunology, Zealand University Hospital, Køge, Denmark. [107]Department of Medicine, McMaster University, Hamilton, Ontario, Canada. [108]Cardiovascular Health Research Unit, University of Washington, Seattle, WA, USA. [109]Kaiser Permanente Washington Health Research Institute, Kaiser Permanente Washington, Seattle, WA, USA. [110]The Institute for Translational Genomics and Population Sciences, Harbor-UCLA Medical Center, Torrance, CA, USA. [111]Departments of Pediatrics and Medicine, Harbor-UCLA Medical Center, Torrance, CA, USA. [112]Los Angeles Biomedical Research Institute, Harbor-UCLA Medical Center, Torrance, CA, USA. [113]BHF Cardiovascular Research Centre, University of Glasgow, Glasgow, UK. [114]Department of Medicine, Division of Cardiology, University of Pittsburgh Medical Center and VA Pittsburgh HCS, Pittsburgh, PA, USA. [115]Department of Epidemiology, University of Washington, Seattle, WA, USA. [116]Department of Veterans Affairs Office of Research & Development, Seattle Epidemiologic Research and Information Center, Seattle, WA, USA. [117]Department of Medicine, University of Iceland, Reykjavik, Iceland. [118]Department of Cardiology, Söderjukhuset, Stockholm, Sweden. [119]Department of Clinical Science and Education-Södersjukhuset, Karolinska Institutet, Stockholm, Sweden. [120]Faculty of Medicine, Université de Montréal, Montreal, Quebec, Canada. [121]Institute for Community Medicine, University Medicine Greifswald, Greifswald, Germany. [122]Department of Internal Medicine, Division of Cardiology, National University Hospital of Iceland, Reykjavik, Iceland. [123]Department of Epidemiology and Biostatistics, Aalborg University Hospital, Aalborg, Denmark. [124]Department of Internal Medicine, Erasmus University Medical Center, Rotterdam, the Netherlands. [125]Statens Serum Institut, Copenhagen, Denmark. [126]Centre for Genomics and Child Health, Blizard Institute, Queen Mary University of London, London, UK. [127]Department Life Course and Health, Centre for Nutrition, Prevention and Health Services, National Institute for Public Health and the Environment, Bilthoven, the Netherlands. [128]Julius Center for Health Sciences and Primary Care, University Medical Center Utrecht, Utrecht, the Netherlands. [129]Interfaculty Institute for Genetics and Functional Genomics, University Medicine Greifswald, Greifswald, Germany. [130]Uppsala Clinical Research Center, Uppsala University, Uppsala, Sweden. [131]Université de Lorraine, CHU de Nancy, Inserm and INI-CRCT (F-CRIN), Institut Lorrain du Coeur et des Vaisseaux, Vandoeuvre Lès Nancy, France. [132]Duke Clinical Research Institute, Durham, NC, USA. [133]Duke Molecular Physiology Institute, Durham, NC, USA. [134]Sections of Cardiology, Preventive Medicine and Epidemiology, Department of Medicine, Boston University Schools of Medicine and Public Health, Boston, MA, USA. [135]Department of Cardiology, Clinical Sciences, Lund University and Skåne University Hospital, Lund, Sweden. [136]Wallenberg Center for Molecular Medicine and Lund University Diabetes Center, Lund University, Lund, Sweden. [137]Cardiac Arrhythmia Service and Cardiovascular Research Center, Massachusetts General Hospital, Cambridge, MA, USA. [138]The Broad Institute of MIT and Harvard, Cambridge, MA, USA. [139]Division of Cardiovascular Medicine, Vanderbilt University, Nashville, TN, USA. [140]These authors contributed equally: Ramachandran S. Vasan, J. Gustav Smith, Hilma Holm, Sonia Shah, Patrick T. Ellinor, Aroon D. Hingorani, Quinn Wells, R. Thomas Lumbers. *Lists of authors and their affiliations appear at the end of the paper. ✉e-mail: t.lumbers@ucl.ac.uk

## Genes & Health Research Team

Sarah Finer[55], Hilary Martin[92] & David van Heel[126]

Full lists of members appear in the Supplementary Information.

## Estonian Biobank Research Team

Erik Abner[19], Tõnu Esko[4,19], Maris Teder-Laving[19] & Mari-Liis Tammesoo[19]

Full lists of members appear in the Supplementary Information.

## DBDS Genomic Consortium

Søren Brunak[36], Mie Topholm Bruun[37], Joseph Dowsett[49], Christian Erikstrup[52,53], Sisse Rye Ostrowski[49,101], Ole Birger Pedersen[101,106], Kari Stefansson[63,117], Erik Sørensen[49] & Henrik Ullum[125]

Full lists of members appear in the Supplementary Information.

## HERMES Consortium

Albert Henry[1,2], Xiaodong Mo[3], Chris Finan[1], Mark D. Chaffin[4], Doug Speed[5], Hanane Issa[2], Spiros Denaxas[2,6,7,8], James S. Ware[4,9,10,11,12], Sean L. Zheng[9,11], Anders Malarstig[13,14], Jasmine Gratton[1], Isabelle Bond[1], Carolina Roselli[4,15], David Miller[16], Sandesh Chopade[1], A. Floriaan Schmidt[1,17,18], Erik Abner[19], Lance Adams[20], Charlotte Andersson[21,22], Krishna G. Aragam[4,23,24], Johan Ärnlöv[25,26], Geraldine Asselin[27], Anna Axelsson Raja[28], Joshua D. Backman[29], Traci M. Bartz[30], Kiran J. Biddinger[4,23], Mary L. Biggs[30,31], Heather L. Bloom[32], Eric Boersma[33], Jeffrey Brandimarto[34], Michael R. Brown[35], Søren Brunak[36], Mie Topholm Bruun[37], Leonard Buckbinder[14], Henning Bundgaard[28], David J. Carey[38], Daniel I. Chasman[39,40], Xing Chen[14], James P. Cook[41], Tomasz Czuba[42], Simon de Denus[27,43], Abbas Dehghan[44], Graciela E. Delgado[45], Alexander S. Doney[46], Marcus Dörr[47,48], Joseph Dowsett[49],

Samuel C. Dudley[50], Gunnar Engström[51], Christian Erikstrup[52,53], Tõnu Esko[4,19], Eric H. Farber-Eger[54], Stephan B. Felix[47,48], Sarah Finer[55], Ian Ford[56], Mohsen Ghanbari[57], Sahar Ghasemi[48,58,59], Jonas Ghouse[60], Vilmantas Giedraitis[61], Franco Giulianini[39], John S. Gottdiener[62], Stefan Gross[47,48], Daníel F. Guðbjartsson[63,64], Hongsheng Gui[65], Rebecca Gutmann[66], Sara Hägg[67], Christopher M. Haggerty[38], Åsa K. Hedman[13], Anna Helgadottir[63], Harry Hemingway[2,6], Hans Hillege[15], Craig L. Hyde[14], Bitten Aagaard Jensen[68], J. Wouter Jukema[69,70], Isabella Kardys[33], Ravi Karra[71,72], Maryam Kavousi[57], Jorge R. Kizer[73], Marcus E. Kleber[45], Lars Køber[74], Andrea Koekemoer[75], Karoline Kuchenbaecker[76,77], Yi-Pin Lai[14], David Lanfear[65,78], Claudia Langenberg[79,80,81], Honghuang Lin[22,82], Lars Lind[83], Cecilia M. Lindgren[4,84,85], Peter P. Liu[86,87,88], Barry London[89], Brandon D. Lowery[54], Jian'an Luan[81], Steven A. Lubitz[4,90], Patrik Magnusson[67], Kenneth B. Margulies[34], Nicholas A. Marston[91], Hilary Martin[92], Winfried März[45,93,94], Olle Melander[95], Ify R. Mordi[46], Michael P. Morley[34], Andrew P. Morris[41,85], Alanna C. Morrison[35], Lori Morton[96], Michael W. Nagle[14], Christopher P. Nelson[75], Alexander Niessner[97], Teemu Niiranen[98,99], Raymond Noordam[100], Christoph Nowak[25], Michelle L. O'Donoghue[91], Sisse Rye Ostrowski[49,101], Anjali T. Owens[34], Colin N. A. Palmer[102], Guillaume Paré[103,104,105], Ole Birger Pedersen[101,106], Markus Perola[99], Marie Pigeyre[105,107], Bruce M. Psaty[108,109], Kenneth M. Rice[30], Paul M. Ridker[39,40], Simon P. R. Romaine[75], Jerome I. Rotter[110,111,112], Christian T. Ruff[91], Marc S. Sabatine[91], Neneh Sallah[2], Veikko Salomaa[99], Naveed Sattar[113], Alaa A. Shalaby[114], Akshay Shekhar[96], Diane T. Smelser[38], Nicholas L. Smith[109,115,116], Erik Sørensen[49], Sundararajan Srinivasan[102], Kari Stefansson[63,117], Garðar Sveinbjörnsson[63], Per Svensson[118,119], Mari-Liis Tammesoo[19], Jean-Claude Tardif[27,120], Maris Teder-Laving[19], Alexander Teumer[48,59,121], Guðmundur Thorgeirsson[63,117,122], Unnur Thorsteinsdottir[63,117], Christian Torp-Pedersen[123], Vinicius Tragante[63], Stella Trompet[69,100], Andre G. Uitterlinden[57,124], Henrik Ullum[125], Pim van der Harst[15,17], Folkert W. Asselbergs[2,8,18], Jessica van Setten[17], Marion van Vugt[17], Abirami Veluchamy[46,102], Monique Verschuuren[127,128], Niek Verweij[15], Christoffer Rasmus Vissing[28], Uwe Völker[48,129], Adriaan A. Voors[15], Lars Wallentin[130], Yunzhang Wang[67], Peter E. Weeke[60], Kerri L. Wiggins[31], L. Keoki Williams[65], Yifan Yang[34], Bing Yu[35], Faiez Zannad[131], Chaoqun Zheng[28], Folkert W. Asselbergs[2,8,18], Thomas P. Cappola[34], Marie-Pierre Dubé[27,120], Michael E. Dunn[96], Chim C. Lang[46], Nilesh J. Samani[75], Svati Shah[71,132,133], Ramachandran S. Vasan[22,134,140], J. Gustav Smith[42,135,136,140], Hilma Holm[63,140], Sonia Shah[3,140], Patrick T. Ellinor[4,90,137,138,140], Aroon D. Hingorani[1,140], Quinn Wells[139,140] & R. Thomas Lumbers[2,6,8,140]

Full lists of members appear in the Supplementary Information.

## Methods

### Ethics statement

This study complies with the ethical regulations provided by the University College London Research Ethics Committee. All participating studies were ethically approved by local committees, and all study participants provided written informed consent (Supplementary Table 21).

### Phenotype definition

The present meta-analysis included 1,946,349 individuals from 42 studies (Supplementary Table 21). Each study defined up to four HF phenotypes, which are as follows: (1) overall HF (HF$_{all}$), (2) ni-HF, (3) ni-HFrEF and (4) ni-HFpEF. The HF$_{all}$ phenotype includes any diagnosis of HF based on a physician's adjudication, hospital record review or diagnosis codes. ni-HF was defined by excluding antecedent ischemic, valvular and congenital heart diseases. ni-HFrEF was defined as ni-HF with LVEF < 50% based on cardiac imaging or diagnosis of LVSD at any point. ni-HFpEF was defined as ni-HF with LVEF ≥ 50% based on cardiac imaging without a record of LVSD at any point. Phenotyping was performed separately in each participating study (Supplementary Table 22), using a harmonized multimodal phenotyping framework (Supplementary Methods).

### Genome-wide association analysis

Study-level GWAS was performed using logistic regression (for prevalent cases) or Cox proportional hazard (for incident cases) assuming additive genetic effect with adjustment for sex, age at DNA draw, genetic PCs and study-specific covariates (Supplementary Table 23). Study-level GWAS results were meta-analyzed using a fixed-effect IVW model implemented in METAL[61]. As study-level estimates include both results from logistic regression and Cox proportional hazard regression, we report the meta-analysis effect estimates as relative risk per additional risk allele. Sample genotyping and quality control of study-level and meta-analysis GWAS summary statistics are detailed in Supplementary Methods and Supplementary Data 2.

### Identification of genetic susceptibility loci

To identify genetic susceptibility loci for HF, we performed a chromosome-wide stepwise conditional-joint analysis in each heart failure phenotype using Genome-wide Complex Trait Analysis (GCTA) software[62]. Conditionally independent variants across HF phenotype with joint $P < 5 \times 10^{-8}$ that are physically located within 500 kb of each other were aggregated into one locus set. A genomic locus was then defined as the genomic region within 500 kb upstream and downstream of the farthest variants in each aggregated set.

The identified genomic loci were labeled with an incremental one-based integer sequence based on phenotype order (HF$_{all}$, ni-HF, ni-HFrEF and ni-HFpEF), chromosome and base pair positions. A genomic locus was declared as new if all conditionally independent variants within the locus and any of the sentinel variants reported at $P < 5 \times 10^{-8}$ in previous GWAS of HF[1–3,6,7] were physically located more than 250 kb away and not in linkage disequilibrium (LD; $r^2 < 0.2$). Analysis was performed using reference genotype from randomly sampled 10,000 UKB participants with proportionally matched ancestry composition (admixed American ancestry was not included due to unavailability in UKB and the small proportion of this ancestry in the overall sample).

Furthermore, genetic associations were categorized based on P values at cutoff values of $P < 5 \times 10^{-8}$ (genome-wide significant), $P < 0.05$/number of identified loci (replicated at a Bonferroni-adjusted threshold), $P < 0.05$ (nominally significant) and $P \geq 0.05$ (no evidence of association). Loci associated with any ni-HF subtype at Bonferroni-adjusted or genome-wide significance thresholds were labeled as ni-HF loci, and the remaining were labeled as other HF loci. ni-HF loci that are associated with ni-HFrEF at Bonferroni-adjusted or genome-wide significance thresholds were labeled as ni-HFrEF loci, and the remaining

were labeled as ni-HFpEF loci. Sentinel variants for HF were defined as conditionally independent variants with the lowest P value for association across HF phenotypes within a locus. For loci associated with both HF$_{all}$ and ni-HF phenotypes, sentinel variants identified in the HF$_{all}$ analysis were prioritized.

### Cross-ancestry allelic effect heterogeneity

To assess the heterogeneity of allelic effects across ancestries, we performed a meta-analysis accounting for ancestry-specific allelic effects with a meta-regression framework implemented in MR-MEGA[63]. Using this technique, we estimated the P value for heterogeneity correlated with ancestry, the P value for residual heterogeneity and the concordance of genetic association P values from the fixed-effect meta-analysis across sentinel variants for the HF$_{all}$ phenotype (Supplementary Fig. 3).

### Genetic effect modification by sex

We estimated potential genetic effect modification by sex across sentinel genetic variants for HF through a meta-regression analysis using the R meta v7.0.0 package[64,65]. For each sentinel variant, we extracted study-specific effect estimates across studies with available sex ratio information (Supplementary Table 1) and aligned the effect alleles to risk-increasing alleles in the GWAS meta-analysis. We then performed a meta-regression of study-level effect estimates on the proportion of males, weighted by the inverse variance of the effect size. We report the regression coefficient and the corresponding P value, whereby a positive value represents a positive correlation between the estimated additive genetic effect of risk-increasing alleles and a larger proportion of males.

### Genetic architecture assessment

To assess genetic architecture and polygenicity across HF phenotypes, we compared quantiles of the expected and observed genome-wide genetic association P values using quantile–quantile plot and calculated the genomic inflation coefficient ($\lambda_{GC}$). To distinguish between polygenicity and confounding by population stratification, we calculated $\lambda_{GC}$ assuming 1,000 participants ($\lambda_{GC-1,000}$), and estimated the LD score regression slope using LDSC[66]. To minimize bias, LDSC regression was performed using the European ancestry meta-analysis subset with reference genotypes from 10,000 random UKB European participants. We investigated the relationship between allelic effect on HF and allele frequency by plotting the relative risk per additional minor allele as a function of MAF and fitted two separate local polynomial regressions with locally estimated scatter-plot smoothing for groups of conditionally independent variants associated with increased and decreased risk of HF. To increase precision, we fitted the regression using conditionally independent variants associated with HF$_{all}$ at an FDR < 1%, estimated using the $q$ value package in R[67].

### SNP-based heritability estimation

The proportion of variance in HF risk explained by common SNPs, that is, SNP-based heritability ($h^2_{SNP}$), was estimated from GWAS meta-analysis summary statistics using Linkage-Disequilibrium-Adjusted Kinships (LDAK) SumHer software[68] with LDAK-Thin and BLD-LDAK heritability models[69] on a liability scale. To minimize bias, we used the European ancestry meta-analysis subset with precomputed tagging files derived from 2,000 UKB European participants[68]. The conversion to liability scale was calculated using population prevalence derived from the meta-analysis sample and sample prevalence derived using correction for effective sample size assuming an equal number of cases and controls[70].

### Polygenic risk score analysis

We explored the association between PGS of HF$_{all}$ (PGS$_{HF}$) and HF$_{all}$ risk in 347,235 UKB European participants, including 13,793 HF$_{all}$ cases. The PGS$_{HF}$ was constructed as a weighted sum of the allelic count of

1,012,059 genetic variants selected using the LDpred2-auto model[71], with weights derived from the present GWAS meta-analysis of HF excluding UKB and reference genotype from 362,320 UKB European participants provided by LDpred2 authors. OR of $HF_{all}$ per s.d. of $PGS_{HF}$ was estimated using logistic regression with binary $HF_{all}$ status as response variable, standardized $PGS_{HF}$ as predictor and sex, age and first ten genetic PCs as covariates. To estimate the risks of HF in individuals with high $PGS_{HF}$, we grouped participants into deciles of $PGS_{HF}$ and calculated ORs of the top decile as compared against the fifth and the first (bottom) decile using logistic regression.

### Fine mapping and functional consequences of causal variants

We performed functionally informed fine mapping using PolyFun[9] and sum of single effects (SuSiE)[72] to identify causal variants within HF genetic susceptibility loci. We used precomputed, functionally informed prior causal probabilities of 19 million genetic variants based on a meta-analysis of 15 UKB traits[9] and genome-wide association estimates from the current analysis to calculate prior causal probability proportional to per-SNP heritability of phenotype under analysis. The resulting estimates were used to calculate per-SNP PIP and to construct 95% credible sets of likely causal variants using the sum-of-single-effects fine-mapping model implemented in SuSiE, assuming at most five causal variants per locus. To minimize bias, we used effect estimates from the European ancestry meta-analysis subset and reference genotypes from 10,000 random UKB European participants. To assess the functional consequences of fine-mapped variants, we extracted variant-level information on the nearest gene(s), genic functions and CADD[10] Phred score from ANNOVAR[73] and Open-Targets Genetics[14].

### Prioritization of effector genes

To identify effector genes for HF, we implemented a two-step prioritization approach. In step 1, we identified candidate genes using a combination of the following three predictors: (1) PoPS[15], (2) V2G−OpenTargets V2G score[14,74] and (3) TWAS−multitissue TWAS using S-MulTiXcan[16]. Genes with the highest PoPS, highest V2G score or lowest TWAS *P* value within a locus were considered as candidate effector genes for HF.

In step 2, we further prioritized these genes using the following three Boolean (true/false) classifiers: (1) ABC−overlap with enhancer as predicted by ABC[17,18] score > 0.02, (2) Mendel−association with enriched Mendelian disease term estimated using MendelVar[20] and (3) Coloc−sharing of causal variants (colocalization) between gene expression in relevant tissue with at least one HF phenotype under study at posterior probability >0.8 estimated using R coloc package[19] using gene expression data from Genotype-Tissue Expression project (GTEx, v.8)[75].

In addition, we derived an overall predictor score based on weighted average of PoPS, highest V2G and $-\log_{10}(P)$ MulTiXcan values with 2:2:1 weight ratio, scaled to 0−100 value using a quantile transformation with uniform output distribution as implemented in the Python scikit-learn[76] library. Finally, for each locus, we ranked genes based on total classifier score (sum of true values) and the overall predictor score. Genes that are top-ranked or have a total classifier score ≥2 were prioritized as effector genes for HF. More details are provided in Supplementary Methods.

### Tissue- and cell-type enrichment

To identify tissues and cell types involved in HF pathology, we estimated heritability enrichment of specifically expressed gene sets across 206 tissues and cell types[77,78] within 12 organ/system categories (Supplementary Table 24). For each set, we performed a one-sided enrichment test for per-SNP heritability attributed to the given set, conditional on the set that includes all genes and a baseline model consisting of 52 genomic annotations using LDSC applied to specifically expressed genes (LDSC-SEG)[79]. For this analysis, we used the European ancestry

meta-analysis subset with precomputed LD score weights derived from the 1000 Genomes European reference panel[80].

Based on the observation that heart tissues are the major contributors to SNP heritability across HF phenotypes, we extended this analysis to 15 cell types derived from single-nucleus RNA-sequencing of 185,185 nuclei from 16 nonfailing human heart donors[43] (Supplementary Table 25). Identification of specifically expressed genes across cardiac cell types is described in Supplementary Methods.

As sensitivity analysis, we further performed a gene-property enrichment analysis using MAGMA[81,82]. We first assigned SNPs to 20,260 coding genes from Ensembl v.92 with a 1-kb window and computed gene-level association statistics using the SNP-wise mean model. We then computed average gene expression per cell type and provided this as gene covariates to MAGMA to calculate a one-sided *P* value for enrichment of a given cell type conditioned on the average expression of other cell types.

### Cellular expression of heart failure effector genes

To assess changes in the expression of HF effector genes in failing heart, we performed single-nucleus differential gene expression analysis of candidate effector genes within heart failure loci by comparing transcription levels in cell nuclei from nonfailing heart described above with cell nuclei from 28 failing heart donors diagnosed with end-stage DCM ($n_{donors} = 12$ and $n_{nuclei} = 142{,}490$) or HCM ($n_{donors} = 16$ and $n_{nuclei} = 202{,}307$) from Myocardial Applied Genetic Network[34,83]. Samples are processed using CellBender[84] and Cell Ranger[85]. Differential gene expression was calculated using the limma−voom[86,87] model adjusting for age and sex (see more details in Supplementary Methods).

Differential expression was defined using the following criteria: two-sided adjusted *P* < 0.01 (following Benjamini−Hochberg adjustment for multiple testing), sign concordance between estimates derived from CellBender and Cell Ranger quantifications and no background contamination in CellBender. To test whether differentially expressed genes in each cell type are overrepresented among prioritized HF effector genes, we performed a one-sided Fisher's exact test implemented in R, using all genes tested in the given cell type as the background set.

### Pathway enrichment

To identify biological pathways relevant to HF, we tested for overrepresentation of biological terms in HF effector gene set using g:Profiler[88]. To account for etiological differences and uncertainty in gene prioritization, we tested six gene sets constructed from combinations of three phenotypic classifications of loci (HF, ni-HF and other HF) and two gene prioritization categories (candidate and prioritized effector genes). For each gene set, we performed an unordered enrichment analysis with a one-sided Fisher's test for biological terms in the Kyoto Encyclopedia of Genes and Genomes[89], Reactome[90], Wiki Pathways[91] and Gene Ontology (GO)[92,93], excluding terms with more than 2,000 genes. Multiple-testing adjustment was performed by multiplying the enrichment *P* values by the ratio of the approximate threshold given the number of genes in the set and the initial experiment-wide type-I error rate of 0.05, using the g:SCS algorithm in g:Profiler. Driver GO terms were identified using the g:Profiler clustering algorithm to represent groups of more specific terms.

### Phenome-wide association of heart failure sentinel variants

To characterize pleiotropic effects of HF genetic loci across human diseases, we performed a phenome-wide association study (PheWAS) in 408,480 UKB European participants. We tested the associations between sentinel variants at 66 HF loci with 294 disease phenotypes derived using a curated phenotype definition and category[39]. For each phenotype, participants were classified as cases if at least one event was recorded in linked hospital admission, death certificate, primary care visit, self-reported cancer diagnosis, noncancer diagnosis or procedure

history. For each phenotype–sentinel variant pair with at least 100 cases and 100 noncases (controls), we ran a case–control additive genetic association test in PLINK2 with a logistic regression model adjusted for sex, genotyping array and first ten genetic PCs. For presentation, we report effects per additional HF risk-increasing allele.

## Pleiotropy network analysis

Using the PheWAS results, we performed a network analysis to evaluate the connections between HF genetic loci based on their pleiotropic effects across human diseases. First, we constructed a network dataset with nodes representing loci or phenotypes and edges representing unidirectional associations of the locus sentinel variant with a phenotype. Eigenvector centrality measure was calculated to estimate node influence within the network (Supplementary Fig. 13 and Supplementary Table 26). Furthermore, we performed a community detection analysis using a walktrap algorithm[94] weighted by the absolute $z$ score of the edges to identify distinct etiologic clusters underlying HF. The network analysis and visualization were performed using igraph, tidygraph and ggraph package in R.

## Locus-specific pleiotropy assessment

We investigated the local pleiotropic effects of sentinel variants in 66 HF genetic loci on 24 related GWAS traits (Supplementary Table 27). Estimates for sentinel variants that were not reported in a target GWAS were substituted with those of a proxy variant with the highest LD $r^2$ (requiring at least LD $r^2 > 0.8$), identified using PLINK[95] with LD reference panel derived from 10,000 random samples of UKB European participants. For presentation (Extended Data Fig. 10), effect estimates were converted to $z$ scores derived from the regression coefficients ($\beta$) divided by their s.e., with direction of effect aligned to reflect the additive effect of HF risk-increasing allele. Loci and GWAS traits were ordered using hierarchical agglomerative clustering (implemented with the hclust function in R 4.2.0.) with average linkage method based on the Euclidean distance measure derived from absolute $z$ scores.

## Cross-trait colocalization

To test whether HFe and related traits share causal genetic variants, we performed pairwise cross-trait colocalization analysis using the R coloc package[19]. Analyses were performed across pairwise combinations of credible sets corresponding to the tested traits identified using SuSiE regression[72]. We used the default marginal prior probabilities for association with trait 1 ($p_1$) and trait 2 ($p_2$) of $10^{-4}$, as well as a prior probability of joint causal association ($p_{12}$) of $10^{-5}$. A posterior probability of common causal variants ($PP_{coloc\,H4}$) > 0.8 in at least one pair of credible sets was considered as evidence of colocalization. The colocalization analysis was performed for each identified HF locus using an overlapping set of variants available in both GWAS of heart failure and trait under analysis situated within the locus. Genetic association estimates for HF were extracted from GWAS of the $HF_{all}$ phenotype for loci with $P_{HF} < 5 \times 10^{-8}$ (locus 1–56) or otherwise from GWAS of ni-HF phenotypes for subtype-specific loci—ni-HF for locus 57–61, ni-HFrEF for locus 62–64 and ni-HFpEF for locus 65–66. Genetic association estimates for tested traits are extracted from the source GWAS (Supplementary Table 27).

## Genetic correlation and MR

We tested genetic correlation across pairs of four HF phenotypes under analysis and 24 related GWAS traits using bivariate LDSC regression[96]. We further estimated the causal effects of each tested trait on four HF phenotypes under analysis using two-sample MR as implemented in the MendelianRandomization package[97]. MR instruments for the exposure traits were selected from overlapping genetic variants available in both GWAS of exposure and outcome traits using an LD-based clumping algorithm with a $P$ value threshold of $5 \times 10^{-8}$ and LD $r^2$ threshold of 0.05 implemented in PLINK[95]. For each exposure trait, we estimated the

causal association with each HF phenotype using the IVW MR estimator and performed sensitivity analyses with MR–Egger and WME[98,99]. An adjusted $P < 0.01$ following the Benjamini–Hochberg correction in IVW analysis with the consistent direction of effect in sensitivity analyses was considered supportive evidence of causality. Reference genotype from 10,000 random UKB European participants was used to perform these analyses, given the majority of European ancestry in the tested GWAS traits. Details of the tested traits, source GWAS and model parameters are provided in Supplementary Table 27.

## Reporting summary

Further information on research design is available in the Nature Portfolio Reporting Summary linked to this article.

## Data availability

GWAS summary statistics are available to download from the Cardiovascular Disease Knowledge Portal for the multi-ancestry meta-analysis (https://api.kpndataregistry.org/api/d/6Ls5Wu) and for the European ancestry subset meta-analysis (https://api.kpndataregistry.org/api/d/6eJqWn). Study-level summary statistics can be requested by contacting individual studies, as described in Supplementary Table 21. Single-cell data for cell-type enrichment and differential gene expression analyses are available at https://singlecell.broadinstitute.org/single_cell/study/SCP1303/. Data from UKB can be requested from the UKB Access Management System. GTEx data can be accessed at https://gtexportal.org. A summary of regional genetic associations, gene prioritization scores, cross-trait association and study-level estimates across identified GWAS loci is provided in Supplementary Data 1 and online at https://hermes2-supp-note.netlify.app/locus_desc.html. A summary of study-level GWAS summary statistics quality control is provided in Supplementary Data 2 and online at https://hermes2-supp-note.netlify.app/hf_subtypes_qc.html.

## Code availability

GWAS meta-analysis was performed centrally using METAL (v2020-05-05). SNP-based heritability was calculated using LDAK SumHer software (v5.2). Genetic correlation between traits was performed using LDSC (v1.0.1). Conditionally independent variants were identified using GCTA software (v1.92.4). Causal variant fine mapping was performed using PolyFun (v2020-11-14) and SuSiE (v0.11.92). Polygenic prediction score (PoPS, v0.1), OpenTargets Variant2Gene (v1.1), S-MultiXcan (v0.7.3), MendelVar (v2023-12-05), coloc R package (v5.2.3), ABC-Max (v2021-04-08) and Python scikit-learn (v1.5.1) were used for effector gene prioritization. Pathway enrichment of prioritized genes was performed using g:Profiler (v0.2.3) package in R. Tissue-based and cell-type enrichment was performed using S-LDSC (v1.0.1) and MAGMA (v1.10). PGSs were derived using the LDpred2-auto model implemented in bigsnpr R package (v1.12.16), with individual PGS scores generated using PLINK (v1.9). PheWAS of sentinel variants was performed using PLINK (v2.0). Network analysis was performed using tidygraph (v1.2.3) and visualized using ggraph (v2.0.0) in R. Custom code to perform the main analyses is available at Zenodo[100] (https://doi.org/10.5281/zenodo.11204854).

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

## Acknowledgements

A. Henry was supported by the British Heart Foundation Cardiovascular Biomedicine PhD studentship (FS/18/65/34186). R.T.L. and A. Henry were partly supported by a Pfizer Innovative Targets Exploration Network Grant. The project was additionally supported by the BigData@Heart Consortium, funded by the Innovative Medicines Initiative-2 Joint Undertaking (grant agreement 116074), the UCL British Heart Foundation Accelerator (AA/18/6/34223), the National Institute for Health Research University College London Hospitals Biomedical Research Centre (NIHR203328) and Health Data Research UK (MR/S003754/1). Analyses using the UKB resource presented in this work were conducted under applications 9922, 15422, 12113, 12505 and 47602. Additional study-level acknowledgements are provided in Supplementary Table 21. We thank all research participants included in this study. The views expressed in this work are those of the authors and not necessarily those of the funders.

## Author contributions

R.S.V., J.G.S., H. Holm, Sonia Shah, P.T.E., A.D.H., Q.W. and R.T.L. conceptualized the study. A. Henry, M.D.C., D.S. and Sonia Shah developed the methodology. A. Henry, H.I., D.M. and R.T.L. managed

project administration. A. Henry and X.M. carried out formal analysis. A. Henry, C.F., D.S., S.C. and A.F.S. conducted software analysis. A. Henry and H.I. curated the data. A. Henry and R.T.L. wrote the original draft of the manuscript. A. Henry handled visualization. C.F., M.D.C., S.D., J. Gratton, S.C., J.G.S., H. Holm, P.T.E. and A.D.H. arranged the resources. M.D.C., I.B., C.R. and D.M. conducted the investigation. F.W.A., T.P.C., M.-P.D., M.E.D., C.C.L., N.J.S., R.S.V., J.G.S., H. Holm, Sonia Shah, P.T.E., A.D.H., Q.W. and R.T.L. provided supervision. R.T.L. secured funding. F.W.A., T.P.C., M.-P.D., M.E.D., C.C.L., N.J.S., Svati Shah, R.S.V., J.G.S., H. Holm, P.T.E., A.D.H., Q.W. and R.T.L. are members of the HERMES Executive Committee who provided additional supervision of the work. J.S.W., S.L.Z., A.M., E.A., L.A., C.A., K.G.A., J.Ä., G.A., A.A.R., J.D.B., T.M.B., K.J.B., M.L.B., H.L.B., E.B., J.B., M.R.B., S.B., M.T.B., L.B., H.B., D.J.C., D.I.C., X.C., J.P.C., T.C., S.d.d., A.D., G.E.D., A.S.D., M.D., J.D., S.C.D., G.E., C.E., T.E., E.H.F.-E., S.B.F, S.F., I.F., M.G., S. Ghasemi, J. Ghouse, V.G., F.G., J.S.G., S. Gross, D.F.G., H.G., R.G., S.H., C.M.H., Ä.K.H., A. Helgadottir, H. Hemingway, H. Hillege, C.L.H., B.A.J., J.W.J., I.K., R.K., M.K., J.R.K., M.E.K., L.K., A.K., K.K., Y.-P.L., D.L., C.L., H.L., L.L., C.M.L., P.P.L., B.L., B.D.L., J.L., S.A.L., P.M., K.B.M., N.A.M., H.M., W.M., O.M., I.R.M., M.P.M., A.P.M., A.C.M., L.M., M.W.N., C.P.N., A.N., T.N., R.N., C.N., M.L.O., S.R.O., A.T.O, C.N.A.P., G.P., O.B.P., M. Perola, M. Pigeyre, B.M.P., K.M.R., P.M.R., S.P.R.R., J.I.R., C.T.R., M.S.S., N. Sallah, V.S., N. Sattar, A.A.S., A.S., D.T.S., N.L.S., E.S., S. Srinivasan, K.S., G.S., P.S., M.-L.T., J.-C.T., M.T.-L., A.T., G.T., U.T., C.T.-P., V.T., S.T., A.G.U., H.U., P.v.d.H., D.v.H., J.v.S., M.v.V., A.V., M.V., N.V., C.R.V., U.V., A.A.V., L.W., Y.W., P.E.W., K.L.W., L.K.W., Y.Y., B.Y., F.Z. and C.Z. contributed to data generation, funding acquisition, formal analysis and supervision at individual study level. All authors reviewed and approved the final version of the manuscript.

## Competing interests

R.T.L. received institutional research support from Pfizer. R.T.L. has served as a paid consultant for Health Lumen and FITFILE. J.S.W. has acted as a consultant for MyoKardia, Pfizer, Foresite Labs and Health Lumen and received institutional support from Bristol Myers Squibb and Pfizer. S.d.D. was supported through grants from AstraZeneca and Roche Molecular Science/DalCor. J.R.K. declares stock ownership in AbbVie, Abbott, Bristol Myers Squibb, Johnson & Johnson, Medtronic, Merck and Pfizer. N.A.M. received speaking honoraria from Amgen and is involved in clinical trials with Ionis, Amgen, Pfizer and Novartis. B.M.P. serves on the Steering Committee of the Yale Open Data Access Project funded by Johnson & Johnson. C.T.R. received honoraria for scientific advisory boards and consulting from Anthos, Bayer, Bristol Myers Squibb, Daiichi Sankyo, Janssen and Pfizer and received institutional research grants from Anthos, AstraZeneca, Daiichi Sankyo, Janssen and Novartis. M.S.S. received substantial research grant support from Abbott Laboratories, Amgen, AstraZeneca, Bayer, Critical Diagnostics, Daiichi Sankyo, Eisai, Genzyme, Gilead, GlaxoSmithKline, Intarcia, Janssen Research and Development, The Medicines Company, MedImmune, Merck, Novartis, Poxel, Pfizer, Quark Pharmaceuticals, Roche Diagnostics and Takeda and has received consulting fees from Alnylam, AstraZeneca, Bristol Myers Squibb, CVS and Amgen. A.A.V. received consultancy fees and/or research support from AnaCardia, AstraZeneca, Bayer, BMS, Boehringer Ingelheim, Corteria, Cytokinetics, Eli Lilly, Moderna, Novartis, Novo Nordisk and Roche Diagnostics. M.-P.D declares holding equity in Dalcor Pharmaceuticals, unrelated to this work. Members of the TIMI Study Group (ENGAGE, FOURIER, PEGASUS, SAVOR and SOLID) have received institutional research grant support through Brigham and Women's Hospital from Abbott, Amgen, Anthos Therapeutics, ARCA Biopharma, AstraZeneca, Bayer HealthCare Pharmaceuticals, Daiichi Sankyo, Eisai, Intarcia, Ionis Pharmaceuticals, Janssen Research and Development, MedImmune, Merck, Novartis, Pfizer, Quark Pharmaceuticals, Regeneron Pharmaceuticals, Roche, Siemens Healthcare Diagnostics, Softcell Medical Limited, The Medicines Company, Zora Biosciences, Caremark, Dyrnamix, Esperon, IFM Pharmaceuticals and MyoKardia. The authors who are affiliated with deCODE genetics/Amgen and the authors affiliated with Pfizer declare competing financial interests as employees. The remaining authors declare no competing interests.

## Additional information

**Extended data** is available for this paper at https://doi.org/10.1038/s41588-024-02064-3.

**Correspondence and requests for materials** should be addressed to R. Thomas Lumbers.

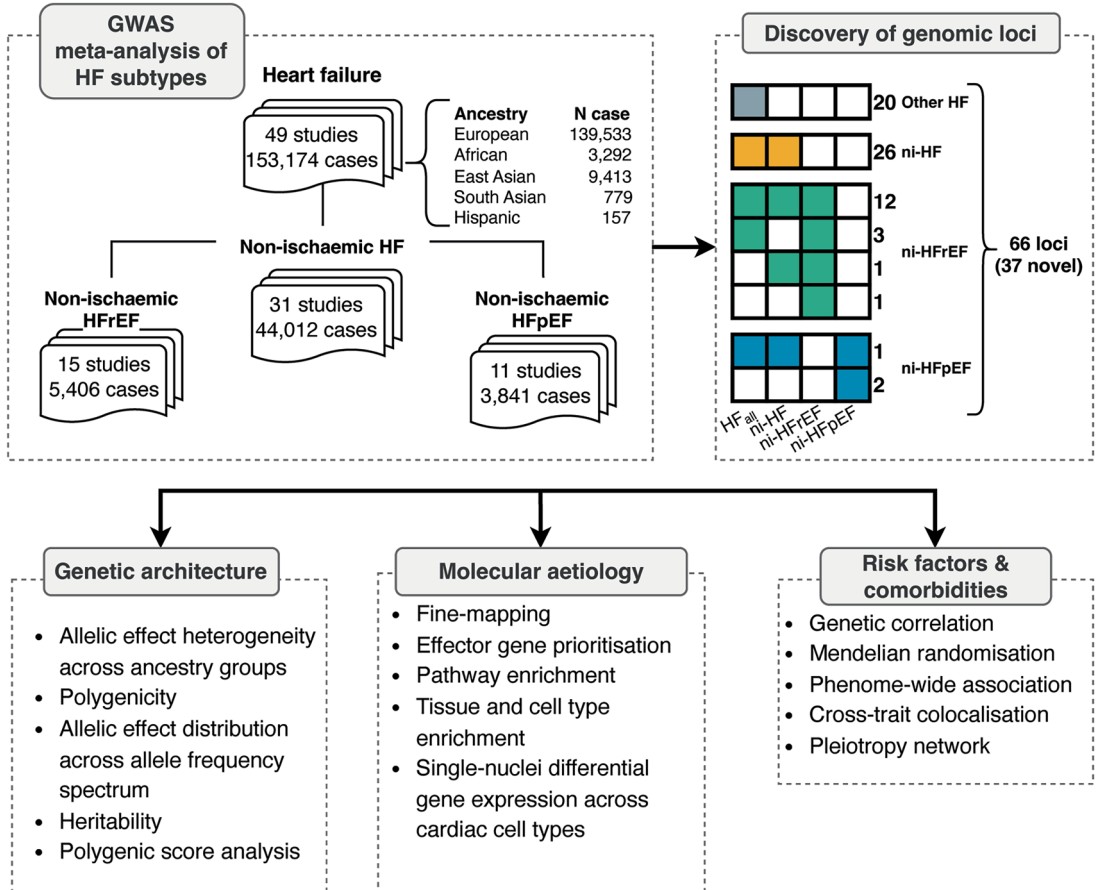

**Extended Data Fig. 1 | Study summary.** Each panel describes key analyses that are undertaken in the study. The discovery of genomic loci panel describe four categories of genomic loci identified in this study based on patterns of associations across the four phenotypes, where nonwhite colour in the heat map represents significant genetic association at a multiple-testing-corrected threshold of two-sided $P < 0.05 / 66$. GWAS, genome-wide association study; HF, heart failure; HF$_{all}$, overall heart failure; ni-HF, non-ischemic heart failure; ni-HFrEF, non-ischemic heart failure with reduced ejection fraction; ni-HFpEF, non-ischemic heart failure with preserved ejection fraction.

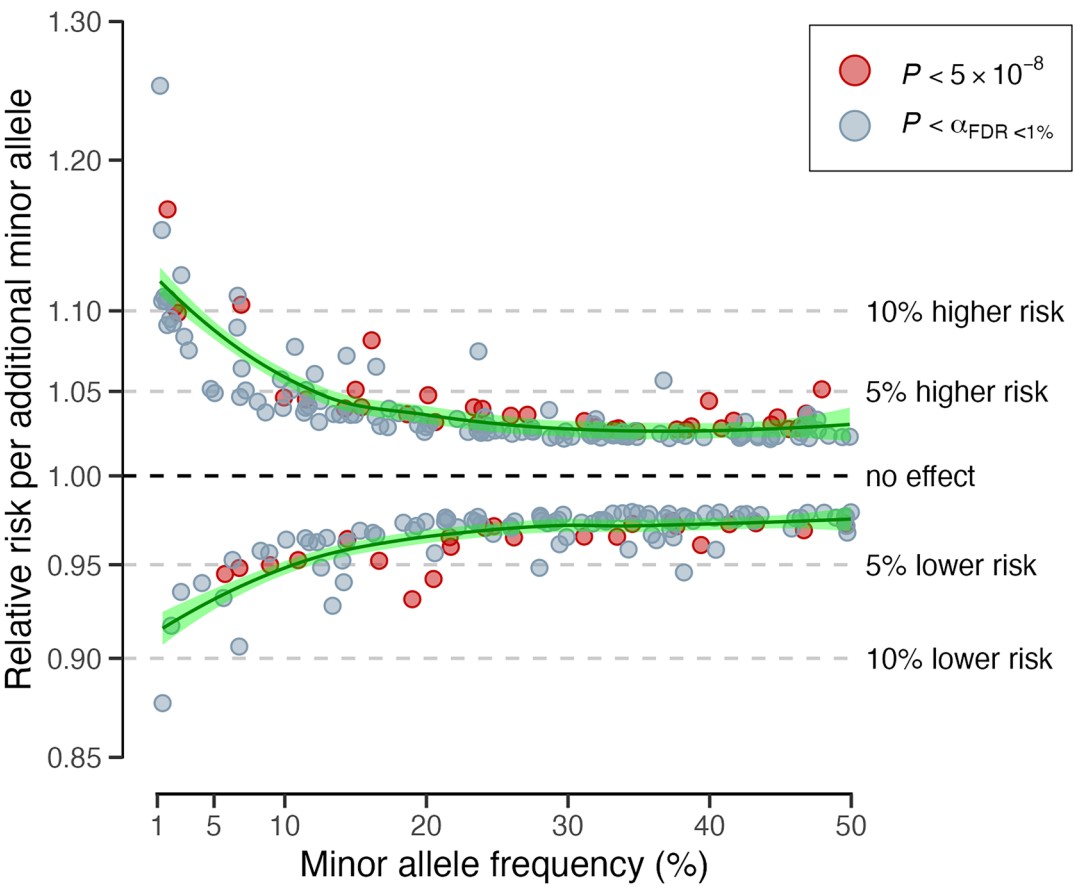

**Extended Data Fig. 2 | Allelic architecture of genetic variants associated with the overall heart failure phenotype.** The presented data points are conditionally independent variants that passed two-sided $P < 5 \times 10^{-8}$ (red colored points) and two-sided $P$ less than equivalent type-I error rate ($\alpha$) at a false discovery rate <1% as estimated using $q$ value package in R (gray colored points). Green lines represent local polynomial regression with locally estimated scatterplot smoothing (LOESS), fitted separately for variants with risk ratio >1 and risk ratio <1 per additional minor allele. Green error bands represent the corresponding 95% confidence intervals of the regression lines.

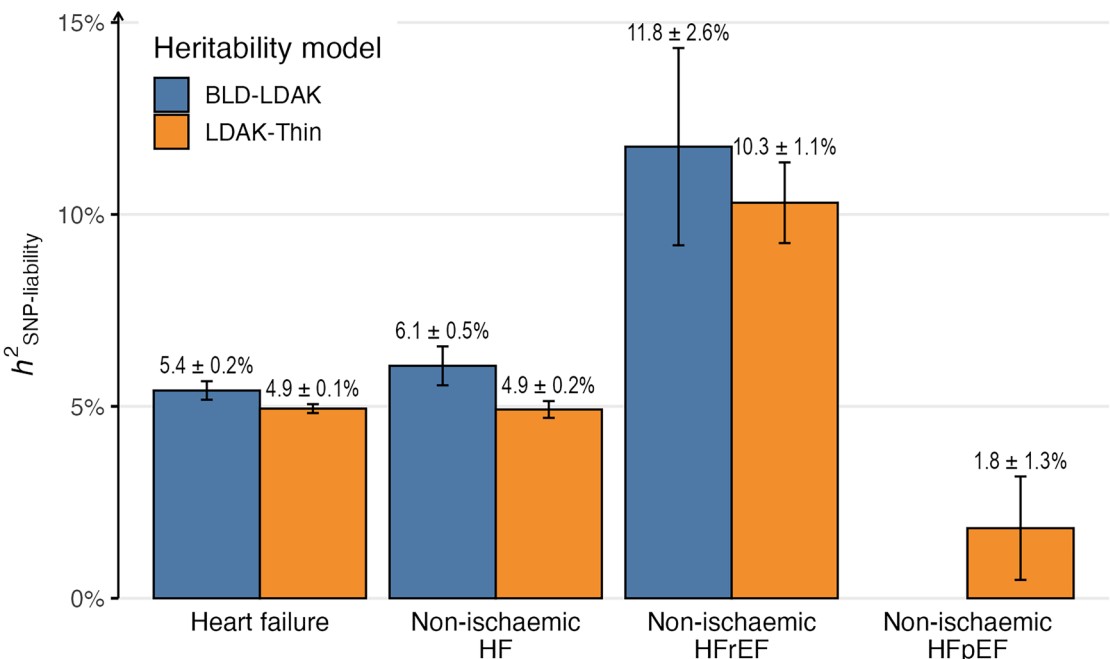

**Extended Data Fig. 3 | SNP heritability across heart failure phenotypes.** The presented estimates represent SNP heritability on a liability scale ($h^2_{\text{SNP-liability}}$) estimated using 66-parameter BLD-LDAK and LDAK-thin heritability models using GWAS meta-analysis of European ancestry cohorts. The BLD-LDAK heritability estimate for non-ischemic HFpEF was unavailable due to limited sample size. Point estimates and confidence intervals for $h^2_{\text{SNP-liability}}$ are represented by the bar height and error bars, with numerical estimates in percentage displayed on the top of the bars.

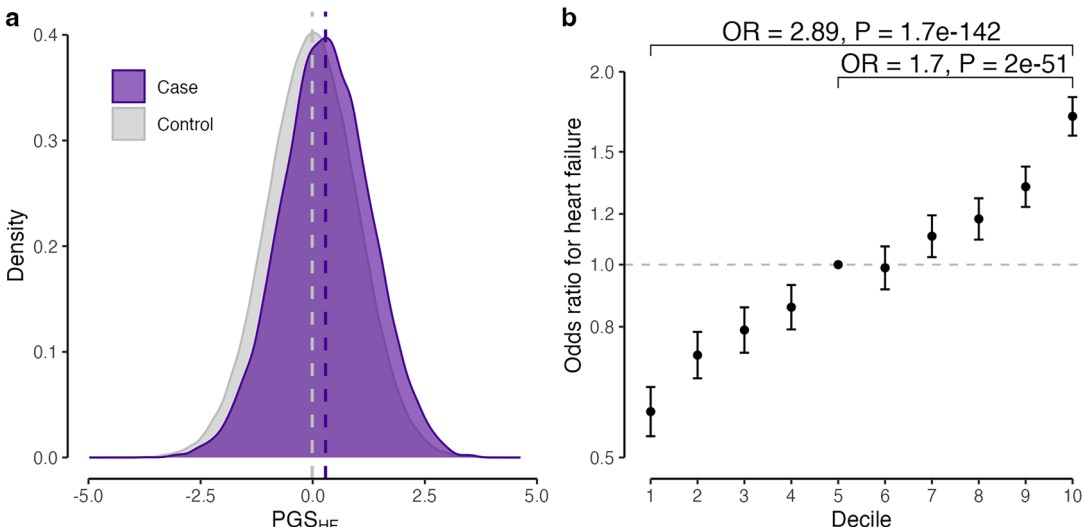

**Extended Data Fig. 4 | Association between polygenic score and risk of heart failure in UK Biobank. a**, Distribution of polygenic score for the overall heart failure phenotype (PGS$_{HF}$) among 13,824 individuals with at least one documented event in linked electronic health record (case) and 332,843 individuals without any documented event (control) in UK Biobank.

**b**, Odds ratios (OR) and 95% confidence intervals (represented as bullet points and error bars) for overall heart failure across deciles of PGS$_{HF}$ in UK Biobank with the fifth decile group as reference. Sample size and the numerical estimate for each decile are presented in Supplementary Table 8.

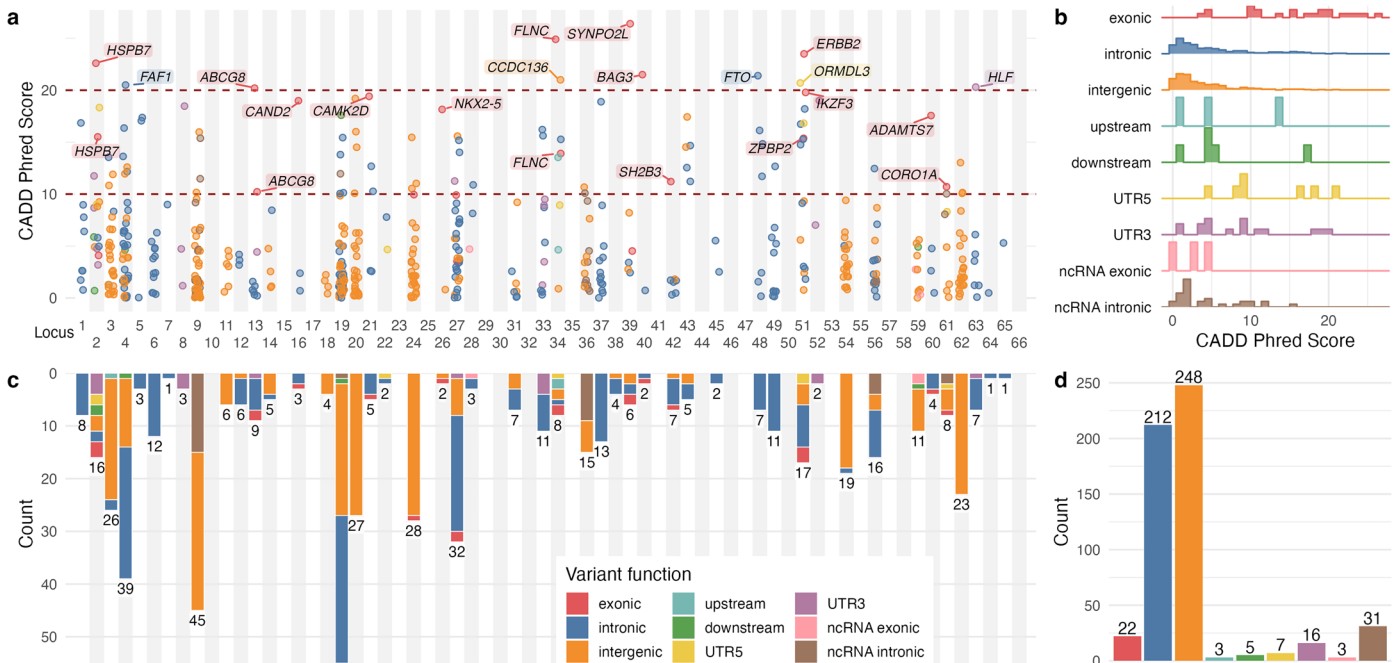

**Extended Data Fig. 5 | Functional consequences of fine-mapped genetic variants across genetic susceptibility loci for heart failure. a,** Predicted deleteriousness of 547 fine-mapped variants within 95% credible sets across 66 independent genomic loci for heart failure as measured by Combined Annotation-Dependent Depletion (CADD) Phred score. Nearest genes of fine-mapped variants with CADD Phred score >20 and fine-mapped exonic variants with CADD Phred score >10 are labeled. **b,** Histogram of CADD Phred score distribution among fine-mapped variants stratified by function. **c,d,** Locus-wide (**c**) and genome-wide (**d**) counts of fine-mapped variants stratified by variant function.

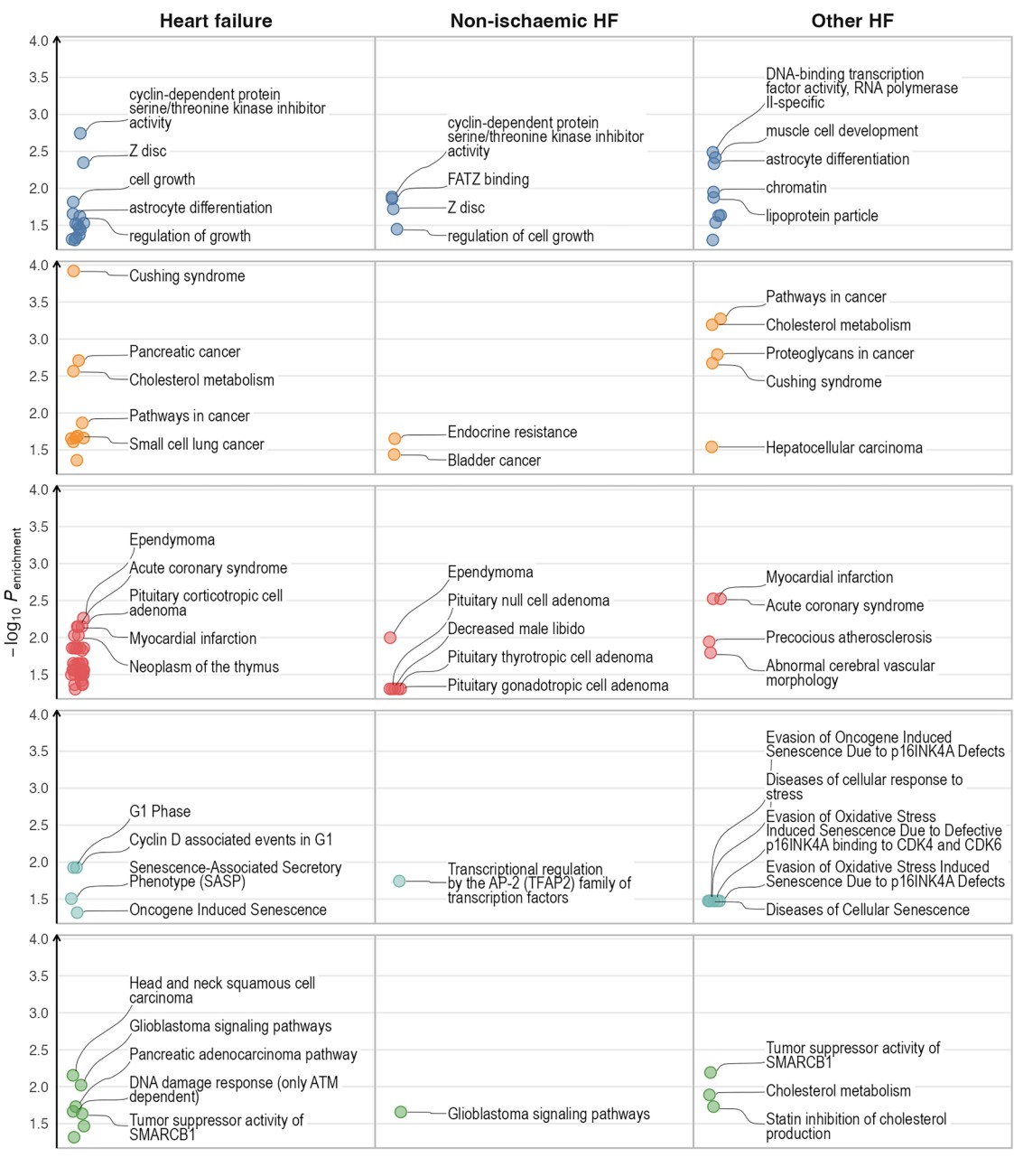

**Extended Data Fig. 6 | Enriched pathways within the effector heart failure gene set.** The presented terms are derived from Gene Ontology (GO), Kyoto Encyclopedia of Genes and Genomes (KEGG), Human Phenotype Ontology (HP), Reactome (REAC) and Wiki Pathways (WP) at a $P_{enrichment} < 0.05$. Up to top 5 enriched terms per source in each heart failure phenotype gene set are labeled. $P_{enrichment}$ is calculated from enrichment analysis of candidate and prioritized gene sets with one-sided Fisher's test, adjusted for multiple testing by multiplying the original $P$ values to the ratio of the approximate threshold given the number of genes in the set and the initial experiment-wide type-I error rate of 0.05 as described in Methods. For display, $P$ values are presented in $-\log_{10}$ scale, and terms that are enriched in both candidate and prioritized gene sets are collapsed by presenting the median values.

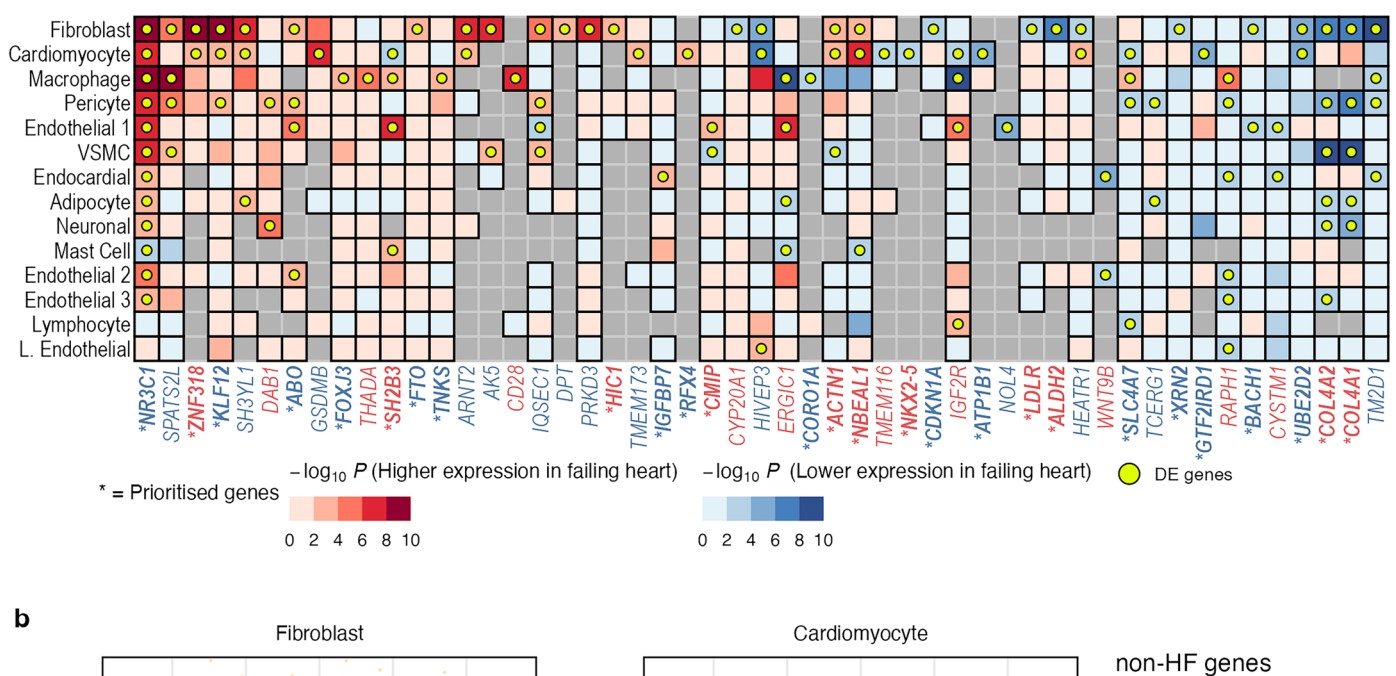

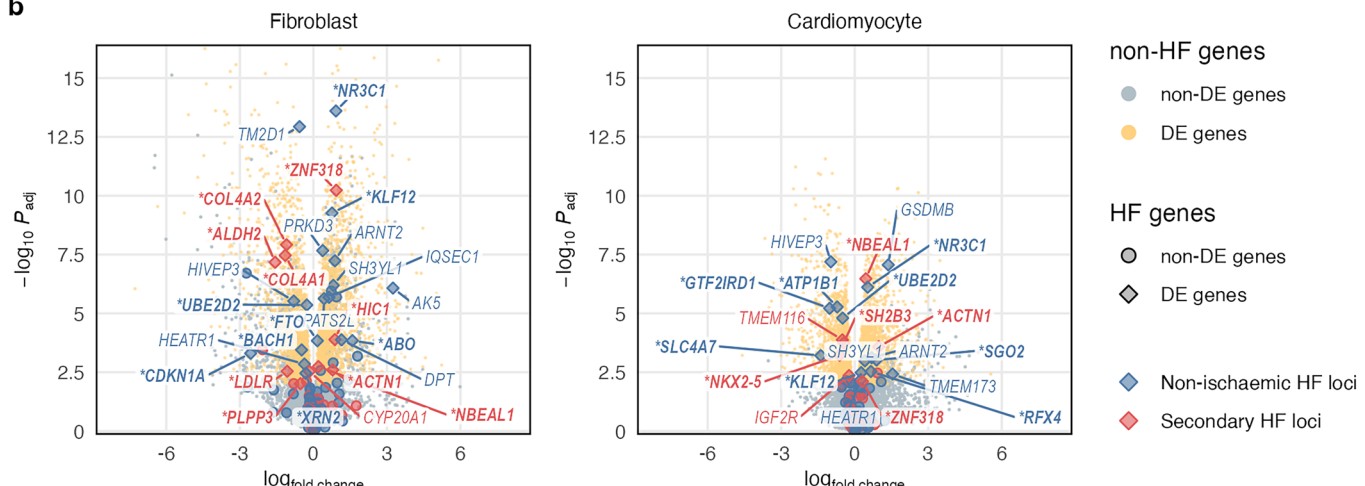

**Extended Data Fig. 7 | Differential gene expression of candidate heart failure effector genes across cardiac cell types from failing and non-failing hearts.** **a**, Differential gene expression profile for 51 candidate heart failure GWAS genes that are differentially expressed (DE) in at least one cardiac cell type, defined as survived multiple-testing correction at adjusted *P* value < 0.01 and showed a concordant differential expression sign in both CellBender and Cell Ranger quantifications and had no background contamination as estimated in CellBender (Methods). Red/blue colors represent higher/lower expression in

cells from failing heart (compared to non-failing heart); gray color represents undetectable expression; yellow dots represent differential expression as defined above. **b**, Volcano plots showing differential gene expression between cells (fibroblasts and cardiomyocytes) from failing and non-failing heart samples. Candidate and prioritized heart failure (HF) genes (the latter is prefixed with *) that are differentially expressed (DE) are highlighted. The presented *P* values are derived from two-sided statistical tests and adjusted for multiple testing using the Benjamini–Hochberg procedure.

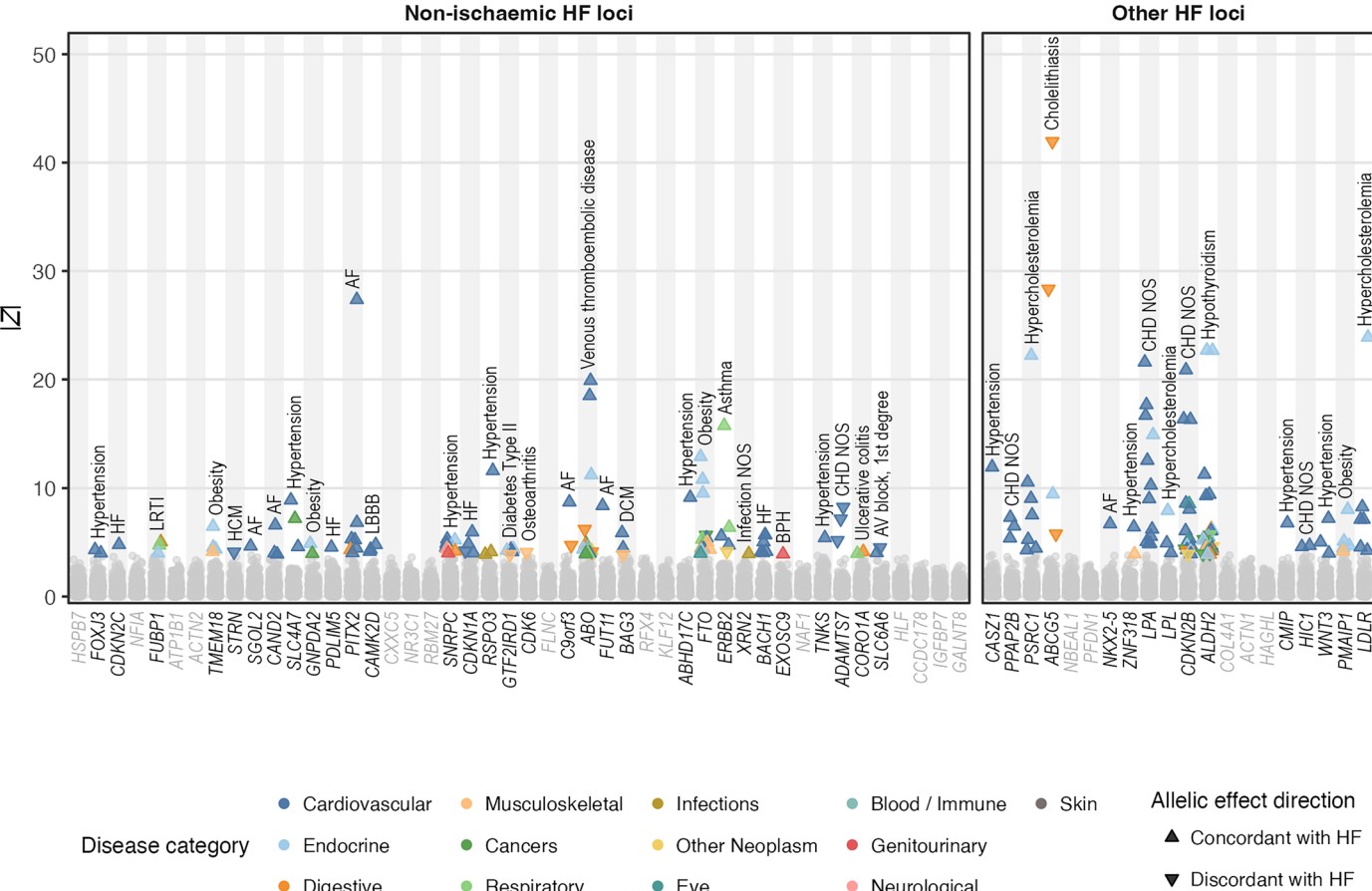

**Extended Data Fig. 8 | Phenome-wide association of heart failure sentinel genetic variants.** Association estimates are presented in absolute *z* score (vertical axis) for 294 disease phenotypes and 66 sentinel genetic variants across heart failure (HF) susceptibility loci (horizontal axis). Estimates which survived multiple-testing correction at false discovery rate (FDR) < 1% are highlighted, with top associated phenotype (largest absolute *z* score) per locus labeled. Loci are labeled by the prioritized gene and grayed out if no FDR-passing association is identified. Phenotype abbreviations and the numerical estimates are presented in Supplementary Table 15.

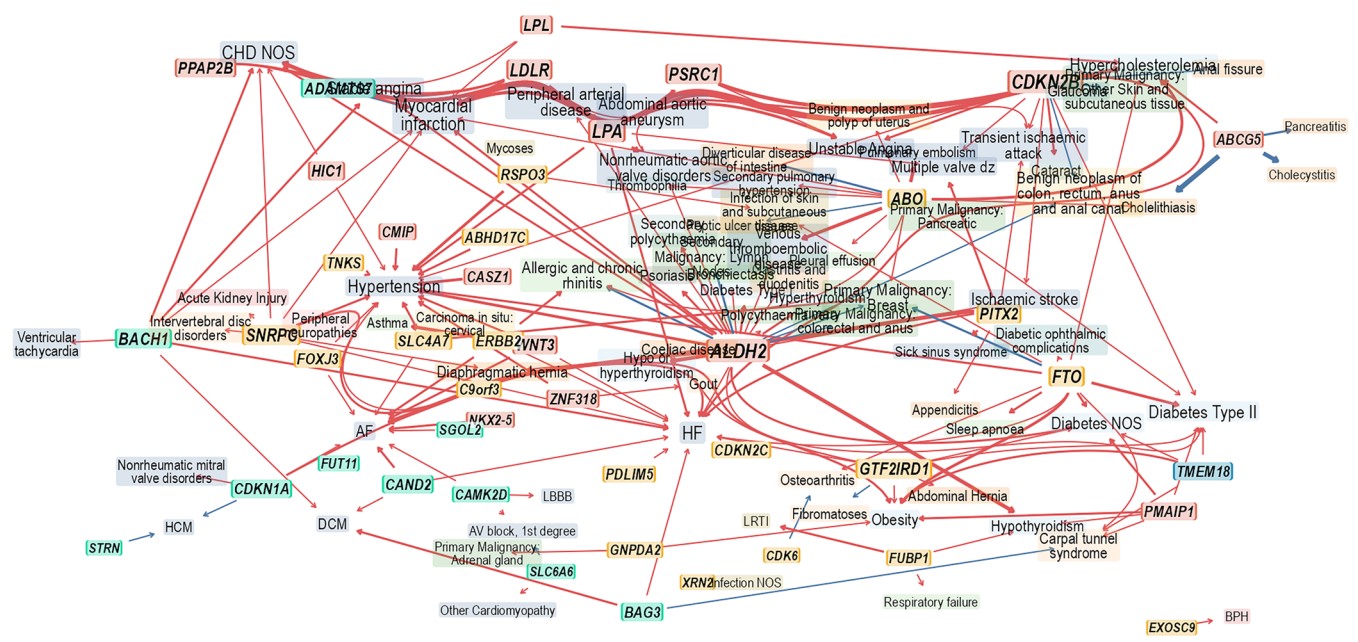

**Extended Data Fig. 9 | See next page for caption.**

**Extended Data Fig. 9 | Pleiotropy network and etiologic clusters of heart failure.** The network is constructed from 207 genotype–phenotype associations across 79 unique diseases and 46 heart failure susceptibility loci identified from phenome-wide association (PheWAS) analysis at false discovery rate <1%. Nodes represent genetic loci labeled by the prioritized gene (solid background with bold–italic label, colored by categorical association across heart failure phenotypes) and phenotypes (translucent background, colored by phenotype category), sized proportionally to centrality measure. Edges (arrows connecting locus nodes to phenotype nodes) represent association, with thickness representing strength of association measured by absolute $z$ score. **a**, Full network constructed using Davidson–Harel layout with edge bundling. **b**, Annotated network showing 18 etiological clusters identified using walktrap community detection algorithm. Phenotype abbreviations, phenotype categories and phenome-wide association results are presented in Supplementary Table 15 and Extended Data Fig. 8. Individual etiological clusters are presented in Fig. 4. Cluster membership and centrality measures of nodes are presented in Supplementary Table 26 and Supplementary Fig. 13.

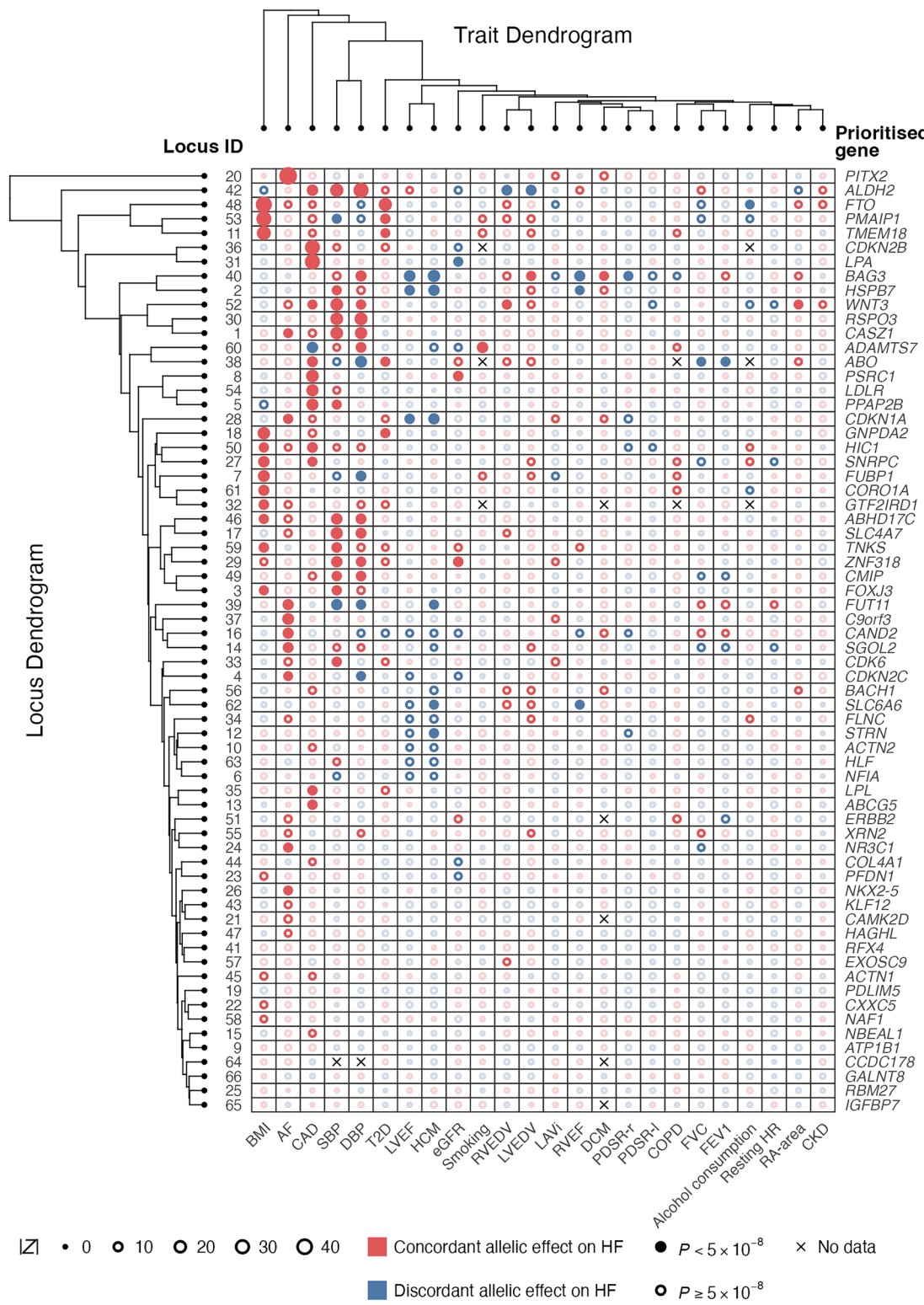

**Extended Data Fig. 10 | Associations between heart failure sentinel genetic variants and related traits.** Circle data points represent association estimates in absolute $Z$ score between 66 sentinel genetic variants within heart failure susceptibility loci (rows) and 24 related traits (columns). Non-transparent circles represent associations passing false discovery rate <1%. Solid circles with full color represent associations at $P < 5 \times 10^{-8}$, tested using two-sided statistical tests for association in the corresponding GWAS. Loci and traits were arranged based on hierarchical agglomerative clustering results, represented by dendrograms on the edges of the plot. Trait description and source for GWAS summary statistics are provided in Supplementary Table 27.

|---|---|

# Reporting Summary

## Statistics

For all statistical analyses, confirm that the following items are present in the figure legend, table legend, main text, or Methods section.

| n/a | Confirmed | |
|---|---|---|
| ☐ | ☒ | The exact sample size (*n*) for each experimental group/condition, given as a discrete number and unit of measurement |
| ☐ | ☒ | A statement on whether measurements were taken from distinct samples or whether the same sample was measured repeatedly |
| ☐ | ☒ | The statistical test(s) used AND whether they are one- or two-sided *Only common tests should be described solely by name; describe more complex techniques in the Methods section.* |
| ☐ | ☒ | A description of all covariates tested |
| ☐ | ☒ | A description of any assumptions or corrections, such as tests of normality and adjustment for multiple comparisons |
| ☐ | ☒ | A full description of the statistical parameters including central tendency (e.g. means) or other basic estimates (e.g. regression coefficient) AND variation (e.g. standard deviation) or associated estimates of uncertainty (e.g. confidence intervals) |
| ☐ | ☒ | For null hypothesis testing, the test statistic (e.g. *F*, *t*, *r*) with confidence intervals, effect sizes, degrees of freedom and *P* value noted *Give P values as exact values whenever suitable.* |
| ☒ | ☐ | For Bayesian analysis, information on the choice of priors and Markov chain Monte Carlo settings |
| ☐ | ☒ | For hierarchical and complex designs, identification of the appropriate level for tests and full reporting of outcomes |
| ☐ | ☒ | Estimates of effect sizes (e.g. Cohen's *d*, Pearson's *r*), indicating how they were calculated |

*Our web collection on statistics for biologists contains articles on many of the points above.*

## Software and code

Policy information about availability of computer code

| Data collection | Details of phenotype definition and softwares used to perform genome-wide association analysis at study level is provided on Supplementary Tables 17, 18, and 19. A sample code to define heart failure phenotypes implemented in UK Biobank is available on: https://github.com/ihi-comp-med/ukb-hf-phenotyping. |
|---|---|
| Data analysis | GWAS meta-analysis was performed centrally using METAL v2020-05-05. SNP-based heritability was calculated using LDAK SumHer software v5.2. Genetic correlation between traits was performed using LDSC v1.0.1. Conditionally independent variants were identified using GCTA software v1.92.4. Causal variant fine-mapping was performed using PolyFun v2020-11-14 and SuSiE v0.11.92. Polygenic prediction score (PoPS) v0.1, OpenTargets Variant2Gene v1.1, S-MultiXcan v0.7.3, MendelVar v2023-12-05, coloc R package v5.2.3, ABC-Max v2021-04-08, and python scikit-learn v1.5.1 were used for effector gene prioritisation. Pathway enrichment of prioritised genes was performed using g:profiler v0.2.3 package in R. Tissue-based and cell-type enrichment was performed using S-LDSC v1.0.1 and MAGMA v1.10. Polygenic scores were derived using LDpred2-auto model implemented in bigsnpr R package v1.12.16, with individual PGS scores generated using PLINK v1.9. Phenome-wide association study of sentinel variants was performed using PLINK v2.0. Network analysis was performed using tidygraph v1.2.3 and visualised using ggraph v2.0.0 in R. Custom code to perform the main analyses is available on Zenodo100 https://doi.org/10.5281/zenodo.11204854. |

For manuscripts utilizing custom algorithms or software that are central to the research but not yet described in published literature, software must be made available to editors and reviewers. We strongly encourage code deposition in a community repository (e.g. GitHub). See the Nature Portfolio guidelines for submitting code & software for further information.

## Data

Policy information about availability of data

All manuscripts must include a data availability statement. This statement should provide the following information, where applicable:
- Accession codes, unique identifiers, or web links for publicly available datasets
- A description of any restrictions on data availability
- For clinical datasets or third party data, please ensure that the statement adheres to our policy

GWAS summary statistics are available to download from the Cardiovascular Disease Knowledge Portal for the multi-ancestry meta-analysis: https://api.kpndataregistry.org/api/d/6Ls5Wu and for the European ancestry subset meta-analysis: https://api.kpndataregistry.org/api/d/6eJqWn. A summary of regional genetic associations, gene prioritization scores, cross-trait association, and study-level estimates across identified GWAS loci is provided on Supplementary Data 1 and online on https://hermes2-supp-note.netlify.app/locus_desc.html. A summary of study-level quality control is provided on Supplementary Data 2 and online on https://hermes2-supp-note.netlify.app/hf_subtypes_qc.html.

## Research involving human participants, their data, or biological material

Policy information about studies with human participants or human data. See also policy information about sex, gender (identity/presentation), and sexual orientation and race, ethnicity and racism.

| | |
|---|---|
| Reporting on sex and gender | This study uses the term sex when referring to biological attribute, and was determined using genetic data where available. Sex was included as a covariate in genetic association, polygenic risk score, and phenome-wide association analyses. A sex-stratified analysis with meta-regression was performed to explore sex-differentiated effects of genetic variants. Findings are relevant to both male and females. |
| Reporting on race, ethnicity, or other socially relevant groupings | The GWAS meta-analysis are performed using samples of 5 major ancestry groups, named accordingly to follow superpopulation grouping used in the 1000 Genomes Projects: European (EUR), African (AFR), East Asian (EAS), South Asian (SAS), and Admixed American / Hispanic (AMR). To control for potential bias, we performed the meta-analysis separately and calculated the heterogeneity of ancestry-specific allelic effect size as described in the Online Method section. |
| Population characteristics | A total of 1,946,349 individuals from 42 studies were included in the meta-analysis, including 153,174 cases of heart failure encompassing 5 ancestry groups: 139,533 (91%) European; 9,413 (6.2%) East Asian; 3,292 (2.2%) African; 779 (0.5%) South Asian; and 157 (0.1%) Admixed American (Supplementary Table 2). The mean age across participating studies ranges from 54.6 years to 78.8 years amongst cases, and 40.0 to 74.6 years amongst controls. Sex proportion ranges from 0% to 100% males. Details of study-level participant characteristics are provided on Supplementary Table 1. |
| Recruitment | Participants were recruited and consented by individual participating study according to local protocol as detailed in Supplementary Table 21. |
| Ethics oversight | This study complies with the ethical regulations provided by the University College London Research Ethics Committee. All participating studies were ethically approved by local committees and all study participants provided written informed consent (Supplementary Table 21). |

Note that full information on the approval of the study protocol must also be provided in the manuscript.

# Field-specific reporting

Please select the one below that is the best fit for your research. If you are not sure, read the appropriate sections before making your selection.

☒ Life sciences ☐ Behavioural & social sciences ☐ Ecological, evolutionary & environmental sciences

For a reference copy of the document with all sections, see nature.com/documents/nr-reporting-summary-flat.pdf

# Life sciences study design

All studies must disclose on these points even when the disclosure is negative.

| | |
|---|---|
| Sample size | No sample size calculations were made.To minimise small sample bias, we filtered variant based on minor allele frequency and effect allele count. We used the maximum number of available cases and controls that passed quality control thresholds/metrics as described in the Online Methods section. |
| Data exclusions | No data were excluded |
| Replication | All available samples were included in the discovery genome-wide association analysis. The consistency of the identified genetic effects across independent datasets, assess by examining heterogeneity, serves an indirect form of replication. This approach is well established for large-scale genetic meta-analysis. |
| Randomization | Observational study - not applicable |

| Blinding | Observational study - not applicable |
|---|---|

# Reporting for specific materials, systems and methods

We require information from authors about some types of materials, experimental systems and methods used in many studies. Here, indicate whether each material, system or method listed is relevant to your study. If you are not sure if a list item applies to your research, read the appropriate section before selecting a response.

## Materials & experimental systems

| n/a | Involved in the study |
|---|---|
| ☒ ☐ | Antibodies |
| ☒ ☐ | Eukaryotic cell lines |
| ☒ ☐ | Palaeontology and archaeology |
| ☒ ☐ | Animals and other organisms |
| ☒ ☐ | Clinical data |
| ☒ ☐ | Dual use research of concern |
| ☒ ☐ | Plants |

## Methods

| n/a | Involved in the study |
|---|---|
| ☒ ☐ | ChIP-seq |
| ☒ ☐ | Flow cytometry |
| ☒ ☐ | MRI-based neuroimaging |

## Plants

| Seed stocks | n/a |
|---|---|

| Novel plant genotypes | n/a |
|---|---|

| Authentication | n/a |
|---|---|

