## [Peer Review File · Nature Genetics]

Genome-wide association study meta-analysis provides insights into the aetiology of heart failure and its subtypes

Corresponding Author: Dr R Lumbers

This manuscript has been previously reviewed at another journal. This document only contains reviewer comments, rebuttal and decision letters for versions considered at Nature Genetics.

Version 0:

Decision Letter:

10th November 2023

Dear Dr. Lumbers,

Your Article "Mapping the aetiological foundations of the heart failure spectrum using human genetics" has been seen by three referees. You will see from their comments below that, while they find your work of interest, they have raised several relevant points. We are interested in the possibility of publishing your study in Nature Genetics, but we would like to consider your response to these points in the form of a revised manuscript before we make a final decision on publication.

To guide the scope of the revisions, the editors discuss the referee reports in detail within the team, including with the chief editor, with a view to identifying key priorities that should be addressed in revision, and sometimes overruling referee requests that are deemed beyond the scope of the current study. In this case, we particularly ask that you clarify the criteria used to classify cases into clinical subtypes, address all technical queries related to the association and Mendelian randomization analyses, extend the analyses where feasible as suggested by the referees, and revise the presentation for clarity throughout. We hope you will find this prioritized set of referee points to be useful when revising your study. Please do not hesitate to get in touch if you would like to discuss these issues further.

We therefore invite you to revise your manuscript taking into account all reviewer and editor comments. Please highlight all changes in the manuscript text file. At this stage, we will need you to upload a copy of the manuscript in MS Word .docx or similar editable format.

*2) If you have not done so already, please begin to revise your manuscript so that it conforms to our Article format instructions, available

[here](http://www.nature.com/ng/authors/article_types/index.html).

*3) Include a revised version of any required Reporting Summary: <https://www.nature.com/documents/hr-reporting-summary.pdf>

Please be aware of our [guidelines](https://www.nature.com/nature-research/editorial-policies/image-integrity) on digital image standards.

Link Redacted

We hope to receive your revised manuscript within 8-12 weeks. If you cannot send it within this time, please let us know.

Nature Genetics is committed to improving transparency in authorship. As part of our efforts in this direction, we are now requesting that all authors identified as 'corresponding author' on published papers create and link their Open Researcher and Contributor Identifier (ORCID) with their account on the Manuscript Tracking System (MTS), prior to acceptance. ORCID helps the scientific community achieve unambiguous attribution of all scholarly contributions. You can create and link your ORCID from the home page of the MTS by clicking on 'Modify my Springer Nature account'. For more information, please visit www.springernature.com/orcid.

Sincerely,
Kyle

Kyle Vogan, PhD
Senior Editor
Nature Genetics
<https://orcid.org/0000-0001-9565-9665>

Referee expertise:

Referee #1: Genetics, cardiovascular diseases, functional genomics

Referee #2: Genetics, cardiovascular diseases, clinical translation

Referee #3: Genetics, cardiovascular diseases, clinical translation

Reviewers' Comments:

Reviewer #1:
Remarks to the Author:

In this study by Henry et al, the authors perform a meta-analysis for heart failure (HF) phenotypes including all forms of HF, non-ischemic HF, non-ischemic HF_rEF, and non-ischemic HF_pEF across 1.9 individuals, including 153,174 cases of HF and smaller fractions of cases for the non-ischemic HF types. They identify 66 loci, of which 37 were not previously reported, and apply a series of analyses to define contributing risk factors, effector genes, correlated phenotypes, enriched pathways, and cell types. They highlight extra-cardiac tissues (e.g. kidney and vasculature) underlying HF_pEF. While this study performed an impressive number of analyses, there are several major limitations that should be addressed to improve the quality of the manuscript. Also, the presentation of the results and the writing is a bit difficult to follow and could be made more accessible to a broad audience.

Major comments

1. One of the limitations of this study is the use of highly heterogenous HF definitions across the different studies, as depicted in Supplementary Table 19. Given these differences in prevalent vs incident cases, physician vs non-physician definitions, historical EHR phecodes, and inconsistent imaging data, it appears challenging to adjudicate these HF cases across widely different studies. Further evidence should be provided to show that the defined cases are consistent with cardiac MRI based phenotypes to support their study design.
2. Another known limitation is the lack of diversity in their cases to identify true ancestry-specific associations. The authors report the percentage of ancestry groups in the total study population however the breakdown by ancestry is not provided for the cases in the text. Based on the numbers in the Supplementary Table, only 0.2% of the cases were from East Asian and only 0.1% of the cases were from Hispanic ancestry groups. This should be clearly stated in the text that the cases were from

99% European ancestry, rather than 86% (which reflects the controls). This is further evident as the authors did not perform cross-ancestry analyses for the non-ischemic HF phenotypes. Thus, the use of “ancestrally diverse individuals” should be omitted from the abstract to avoid over-stating this very minimal diversity.

3. Another obvious limitation is the lack of any sex-stratified analyses for HF. The authors should have sufficient power to perform these analyses, so it would be helpful to provide the sex interaction results rather than adjusting for sex as a covariate.

4. This study identified several biologically relevant genes and highlight one of the top effector genes for HFpEF, IGFBP7, but do not provide any external data or functional studies to support this finding. Given the breadth of analyses performed here, it would strengthen the study to go deeper into this locus to provide more functional and mechanistic insights for the association.

5. Surprisingly, the authors do not leverage any epigenomic based methods as part of their gene prioritization strategy. For instance, they could apply enhancer-promoter or higher order chromatin contact data to assign target genes, which would be non-overlapping from their gene expression-based prioritization methods.

6. The pathway and tissue enrichment are not as informative, and the authors should consider moving this to the supplement or combining with a snRNA-seq based LDSC analysis to prioritize the cell types for different HF traits. It is surprising the authors did not observe greater enrichment in their LDSC-SEG analysis using the single-cell datasets. The authors should repeat this analysis using different input gene list sizes. Also, the authors should compare these results using gene-based association statistics (e.g., MAGMA).

Minor comments

7. Some of the writing style is a bit convoluted and difficult to follow. For example, “By leveraging the de-confounding properties of germline variation, we systematically appraise the aetiology of HF.” This could be more clearly written, and there are several places throughout the text. Also, some of the sentences in the intro and discussion are a bit distracting. For example, “The processes of evolution to animal life on land led to exquisite systems of body water regulation but an inherent susceptibility to systemic and pulmonary oedema”. The authors should avoid the prose and get more to the point of the manuscript.

8. Figure 1: Overall, this Figure is not as informative, and it seems like it would be better suited as a Table. It might be better to create a summary plot of the effect size by effect allele frequency, along with the annotated genes. The boxes with different p-values for evidence strength are not explained in the legend.

9. Figure 2: This figure is also very complicated and difficult to follow. It might be easier to separate out the different non-ischemic HF and secondary HF overlaps with prioritization methods.

10. The authors prioritization of effector genes mainly relies on gene expression datasets from GTEx, however they also mention data from the MAGnet study, which is from heart failure patients. It would be more ideal to use gene expression data from MAGnet if possible.

11. The methods for the snRNA-seq data generation and processing are inadequate and further analyses are justified to properly assign target cell types from these association data. Otherwise, it is not clear how these data were critical to the prioritization pipeline.

12. The bivariate Mendelian randomization results showing the lack of effects of CAD on non-ischemic heart failure outcomes is surprising given the authors previous work demonstrating that CAD is causally associated with HR (OR ~1.4). Were patients with CAD excluded in the non-ischemic HF groups but not the HF group?

13. The authors should provide the number of instruments used for each of the MR run multivariable MR to test the causal effects of CAD and BP traits to limit the potential confounding effects when doing bivariate MR with binary outcomes. Also, some of the MR forest plots are missing diamonds in the figure. This should be explained in the figure legends and text.

Reviewer #2:

Remarks to the Author:

Thank you for the opportunity to review the manuscript, “Mapping the aetiological foundations of the heart failure spectrum using human genetics,” by Henry and colleagues. Here, the authors conduct a large-scale predominantly European-ancestry GWAS meta-analysis of heart failure NOS (>150,000 cases), as well as moderately-well defined heart failure subtypes (“non-ischemic heart failure” NOS [~44,000 cases], non-ischemic heart failure with LV systolic dysfunction [~5,000 cases], and non-ischemic heart failure with normal LV systolic function [~4,000 cases]). Notably, the smaller GWAS are nested sub-studies of the larger GWAS. From these analyses, the authors identify 37 novel loci and use multiple strategies to identify candidate effector genes. Beyond the novel loci, the authors conduct typical downstream analyses that provide reassurance of the quality of the GWAS. The pleiotropy clustering analysis is interesting, highlighting some specific (known) pathways to heart failure (e.g. atherosclerosis [cluster 1], arrhythmia/cardiomyopathy [cluster 2], hypertension [cluster 4], and metabolic

syndrome [cluster 5]), but the other clusters only contain single genes, somewhat weakening their importance. A primary strength of this manuscript is the HF phenotype. The authors go beyond diagnosis codes and, when possible, use adjudication and medications (i.e. use of diuretics) to identify subjects with clinical heart failure. Overall, I find this manuscript interesting and well written. Summary statistics for HF will be valuable to the community, and they highlight some novel loci for further investigation.

MAJOR:

1. As the authors note, heart failure is a clinical diagnosis that reflects a syndrome rather than a specific disease/etiology. Classically, we first categorize heart failure into heart failure with reduced ejection fraction (HFrEF) and heart failure with preserved ejection fraction (HFpEF). Each of these sub-categories is extremely heterogeneous. HFpEF may be caused by hypertensive heart disease, aortic stenosis, infiltrative disease, hypertrophic cardiomyopathy, etc. As for HFrEF, we traditionally assess this syndrome by first determining if it is most likely caused by ischemic cardiomyopathy or non-ischemic cardiomyopathy. Rather than follow these conventions, the authors chose a different approach. Their approach may be reasonable but requires some explanation and better clarity in the text. Specifically, they chose to first identify the subset of HF cases for which there was no evidence of coronary artery disease, valvular disease, or congenital heart disease. To me, this choice implies that the authors are hoping to specifically study intrinsic cardiomyopathies. They label this set non-ischemic heart failure (ni-HF), which is not a conventional term. Then, among the ni-HF group, they stratify into ni-HF with reduced ejection fraction and ni-HF with preserved ejection fraction. They again use an unconventional labeling, by defining HFrEF as LVEF<50%, whereas the standard definition is LVEF<40%. Given these definitions and terminologies are not standard, it would be helpful to the reader if the authors could more clearly define each phenotype and their relationship to one another early on in the Results section. Additionally, it may be helpful to include Appendix 7 as a supplementary Figure. Currently, some of the key details are spread across the Methods, Supplementary Note, and Appendix.
2. According to Supplementary Table 16, some of the cases and controls for the non-ischemic phenotypes have myocardial infarction (MI) at baseline. Should these subjects have been excluded from the non-ischemic phenotypes?
3. Can the authors provide some clarification on how the GWAS meta-analysis was conducted? By my reading, I believe they simply meta-analyzed all the GWAS together. However, I can also imagine an approach where they first meta-analyze GWAS stratified by ancestry and then meta-analyze those ancestry-specific GWAS together.
4. Throughout, the authors describe their analysis as “multi-ancestry” or “ancestrally diverse.” I think this description is disingenuous. >90% of their HF cases are subjects of European ancestry. As for the HF sub-categories, I did not see explicit numbers, but based on Supplementary Figure 2, I would estimate that these analyses harbor even less diversity. Furthermore, several of the downstream analyses used the only the European-ancestry GWAS. I think this study is better described as a predominantly European-ancestry GWAS. Additionally, since the authors perform analyses with the European-ancestry subset, and since many readers may want to use those data, the authors should make sure to report the number of cases and controls for this subset for each phenotype.
5. I agree with reporting all associations that meet genome-wide significance (GWS). However, it is worth noting that these GWAS reflect repeat analysis on subsets of the same cases/controls. Thus, there is additional multiple hypothesis testing going on. In this setting, the authors should note if some GWS loci are no longer significant after correcting for the fact that they conducted GWAS on different cuts of the same cases/controls four times. This caveat is important, as they highlight the IGFBP7 association with HFpEF as a particularly novel finding, and if it is not significant after accounting for multiple testing, this should be noted for transparency.
6. The authors find evidence of genomic inflation in their GWAS. They then argue that this inflation is due to polygenicity rather than population structure based on the LDSC intercept. However, they conducted their LDSC analysis only using the European-ancestry GWAS, which reduces confounding from population structure compared to the full GWAS. This approach seems a bit circular.
7. The authors state that ni-HFpEF had stronger genetic correlations with HF and ni-HF than ni-HFrEF had with HF and ni-HF (lines 339-342). This finding is counterintuitive. Further, Figures 1b and 1c suggest the opposite pattern, with ni-HFrEF seeming to have greater potential overlap in signals across the GWS loci. Can the authors confirm the genome-wide genetic correlation findings, and if so, what might be some plausible explanations for this discrepancy?
8. The authors conduct several 2-sample MR analyses. Many of the GWAS used as outcomes were meta-analyses that included data from various cohorts and biobanks. Thus, there is the potential for significant sample overlap between their exposure and outcome GWAS. For example, the CAD GWAS used (PMID: 36474045) includes subjects from deCode, UKB, MGB, etc. For each MR, the authors should use outcome GWAS from non-overlapping samples.

MINOR:

1. There is a typo in the legend for Supp Fig 1. “ni-HFrEF: non-*ischaemic* heart failure with preserved ejection fraction”
2. Line 523: Should the colocalization analyses reference Figure 5 instead of 6?

3. What LD reference panel was used for LDpred2-auto?

Reviewer #3:

Remarks to the Author:

A: Summary of key results:

A GWAS of 1.9 million individuals including 153,174 heart failure patients is presented. It reveals 66 genetic loci, of which 37 are newly reported. There is a great deal of downstream analysis which tends to support the notion that loci of relevance to heart failure have indeed been identified. However, this reviewer found those analyses rather disappointing regarding new insights into HF pathophysiology. It was not straightforward to extract the main messages from the very large amount of data presented.

There are several statements in the causal analyses that would be considered controversial with regard to causal analyses - for example, an opposite-direction association between CKD and HFpEF; no causal association between T2D and HF - and these should be more critically examined in the discussion rather than speculatively treated at present. It is stated (line 562) that there was evidence of the contribution of epicardial coronary artery disease to HF susceptibility, but only in patients who had had revascularisation or had an MI. Where was the evidence for this presented in the results?

Asserting in the Discussion that HFpEF is chiefly ascribed to diastolic dysfunction, for which the study finds no evidence and therefore challenges received wisdom, is putting up a straw man. In the opinion of this reviewer, HFpEF is already widely recognised as a highly complex and heterogeneous phenotype, not simply the result of primary diastolic dysfunction. It is, however, surprising that no association between diastolic dysfunction and HFpEF was found, and clinician readers will likely find it so. Accordingly, please set out the strength of the evidence underpinning this conclusion (how many patients is it based on, what phenotypic measurements) and more critically evaluate the conclusion. Similarly, this reviewer believes it is already accepted by HFpEF researchers that the kidney, metabolic and vascular tissues play important roles in the aetiology of the condition. While this genetic evidence is (line 580) in agreement, it is not the first recognition of this.

The clarity of the results could be improved if the associated variants were presented classified dependent on whether they had been previously found in GWAS studies of coronary disease and the major clinical risk factors for HF. This is not in any single place in the manuscript, rather it is piecemeal through the text. It is not possible to tell how many variants are likely working through increasing risk of coronary disease, risk of hypertension, risk of obesity and so forth (though these loci are present in the gene list e.g. CDKN2A/B, FTO, etc.) This reviewer believes it to be important to report the variants in this way as it will help reveal the extent to which this work has indeed identified new biology and increased our understanding of the HF phenotype's dependence on known risk factor pathways. "Cluster 4" in the graph modelling analyses (line 451 ff) centres on ALDH2. Alcohol is an important risk factor for HF - what can be said about the interplay of ALDH2 genotype, alcohol and HF risk in the dataset?

The authors calculate a PRS with a view to evaluating its clinical utility. But they do not present the PRS discriminatory value in a way that would be useful for screening, or comment further in the discussion on this. Please present the PRS in a way that enables its potential utility to be estimated - e.g. stating what proportion of cases would be picked up with a 5% false positive rate. Please make comments on this in the discussion.

This reviewer believes that an important limitation of this study, that should be discussed, is the small size of the non-ischemic HFpEF cohort even in such a large international collaborative study. Only a small number of GWAS significant loci were detected. The study seems to reveal a clear need for larger genetic studies specifically focused on HFpEF and could call for this.

B: Originality and significance

This is the largest GWAS study of heart failure yet performed, involving a very large amount of data contributed by many groups around the world, and it has identified a number of new loci.

C: Data and methodology

Patient selection is complex and difficult to follow. 153174 HF patients were in the overall cohort. ~44K were identified as "non-ischemic". Does that mean the remainder were all "ischemic" or could some in that group not be classified? Among the 44K, much smaller numbers are classified as "non-ischemic HFrEF" (5,406) and "non-ischemic HFpEF" (3,841), but in the Figure 1 schema it seems that all of the 44K should be classified into HFpEF or HFrEF dependent on EF <=>50%. The current definition of HFrEF requires EF<40%, with HFpEF classified as EF>50%, why was that not followed here? Alternatively, and since the Figure 1a classification scheme seems tautologous, is there a misprint (should "<50%" be "<40%")? Does the classification imply that the remaining ~34K patients could not be classified as either HFpEF or HFrEF? These questions need to be answered in the flow of the text, or the schema of Figure 1 needs to be expanded to answer them.

How does the inability to classify the majority of the non-ischemic patients as HFpEF or HFrEF affect what readers should take from the study? Is it a limitation? In the Discussion, it would be helpful to have some comment on these unclassified patients.

The presentation of the results is not optimal to clearly identify any new pathophysiological insights. The previous

identification of HF loci identified here, in GWAS studies of HF risk factors, including coronary artery disease, hypertension, obesity, diabetes, etc. is not straightforward to work out. The discussion contains no attempt to place the present work in context of previous studies of HF genetics (there is an assessment of agreement in the results). Basic information such as how many cases this study adds to previous work is needed by readers to interpret the significance of the study. Have any of the patients in this study been participants in previous HF GWAS studies?

This reviewer had difficulty following the rationale for classifying loci into “non ischemic” and “secondary” (line 320) – these are not complementary terms. Does it mean secondary to ischemia?

D: Statistics

Not the expertise of this reviewer.

E: Conclusions

Of the four key conclusions, the first, about the relationship with risk factors, is not clearly presented and some of the statements made would merit more critical evaluation. The second – the relationship between contractility traits and HF – is already known in single gene disease and the appropriate work should be discussed. The conclusions about HFpEF need to be more contextualised with the relatively small numbers of patients studied, a clear need for larger GWAS studies of HFpEF, and previous clinical studies that have also pointed the same way as these genetic findings. The fourth conclusion regarding novel molecular mechanisms is perhaps the most interesting, but in the presentation of the results this rather got lost in the profusion of downstream analyses.

F: Suggested improvements

See above comments.

G: References

See above comments.

H: Clarity and context

Suggestions for improvement of these aspects made above.

Version 1:

Decision Letter:

14th March 2024

Dear Dr. Lumbers,

Your revised Article "Mapping the aetiology of heart failure and its subtypes using human genetics" has been seen by two of the original referees. (Reviewer #2 was also contacted but was unable to provide comments on the revision.) You will see from the comments below that, while Reviewer #3 is satisfied and has no further requests, Reviewer #1 has a few ongoing concerns. We remain interested in the possibility of publishing your study in Nature Genetics, but we would like to consider your response to these ongoing concerns in the form of a further revision before we make a final decision on publication.

As before, to guide the scope of the revisions, the editors discuss the referee reports in detail within the team with a view to identifying key priorities that should be addressed in revision, and sometimes overruling referee requests that are deemed beyond the scope of the current study. In this case, although we appreciate that you intend to address some of the remaining requests at greater depth in future studies, we ask that you present sex-stratified association results for all genome-wide significant loci and state more explicitly the limitations and caveats of the two-sample Mendelian randomization analyses. We again hope you will find this prioritized set of referee points to be useful when revising your study. Please do not hesitate to get in touch if you would like to discuss these issues further.

We therefore invite you to revise your manuscript again taking into account all reviewer and editor comments. Please highlight all changes in the manuscript text file. At this stage, we will need you to upload a copy of the manuscript in MS Word .docx or similar editable format.

*1) Include a “Response to referees” document detailing, point-by-point, how you addressed each referee comment. If no action was taken to address a point, you must provide a compelling argument. This response will be sent back to the referees along with the revised manuscript.

*2) If you have not done so already, please begin to revise your manuscript so that it conforms to our Article format instructions, available

[here](http://www.nature.com/ng/authors/article_types/index.html).

*3) Include a revised version of any required Reporting Summary: <https://www.nature.com/documents/nr-reporting-summary.pdf>

Link Redacted

We hope to receive your revised manuscript within 4-8 weeks. If you cannot send it within this time, please let us know.

Nature Genetics is committed to improving transparency in authorship. As part of our efforts in this direction, we are now requesting that all authors identified as 'corresponding author' on published papers create and link their Open Researcher and Contributor Identifier (ORCID) with their account on the Manuscript Tracking System (MTS), prior to acceptance. ORCID helps the scientific community achieve unambiguous attribution of all scholarly contributions. You can create and link your ORCID from the home page of the MTS by clicking on 'Modify my Springer Nature account'. For more information, please visit www.springernature.com/orcid.

Sincerely,
Kyle

Kyle Vogan, PhD
Senior Editor
Nature Genetics
<https://orcid.org/0000-0001-9565-9665>

Referee expertise:

Referee #1: Genetics, cardiovascular diseases, functional genomics

Referee #3: Genetics, cardiovascular diseases, clinical translation

Reviewers' Comments:

Reviewer #1:
Remarks to the Author:

The authors attempt to address several of these comments, however there are too many unresolved concerns.

We appreciate the clarity on the phenotype definitions, however a sex-stratified analysis or sex interaction analysis for the top loci would seem appropriate here.

Also, it should be possible to add some functional evidence to support top loci.

Also, the authors should perform multivariable MR analyses instead of two sample MR, especially given the sample overlap between the exposure and outcome GWAS used.

Finally, the authors will need to modify the title as it should include "in European-ancestry populations". Also "human genetics" is too vague and does not provide details on the methodology used.

Reviewer #3:
Remarks to the Author:

Thanks to the authors for carefully considering and responding to the points made at first review. No further comments.

Version 2:

Decision Letter:

Our ref: NG-A63653R1

28th June 2024

Dear Dr. Lumbers,

Thank you for submitting your revised manuscript "Mapping the aetiology of heart failure and its subtypes using human genetics" (NG-A63653R1). In light of your responses to Reviewer #1, we will be happy in principle to publish your study in Nature Genetics as an Article pending final revisions to comply with our editorial and formatting guidelines.

We are now performing detailed checks on your paper, and we will send you a checklist detailing our editorial and formatting requirements soon. Please do not upload the final materials or make any revisions until you receive this additional information from us.

Thank you again for your interest in Nature Genetics. Please do not hesitate to contact me if you have any questions.

Sincerely,
Kyle

Kyle Vogan, PhD
Senior Editor
Nature Genetics
<https://orcid.org/0000-0001-9565-9665>

Editor's comment

Dear Dr. Lumbers,

Your Article "Mapping the aetiological foundations of the heart failure spectrum using human genetics" has been seen by three referees. You will see from their comments below that, while they find your work of interest, they have raised several relevant points. We are interested in the possibility of publishing your study in Nature Genetics, but we would like to consider your response to these points in the form of a revised manuscript before we make a final decision on publication.

To guide the scope of the revisions, the editors discuss the referee reports in detail within the team, including with the chief editor, with a view to identifying key priorities that should be addressed in revision, and sometimes overruling referee requests that are deemed beyond the scope of the current study. In this case, we particularly ask that you clarify the criteria used to classify cases into clinical subtypes, address all technical queries related to the association and Mendelian randomization analyses, extend the analyses where feasible as suggested by the referees, and revise the presentation for clarity throughout. We hope you will find this prioritized set of referee points to be useful when revising your study. Please do not hesitate to get in touch if you would like to discuss these issues further.

We therefore invite you to revise your manuscript taking into account all reviewer and editor comments. Please highlight all changes in the manuscript text file. At this stage, we will need you to upload a copy of the manuscript in MS Word .docx or similar editable format.

*2) If you have not done so already, please begin to revise your manuscript so that it conforms to our Article format instructions, available here.

*3) Include a revised version of any required Reporting Summary: <https://www.nature.com/documents/nr-reporting-summary.pdf>

Please be aware of our guidelines on digital image standards.

(redacted)

We hope to receive your revised manuscript within 8-12 weeks. If you cannot send it within this time, please let us know.

Nature Genetics is committed to improving transparency in authorship. As part of our efforts in this direction, we are now requesting that all authors identified as 'corresponding author' on published papers create and link their Open Researcher and Contributor Identifier (ORCID) with their account on the Manuscript Tracking System (MTS), prior to acceptance. ORCID helps the scientific community achieve unambiguous attribution of all scholarly contributions. You can create and link your ORCID from the home page of the MTS by

clicking on 'Modify my Springer Nature account'. For more information, please visit please visit www.springernature.com/orcid.

Sincerely,

Kyle

Kyle Vogan, PhD

Senior Editor

Nature Genetics

<https://orcid.org/0000-0001-9565-9665>

Response to reviewers' comments

Authors' responses are displayed in blue

Dear *Nature Genetics* Editorial Team,

We thank the editorial team and the reviewers for helpful feedback on our manuscript. We have revised the manuscript as suggested and the key changes are highlighted in red in the updated version. We have detailed our response to the reviewers below.

In response to the points highlighted by the editorial team: Firstly, we have improved the clarity of the description of the heart failure subtypes under study in the main text (**lines 282-289**) and in **Supplementary Figure 2** and **Supplementary Note**. Secondly, we have clarified technical queries related to genetic association and Mendelian randomization (MR) analyses, including multiple-testing adjustment (**lines 315-318**), limiting genomic inflation and polygenicity analysis to European-ancestry samples (**Supplementary Table 6**) and reporting the number of instruments for MR (**Supplementary Table 18**). Thirdly, we added the results of additional analyses suggested by reviewers, including reporting 19% of cases identified at 5% FPR in the polygenic risk score analysis (**line 341-344**), including predicted enhancer-gene activity in our gene prioritization pipeline (**line 794-802**), and performing gene-based association enrichment using MAGMA (**Supplementary Figure 14, Supplementary Table 11**).

Finally, we have revised the text throughout to improve clarity and have updated **Figure 1** and **Figure 2** as suggested by the reviewers. We have also reformatted the manuscript according to the *Nature Genetics* formatting guide.

We thank you for considering this revised manuscript for publication in *Nature Genetics*.

Yours sincerely

Tom

Dr Tom Lumbers MA PhD MRCP FESC

Referee expertise:

Referee #1: Genetics, cardiovascular diseases, functional genomics

Referee #2: Genetics, cardiovascular diseases, clinical translation

Referee #3: Genetics, cardiovascular diseases, clinical translation

Reviewers' Comments:

Reviewer #1:

Remarks to the Author:

In this study by Henry et al, the authors perform a meta-analysis for heart failure (HF) phenotypes including all forms of HF, non-ischemic HF, non-ischemic HF_rEF, and non-ischemic HF_pEF across 1.9 individuals, including 153,174 cases of HF and smaller fractions of cases for the non-ischemic HF types. They identify 66 loci, of which 37 were not previously reported, and apply a series of analyses to define contributing risk factors, effector genes, correlated phenotypes, enriched pathways, and cell types. They highlight extra-cardiac tissues (e.g. kidney and vasculature) underlying HF_pEF. While this study performed an impressive number of analyses, there are several major limitations that should be addressed to improve the quality of the manuscript. Also, the presentation of the results and the writing is a bit difficult to follow and could be made more accessible to a broad audience.

Major comments

1. One of the limitations of this study is the use of highly heterogeneous HF definitions across the different studies, as depicted in Supplementary Table 19. Given these differences in prevalent vs incident cases, physician vs non-physician definitions, historical EHR phecodes, and inconsistent imaging data, it appears challenging to adjudicate these HF cases across widely different studies. Further evidence should be provided to show that the

defined cases are consistent with cardiac MRI based phenotypes to support their study design.

We thank the reviewer for this comment. We included the largest number of studies possible to maximise statistical power, however, we recognise that there is a trade-off with the specificity of the phenotype. We took steps to harmonise case ascertainment as much as possible, given the data types available in each study, by pre-specifying multimodal phenotype definitions in a common analysis plan (**Supplementary Note**). Where cases had been highly adjudicated in the context of a research study, we have ensured that the pre-existing definitions align with our recommendations, and described this in detail on **Supplementary Table 21**. Our pre-specified definitions allowed for some variation across studies while retaining specificity for different HF subtypes. We and others are currently engaged in prospective research studies to ascertain heart failure at scale (UK HFpEF and HeartShare), recognising the challenges of ascertainment from existing genetically-enriched resources.

2. Another known limitation is the lack of diversity in their cases to identify true ancestry-specific associations. The authors report the percentage of ancestry groups in the total study population however the breakdown by ancestry is not provided for the cases in the text. Based on the numbers in the Supplementary Table, only 0.2% of the cases were from East Asian and only 0.1% of the cases were from Hispanic ancestry groups. This should be clearly stated in the text that the cases were from 99% European ancestry, rather than 86% (which reflects the controls). This is further evident as the authors did not perform cross-ancestry analyses for the non-ischemic HF phenotypes. Thus, the use of “ancestrally diverse individuals” should be omitted from the abstract to avoid over-stating this very minimal diversity.

We thank the reviewer for this comment. We have edited the abstract, results reporting, and the discussion to reflect the limited proportion of the overall case sample with non-European ancestry. The limited ancestral diversity of the sample is identified as a limitation of the study (**line 573-575**).

3. Another obvious limitation is the lack of any sex-stratified analyses for HF. The authors should have sufficient power to perform these analyses, so it would be helpful to provide the sex interaction results rather than adjusting for sex as a covariate.

Thank you for this important suggestion. A sex-stratified analysis was beyond the scope of the current study, however, we are currently designing a subsequent collaborative project with a major focus on sex differences, including sex-stratified and X-chromosome analysis.

4. This study identified several biologically relevant genes and highlight one of the top effector genes for HFpEF, *IGFBP7*, but do not provide any external data or functional studies to support this finding. Given the breadth of analyses performed here, it would strengthen the study to go deeper into this locus to provide more functional and mechanistic insights for the association.

We thank the reviewer and agree that functional follow-up studies are essential to provide further evidence for the causal variants and genes at the identified genomic loci and to understand disease mechanisms in detail. We are planning a series of experimental follow-up studies. In the case of *IGFBP7*, cardiac dysfunction, adverse ventricular remodelling, and cellular senescence of cardiomyocytes were attenuated in a recently reported murine haploinsufficiency model ¹. We have edited the discussion to highlight the need and opportunity for further functional validation (**line 582-583**).

5. Surprisingly, the authors do not leverage any epigenomic based methods as part of their gene prioritization strategy. For instance, they could apply enhancer-promoter or higher order chromatin contact data to assign target genes, which would be non-overlapping from their gene expression-based prioritization methods.

Thank you for this helpful suggestion. In response, we have performed an additional analysis based on enhancer-gene activity maps in 131 human cell types and tissues as measured using the activity-by-contact (ABC) model, and included this in our gene prioritization pipeline (**line 794-802**)². Functional data from promoter capture Hi-C, Enhancer-TSS correlation, and DHS-promoter correlation are also included in the variant-to-gene (V2G) model which we implemented in our gene prioritization strategy³

6. The pathway and tissue enrichment are not as informative, and the authors should consider moving this to the supplement or combining with a snRNA-seq based LDSC analysis to prioritize the cell types for different HF traits. It is surprising the authors did not observe greater enrichment in their LDSC-SEG analysis using the single-cell datasets. The authors should repeat this analysis using different input gene list sizes. Also, the authors should compare these results using gene-based association statistics (e.g., MAGMA).

Many thanks for this comment. In response, we have moved the pathway enrichment analysis figure to **Supplementary Figure 13**, while retaining the tissue enrichment in the main displays (**Figure 3**). As suggested, we have repeated the cell-type enrichment analysis using MAGMA (**line 859-865**) and observed enrichments for cardiomyocytes and vascular smooth muscle, consistent with the results of our prior analysis (**line 399-402, Supplementary Figure 14, Supplementary Table 11**).

Minor comments

7. Some of the writing style is a bit convoluted and difficult to follow. For example, “By leveraging the de-confounding properties of germline variation, we systematically appraise the aetiology of HF.” This could be more clearly written, and there are several places throughout the text. Also, some of the sentences in the intro and discussion are a bit distracting. For example, “The processes of evolution to animal life on land led to exquisite systems of body water regulation but an inherent susceptibility to systemic and pulmonary oedema”. The authors should avoid the prose and get more to the point of the manuscript.

We appreciate this suggestion and have extensively updated the manuscript throughout, to simplify the presentation and improve clarity.

8. Figure 1: Overall, this Figure is not as informative, and it seems like it would be better suited as a Table. It might be better to create a summary plot of the effect size by effect allele frequency, along with the annotated genes. The boxes with different p-values for evidence strength are not explained in the legend.

We have updated **Figure 1** to improve the clarity as suggested, including (1) moving the data on effect allele, effect allele frequency, and heterogeneity to **Supplementary Table 3**; (2) simplifying the presentation of locus effects by HF subtype (**Figure 1c**); and (3) updating the legend for clarity and to explain the significance thresholds.

9. Figure 2: This figure is also very complicated and difficult to follow. It might be easier to separate out the different non-ischemic HF and secondary HF overlaps with prioritization methods.

Thank you for the suggestion. We have simplified the figure by 1) separating the circular heatmap with HF locus overlaps, 2) removing the histogram of prioritised gene counts, and 3) improving the readability by providing a schematic of the locus categorisation (**Figure 2b**) and a schematic of the gene prioritization strategy (**Figure 2d**).

10. The authors prioritization of effector genes mainly relies on gene expression datasets from GTEx, however they also mention data from the MAGnet study, which is from heart failure patients. It would be more ideal to use gene expression data from MAGnet if possible. We used data from GTEx for our primary gene prioritisation analyses because of the ability to evaluate genes that may not be primarily expressed in the heart, however, we agree that these data are limited by the available sample sizes. While we had access to single nucleotide expression data from a subset of MAGNet samples from failing and non-failing hearts, we did not have access to bulk gene expression data for use in QTL-based gene prioritisation. This is a limitation of the study and highlights the ongoing need for publicly available large QTL datasets from heart tissues.

11. The methods for the snRNA-seq data generation and processing are inadequate and further analyses are justified to properly assign target cell types from these association data. Otherwise, it is not clear how these data were critical to the prioritization pipeline.

Thank you for this comment. We have updated the methods section to provide a more detailed description of the data generation, processing, and downstream analyses (**line 847-858**). Our analysis method was implemented according to the approach described in detail in Chaffin et al. 2022 ⁴. Our analysis of single-cell data aimed to understand whether the prioritized heart failure genes were differentially expressed in failing vs non-failing hearts, but was not used in the prioritization pipeline.

12. The bivariate Mendelian randomization results showing the lack of effects of CAD on non-ischemic heart failure outcomes is surprising given the authors previous work demonstrating that CAD is causally associated with HR (OR ~1.4). Were patients with CAD excluded in the non-ischemic HF groups but not the HF group?

Individuals with coronary artery disease, defined by myocardial infarction or coronary revascularisation, were excluded from the non-ischaemic HF groups, in addition to congenital and severe valvular heart disease. Our Mendelian randomisation analysis demonstrated robust evidence consistent with the effects of liability to CAD on the risk of heart failure [$OR_{MR-IVW} = 1.20$, 95% CI = 1.18-1.22] but not on non-ischaemic HF however, we did not find evidence of the effects of liability to CAD on non-ischaemic HF [$OR_{MR-IVW} = 1.02$, 1.00-1.04]. These findings are consistent with our previous finding where we showed that the effect of CAD on HF is attenuated to the null after conditioning for CAD using mtCOJO (Figure 4, Shah et al.) ⁵.

13. The authors should provide the number of instruments used for each of the MR run multivariable MR to test the causal effects of CAD and BP traits to limit the potential confounding effects when doing bivariate MR with binary outcomes. Also, some of the MR forest plots are missing diamonds in the figure. This should be explained in the figure legends and text.

Thank you for the suggestion. In response, we have added the number of instruments used in each MR analysis in **Supplementary Table 18**. The legends on **Figure 6** have also been updated to improve clarity. The current estimates against the non-ischaemic heart failure phenotype estimate the effects of risk factors independent of CAD. We agree, however, that multivariable Mendelian randomisation may provide further insights into the mediation of specific risk factor effects. A detailed examination of mediation, including multivariable MR, is the subject of a planned follow-up manuscript. We have taken steps to identify and mitigate any possible confounding effects due to directional horizontal pleiotropy. We have included sensitivity analyses using pleiotropy robust methods, including estimates from MR-Egger and weighted median estimator.⁶

Reviewer #2:

Remarks to the Author:

Thank you for the opportunity to review the manuscript, “Mapping the aetiological foundations of the heart failure spectrum using human genetics,” by Henry and colleagues. Here, the authors conduct a large-scale predominantly European-ancestry GWAS meta-analysis of heart failure NOS (>150,000 cases), as well as moderately-well defined heart failure subtypes (“non-ischemic heart failure” NOS [~44,000 cases], non-ischemic heart failure with LV systolic dysfunction [~5,000 cases], and non-ischemic heart failure with normal LV systolic function [~4,000 cases]). Notably, the smaller GWAS are nested sub-studies of the larger GWAS. From these analyses, the authors identify 37 novel loci and use multiple strategies to identify candidate effector genes. Beyond the novel loci, the authors conduct typical downstream analyses that provide reassurance of the quality of the GWAS. The pleiotropy clustering analysis is interesting, highlighting some specific (known) pathways to heart failure (e.g. atherosclerosis [cluster 1], arrhythmia/cardiomyopathy [cluster 2], hypertension [cluster 4], and metabolic syndrome [cluster 5]), but the other clusters only

contain single genes, somewhat weakening their importance. A primary strength of this manuscript is the HF phenotype. The authors go beyond diagnosis codes and, when possible, use adjudication and medications (i.e. use of diuretics) to identify subjects with clinical heart failure. Overall, I find this manuscript interesting and well written. Summary statistics for HF will be valuable to the community, and they highlight some novel loci for further investigation.

MAJOR:

1. As the authors note, heart failure is a clinical diagnosis that reflects a syndrome rather than a specific disease/etiology. Classically, we first categorize heart failure into heart failure with reduced ejection fraction (HFrEF) and heart failure with preserved ejection fraction (HFpEF). Each of these sub-categories is extremely heterogeneous. HFpEF may be caused by hypertensive heart disease, aortic stenosis, infiltrative disease, hypertrophic cardiomyopathy, etc. As for HFrEF, we traditionally assess this syndrome by first determining if it is most likely caused by ischemic cardiomyopathy or non-ischemic cardiomyopathy. Rather than follow these conventions, the authors chose a different approach. Their approach may be reasonable but requires some explanation and better clarity in the text. Specifically, they chose to first identify the subset of HF cases for which there was no evidence of coronary artery disease, valvular disease, or congenital heart disease. To me, this choice implies that the authors are hoping to specifically study intrinsic cardiomyopathies. They label this set non-ischemic heart failure (ni-HF), which is not a conventional term. Then, among the ni-HF group, they stratify into ni-HF with reduced ejection fraction and ni-HF with preserved ejection fraction. They again use an unconventional labeling, by defining HFrEF as LVEF<50%, whereas the standard definition is LVEF<40%. Given these definitions and terminologies are not standard, it would be helpful to the reader if the authors could more clearly define each phenotype and their relationship to one another early on in the Results section. Additionally, it may be helpful to include Appendix 7 as a supplementary Figure. Currently, some of the key details are spread across the Methods, Supplementary Note, and Appendix.

We thank the reviewer for this helpful suggestion. In response, we have added a text in the introductory paragraph to introduce the phenotyping approach and rationale (line 282-288). The methods section has been edited to ensure that all the key information is provided. We

have removed the appendix and included all the information in the phenotype definition section of the **Supplementary Note**.

2. According to Supplementary Table 16, some of the cases and controls for the non-ischemic phenotypes have myocardial infarction (MI) at baseline. Should these subjects have been excluded from the non-ischemic phenotypes?

Thank you for highlighting this discrepancy. In our initial report, we note 417 MI cases from 6 studies were reported within 44,012 non-ischaemic HF cases. Upon investigation, we found 104 / 417 MI cases from 3 studies were incorrectly reported in the non-ischaemic HF group. The remaining studies had made an error in implementing the case definition that led to the inclusion of 313 / 417 prevalent MI cases.

Our aim in defining the non-ischaemic phenotypes was to examine HF in a sample depleted of CAD and other major risk factors by enriching the sample for genetic determinants to HF susceptibility independent of these. Cases with previous MI comprised 29.3% of the total in the overall HF group compared to only 0.7% in the non-ischaemic HF group (a more than 30-fold depletion). We have reported this number in the main text (line **302-303**) and updated the participant characteristics table (now **Supplementary Table 1**).

3. Can the authors provide some clarification on how the GWAS meta-analysis was conducted? By my reading, I believe they simply meta-analyzed all the GWAS together. However, I can also imagine an approach where they first meta-analyze GWAS stratified by ancestry and then meta-analyze those ancestry-specific GWAS together.

Thank you for this question. Indeed, the main meta-analysis included all study-level GWAS from all ancestry groups. As a sensitivity analysis, we performed ancestry-specific and trans-ancestry meta-analysis using the MR-MEGA method, to identify and account for allelic effect heterogeneity across ancestries.⁷ The MR-MEGA results were consistent with our main fixed-effect meta-analysis, with minimal allelic effect heterogeneity across ancestries (**Supplementary Figure 4**), and these are presented as the main GWAS findings. We have updated the main text to clarify this (**line 307-309**):

We performed a cross-ancestry, fixed-effect inverse variance-weighted (IVW) meta-analysis including all contributing studies to investigate the associations of 10,199,961 genetic variants (minor allele frequency (MAF) >1%) with risk of HF and its subtypes

4. Throughout, the authors describe their analysis as “multi-ancestry” or “ancestrally diverse.” I think this description is disingenuous. >90% of their HF cases are subjects of European ancestry. As for the HF sub-categories, I did not see explicit numbers, but based on Supplementary Figure 2, I would estimate that these analyses harbor even less diversity. Furthermore, several of the downstream analyses used the only the European-ancestry GWAS. I think this study is better described as a predominantly European-ancestry GWAS. Additionally, since the authors perform analyses with the European-ancestry subset, and since many readers may want to use those data, the authors should make sure to report the number of cases and controls for this subset for each phenotype.

Thank you for this suggestion. We agree that the ancestral diversity of the sample is limited reflecting the persistent lack of non-European ancestry studies globally; we have updated the text to highlight this limitation (**line 573-575**). After cross-ancestry GWAS locus discovery, we used the European-only GWAS summary statistics for fine mapping and gene prioritisation because of the requirement to match the out-of-sample linkage disequilibrium matrix reference as closely as possible for applications in many downstream analysis methods including LDSC, LDAK, and TWAS. We have now provided a summary of the number of cases and controls per phenotype per ancestry in **Supplementary Table 2**.

5. I agree with reporting all associations that meet genome-wide significance (GWS). However, it is worth noting that these GWAS reflect repeat analysis on subsets of the same cases/controls. Thus, there is additional multiple hypothesis testing going on. In this setting, the authors should note if some GWS loci are no longer significant after correcting for the fact that they conducted GWAS on different cuts of the same cases/controls four times. This caveat is important, as they highlight the IGFBP7 association with HFpEF as a particularly novel finding, and if it is not significant after accounting for multiple testing, this should be noted for transparency.

We thank the reviewer for this comment. We implemented the conventional GWS threshold for locus discovery, consistent with previous GWAS of multiple correlated traits from the same sample (e.g. lipid⁸, cardiac MRI⁹, ECG traits¹⁰). In response to the reviewer, we have added a sentence to the results to report the proportion of identified loci that would meet this threshold: *In total, 66 independent genomic loci were identified across HF phenotypes under analysis at $P < 5 \times 10^{-8}$, including 46 (70%) loci which passed a multiplicity-adjusted threshold of $P < 5 \times 10^{-8} / 4$ phenotypes* (**line 315-319**).

6. The authors find evidence of genomic inflation in their GWAS. They then argue that this inflation is due to polygenicity rather than population structure based on the LDSC intercept. However, they conducted their LDSC analysis only using the European-ancestry GWAS, which reduces confounding from population structure compared to the full GWAS. This approach seems a bit circular.

Thanks for this comment. The motivation to use LDSC with the European-ancestry GWAS summary statistics is to minimise bias due to the mismatch of reference genotype used in the analysis (in this case, a random sample of 10,000 UK Biobank European ancestry). In response to the reviewer's comment, we have recalculated the genomic inflation coefficient using the European-ancestry meta-analysis subset in **Supplementary Table 6** ($\lambda = 1.22$ for HF). With an LDSC intercept of 1.011 in the same European meta-analysis subset, we interpret the observed genomic inflation as primarily driven by polygenicity rather than confounding by population structure.

7. The authors state that ni-HFpEF had stronger genetic correlations with HF and ni-HF than ni-HFrEF had with HF and ni-HF (lines 339-342). This finding is counterintuitive. Further, Figures 1b and 1c suggest the opposite pattern, with ni-HFrEF seeming to have greater potential overlap in signals across the GWS loci. Can the authors confirm the genome-wide genetic correlation findings, and if so, what might be some plausible explanations for this discrepancy?

We thank the reviewer for their comment. It is the case that the point estimate for the genome-wide genetic correlation between HF and ni-HFpEF was higher than between HF and ni-HFrEF, however the confidence intervals are overlapping (**Supplementary Figure 10**). On this basis, we have removed this interpretation from the results section.

8. The authors conduct several 2-sample MR analyses. Many of the GWAS used as outcomes were meta-analyses that included data from various cohorts and biobanks. Thus, there is the potential for significant sample overlap between their exposure and outcome GWAS. For example, the CAD GWAS used (PMID: 36474045) includes subjects from deCode, UKB, MGB, etc. For each MR, the authors should use outcome GWAS from non-overlapping samples.

We agree that substantial overlap of the exposure and outcome sample datasets in two-sample Mendelian randomisation can be a source of bias and inflated Type 1 error, but the degree of bias would be minimal if the exposure trait has strong genetic instruments as shown by Burgess et al.¹¹

In our study, all the exposure traits analysed were derived from very large GWAS and had good instrument strength, except HCM and DCM where there was no exposure-outcome sample overlap. For continuous exposure variables, sample overlap was minimal because the GWAS studies excluded participants with major cardiovascular diseases. The GWAS of cardiac MRI-derived traits, for example, excluded participants with prevalent HF.

Among the 24 exposure traits investigated, we note potential sample overlap only for AF and CAD on the overall HF phenotype. The current results, however, are similar to findings from our previous study⁵ and from MVP study with minimal sample overlap¹², suggesting that the estimates are robust.

MINOR:

1. There is a typo in the legend for Supp Fig 1. “ni-HFrEF: non-ischaemic heart failure with preserved ejection fraction”

This is now corrected.

2. Line 523: Should the colocalization analyses reference Figure 5 instead of 6?

This is now corrected.

3. What LD reference panel was used for LDpred2-auto?

The LD reference panel used for PRS analysis with LDpred2-auto was built using 362,320 European individuals in the UK Biobank provided by the LDpred2 software author. (now added on **line 747-748**)

Reviewer #3:

Remarks to the Author:

A: Summary of key results:

A GWAS of 1.9 million individuals including 153,174 heart failure patients is presented. It reveals 66 genetic loci, of which 37 are newly reported. There is a great deal of downstream

analysis which tends to support the notion that loci of relevance to heart failure have indeed been identified. However, this reviewer found those analyses rather disappointing regarding new insights into HF pathophysiology. It was not straightforward to extract the main messages from the very large amount of data presented.

There are several statements in the causal analyses that would be considered controversial with regard to causal analyses - for example, an opposite-direction association between CKD and HFpEF; no causal association between T2D and HF – and these should be more critically examined in the discussion rather than speculatively treated at present. It is stated (line 562) that there was evidence of the contribution of epicardial coronary artery disease to HF susceptibility, but only in patients who had had revascularisation or had an MI. Where was the evidence for this presented in the results?

We thank the reviewer for these comments. In response to these comments and those from the other reviewers, we have extensively revised the manuscript for clarity and added results from additional analyses. We have moved some content from the main manuscript to the supplementary material to improve the readability. We have reworked the text to clearly highlight new insights into HF pathophysiology, including critical examination and discussion of findings.

The reporting of the results section has been reworked to highlight only those associations by Mendelian randomisation that were robust to multiple testing adjustments and sensitivity analyses. Accordingly, we report that we were unable to identify evidence for a causal relationship between CKD and eGFR, consistent with prior Mendelian randomisation analyses that have reported null associations with cardiovascular disease^{13,14}.

Regarding our estimates of the effects of liability to T2D on HF, although the IVW estimate was consistent with an association [$OR_{MR-IVW} = 1.04$, 1.03-1.05 per doubling prevalence], this was not robust to our sensitivity analyses. The MR Egger intercept was significantly non-zero ($P < 0.0002$) suggesting potential confounding by pleiotropy. The effects were attenuated using the pleiotropy-robust weighted median estimator and did not meet our multiplicity-adjusted significance threshold [$OR_{MR-Egger} = 0.98$, 0.96-1.00, $OR_{MR-WME} = 1.01$, 1.00 - 1.02 per doubling prevalence]. We have reframed our interpretation as: “These findings were suggestive that liability to T2D may have little or no effect on HF” (**line 485-486**).

We have updated the reporting of the estimated effects of liability to CAD on HF and ni-HF to clarify that the ni-HF patients had heart failure in the absence of revascularization, myocardial infarction, or other major structural heart diseases (**line 565-568**).

Asserting in the Discussion that HFpEF is chiefly ascribed to diastolic dysfunction, for which the study finds no evidence and therefore challenges received wisdom, is putting up a straw man. In the opinion of this reviewer, HFpEF is already widely recognised as a highly complex and heterogeneous phenotype, not simply the result of primary diastolic dysfunction. It is, however, surprising that no association between diastolic dysfunction and HFpEF was found, and clinician readers will likely find it so. Accordingly, please set out the strength of the evidence underpinning this conclusion (how many patients is it based on, what phenotypic measurements) and more critically evaluate the conclusion. Similarly, this reviewer believes it is already accepted by HFpEF researchers that the kidney, metabolic and vascular tissues play important roles in the aetiology of the condition. While this genetic evidence is (line 580) in agreement, it is not the first recognition of this.

Thanks for highlighting this. We agree the findings of no association between the imaging biomarkers of diastolic dysfunction and niHFpEF represents a notable finding. Cardiac MRI-derived biomarkers for diastolic dysfunction are not well established clinically and diagnosis relies on combinations of multiple measures from echo Doppler imaging. The diastolic function biomarkers that were available from published GWAS were for radial and longitudinal peak diastolic strain rates (PDSR_r and PDSR_l) derived from 40,000 UK Biobank participants¹⁵. There is uncertainty on how well these imaging biomarkers capture the physiology of diastolic dysfunction, but no other measures were available. We have reworked the discussion to more critically evaluate the evidence (**line 519-523**). We have also reframed the discussion to report that our study provides genetic evidence supporting the widely-held hypothesis of HFpEF as a multisystem disorder (**line 557-560**).

The clarity of the results could be improved if the associated variants were presented classified dependent on whether they had been previously found in GWAS studies of coronary disease and the major clinical risk factors for HF. This is not in any single place in the manuscript, rather it is piecemeal through the text. It is not possible to tell how many variants are likely working through increasing risk of coronary disease, risk of hypertension, risk of obesity and so forth (though these loci are present in the gene list e.g. CDKN2A/B,

FTO, etc.) This reviewer believes it to be important to report the variants in this way as it will help reveal the extent to which this work has indeed identified new biology and increased our understanding of the HF phenotype's dependence on known risk factor pathways.

Thanks for this suggestion. We have provided the associations of HF with other known risk factors in **Supplementary Figure 19** and **Supplementary Table 15**. We also report in the main text that *24 of the 66 HF loci (36%) were not previously reported in any of the GWAS of tested traits at $P < 5 \times 10^{-8}$* (line **445-446**).

“Cluster 4” in the graph modelling analyses (line 451 ff) centres on ALDH2. Alcohol is an important risk factor for HF – what can be said about the interplay of ALDH2 genotype, alcohol and HF risk in the dataset?

Thank you for this suggestion. We have expanded the discussion on the potential interplay between *ALDH2* genotype, alcohol, and HF risk as follows (line **433-438**):

Cluster 3 centred around the pleiotropic effects of a locus at which ALDH2 was prioritised. A lack of ALDH2 (aldehyde dehydrogenase 2), a key enzyme for removing toxic acetaldehydes, has been associated with increased risk of cancer and cardiovascular diseases; and the lead variant at the locus has been associated with a reduced ALDH2 expression in blood

The authors calculate a PRS with a view to evaluating its clinical utility. But they do not present the PRS discriminatory value in a way that would be useful for screening, or comment further in the discussion on this. Please present the PRS in a way that enables its potential utility to be estimated – e.g. stating what proportion of cases would be picked up with a 5% false positive rate. Please make comments on this in the discussion.

Thank you for this suggestion. We estimated that 19% of the cases were identified at a 5% false positive rate using the model with PRS, Sex, Age, and first ten 10 genetic principal components (now reported on line **341-344**)

This reviewer believes that an important limitation of this study, that should be discussed, is the small size of the non-ischemic HFpEF cohort even in such a large international collaborative study. Only a small number of GWAS significant loci were detected. The study seems to reveal a clear need for larger genetic studies specifically focused on HFpEF and could call for this.

We agree. We have added this as a limitation and highlighted the need for larger prospective studies with more accurate phenotype measurement to better characterise the genetic basis of HF subtypes (**line 578-582**). Our group is also actively engaged in efforts to address this critical need for larger HFpEF samples. These include two major prospective studies: the NIHR UK HFpEF study and the NIH HeartShare programme.

B: Originality and significance

This is the largest GWAS study of heart failure yet performed, involving a very large amount of data contributed by many groups around the world, and it has identified a number of new loci.

C: Data and methodology

Patient selection is complex and difficult to follow. 153174 HF patients were in the overall cohort. ~44K were identified as “non-ischemic”. Does that mean the remainder were all “ischemic” or could some in that group not be classified? Among the 44K, much smaller numbers are classified as “non-ischemic HFrEF” (5,406) and “non-ischemic HFpEF” (3,841), but in the Figure 1 schema it seems that all of the 44K should be classified into HFpEF or HFrEF dependent on $EF < 50\%$. The current definition of HFrEF requires $EF < 40\%$, with HFpEF classified as $EF > 50\%$, why was that not followed here?

Alternatively, and since the Figure 1a classification scheme seems tautologous, is there a misprint (should “ $< 50\%$ ” be “ $< 40\%$ ”)? Does the classification imply that the remaining ~34K patients could not be classified as either HFpEF or HFrEF? These questions need to be answered in the flow of the text, or the schema of Figure 1 needs to be expanded to answer them.

Thanks for this comment. There were indeed non-ischaemic HF cases that could not be classified based on ejection fraction due to a lack of LVEF measurements. For HFpEF, we further required LVEF measures at or after HF diagnosis to avoid potential misclassification of patients with recovered EF. We note that our definition of HFrEF includes both HF with mildly reduced (LVEF 41-49%) and reduced ejection fraction (LVEF $< 40\%$), and have added this to the discussion (**line 540-543**).

How does the inability to classify the majority of the non-ischemic patients as HFpEF or HFrEF affect what readers should take from the study? Is it a limitation? In the Discussion, it would be helpful to have some comments on these unclassified patients.

The inability to classify patients relates to the requirement of data from echocardiography in patients at appropriate time points. This is a limitation inherent in the data that are typically available from large biobanks linked to coded health record data. Prospective studies will be required to address this at scale as highlighted above. This is now included in the limitations section (**line 578-582**).

The presentation of the results is not optimal to clearly identify any new pathophysiological insights. The previous identification of HF loci identified here, in GWAS studies of HF risk factors, including coronary artery disease, hypertension, obesity, diabetes, etc. is not straightforward to work out. The discussion contains no attempt to place the present work in context of previous studies of HF genetics (there is an assessment of agreement in the results). Basic information such as how many cases this study adds to previous work is needed by readers to interpret the significance of the study. Have any of the patients in this study been participants in previous HF GWAS studies?

Thank you for this comment. We have provided a comparison of genetic association reported in GWAS of 24 related traits in **Supplementary Figure 19** and **Supplementary Table 15**. Of the identified 66 HF loci, 24 (36%) were not previously reported in any of the GWAS of related traits at $P < 5 \times 10^{-8}$ (**line 445-446**). We now also provide a comparison with previously published GWAS of HF in **Supplementary Table 5**. We found that 37 of the 66 loci (56%) identified in the present study were not reported in previous GWAS of HF, and replicated 76/87 (87%) previously reported HF sentinel variants that were available in our study at $P < 0.05/87$ (**line 315-318**)

This reviewer had difficulty following the rationale for classifying loci into “non ischemic” and “secondary” (line 320) – these are not complementary terms. Does it mean secondary to ischemia?

Thank you for this comment. We have relabelled secondary as Other HF loci to indicate that they are loci that are associated with all HF but did not pass significance when major secondary causes were excluded (myocardial infarction, revascularisation, major valvular and structural heart disease).

D: Statistics

Not the expertise of this reviewer.

E: Conclusions

Of the four key conclusions, the first, about the relationship with risk factors, is not clearly presented and some of the statements made would merit more critical evaluation. The second – the relationship between contractility traits and HF – is already known in single gene disease and the appropriate work should be discussed. The conclusions about HFpEF need to be more contextualised with the relatively small numbers of patients studied, a clear need for larger GWAS studies of HFpEF, and previous clinical studies that have also pointed the same way as these genetic findings. The fourth conclusion regarding novel molecular mechanisms is perhaps the most interesting, but in the presentation of the results this rather got lost in the profusion of downstream analyses.

Thank you for this comment. We have updated the discussion to better highlight new insights with potential translational implications from the present study (line 546-572).

F: Suggested improvements

See above comments.

G: References

See above comments.

H: Clarity and context

Suggestions for improvement of these aspects made above.

REFERENCES

1. Zhang, L. *et al.* Insulin-like growth factor-binding protein-7 (IGFBP7) links senescence to heart failure. *Nature Cardiovascular Research* **1**, 1195–1214 (2022).
2. Nasser, J. *et al.* Genome-wide enhancer maps link risk variants to disease genes. *Nature* **593**, 238–243 (2021).
3. Ghousaini, M. *et al.* Open Targets Genetics: systematic identification of trait-associated genes using large-scale genetics and functional genomics. *Nucleic Acids Res.* **49**, D1311–D1320 (2021).
4. Chaffin, M. *et al.* Single-nucleus profiling of human dilated and hypertrophic cardiomyopathy. *Nature* **608**, 174–180 (2022).
5. Shah, S. *et al.* Genome-wide association and Mendelian randomisation analysis provide insights into the pathogenesis of heart failure. *Nat. Commun.* **11**, 163 (2020).
6. Burgess, S., Small, D. S. & Thompson, S. G. A review of instrumental variable estimators for Mendelian randomization. *Stat. Methods Med. Res.* **26**, 2333–2355 (2017).
7. Mägi, R. *et al.* Trans-ethnic meta-regression of genome-wide association studies accounting for ancestry increases power for discovery and improves fine-mapping resolution. *Hum. Mol. Genet.* **26**, 3639–3650 (2017).
8. Graham, S. E. *et al.* The power of genetic diversity in genome-wide association studies of lipids. *Nature* **600**, 675–679 (2021).
9. Pirruccello, J. P. *et al.* Genetic analysis of right heart structure and function in 40,000 people. *Nat. Genet.* **54**, 792–803 (2022).
10. van Setten, J. *et al.* Genome-wide association meta-analysis of 30,000 samples identifies seven novel loci for quantitative ECG traits. *Eur. J. Hum. Genet.* **27**, 952–962 (2019).
11. Burgess, S., Davies, N. M. & Thompson, S. G. Bias due to participant overlap in two-sample Mendelian randomization. *Genet. Epidemiol.* **40**, 597–608 (2016).
12. Joseph, J. *et al.* Genetic architecture of heart failure with preserved versus reduced ejection fraction. *Nat. Commun.* **13**, 7753 (2022).
13. Charoen, P. *et al.* Mendelian Randomisation study of the influence of eGFR on coronary heart disease. *Sci. Rep.* **6**, 28514 (2016).
14. Morris, A. P. *et al.* Trans-ethnic kidney function association study reveals putative causal genes and effects on kidney-specific disease aetiologies. *Nat. Commun.* **10**, 29 (2019).

15. Thanaj, M. *et al.* Genetic and environmental determinants of diastolic heart function. *Nat Cardiovasc Res* **1**, 361–371 (2022).

Response to reviewers' comments

Authors' responses are displayed in blue

Dear *Nature Genetics* Editorial Team,

We thank the editorial team and the reviewers for providing feedback on our revised manuscript. We have now addressed the main concerns highlighted in the editorial letter, as indicated by tracked changes in the submitted manuscript file. We also provide detailed responses to the remaining comments from the reviewers below.

In response to the points highlighted by the editorial team: firstly, we have performed a meta-regression analysis to estimate potential effect modification by the proportion of males / females across participating studies, which ranges from 0% to 100% males. In summary, we found 3 sentinel variants within the 66 identified genomic loci across HF phenotypes with nominal evidence of a sex-modified genetic effect ($P < 0.05$). These results are now included in the main text (**line 322-326**), with methods described in details (**line 712-727**) and full results provided in **Supplementary Table 6**. Performing an individual-level sex-stratified analysis across participating cohorts at this stage would be technically impractical and challenging for us due to unavailability of sex-specific results in the current data freeze. The meta-regression analysis result informs a better-powered, *de novo* sex-stratified analysis that we will embark on after publication. Secondly, we have added the limitations of the two-sample Mendelian Randomization as well as the new meta-regression analysis the Discussion (**line 584-592**).

We thank the editors for considering our work for publication in *Nature Genetics*. We also welcome further feedback and discussion if there is any remaining concern regarding our manuscript.

Yours sincerely

Tom

Dr Tom Lumbers MA PhD MRCP FESC

Referee expertise:

Referee #1: Genetics, cardiovascular diseases, functional genomics

Referee #3: Genetics, cardiovascular diseases, clinical translation

Reviewers' Comments:

Reviewer #1:

Remarks to the Author:

The authors attempt to address several of these comments, however there are too many unresolved concerns.

We appreciate the clarity on the phenotype definitions, however a sex-stratified analysis or sex interaction analysis for the top loci would seem appropriate here.

Also, it should be possible to add some functional evidence to support top loci.

Also, the authors should perform multivariable MR analyses instead of two sample MR, especially given the sample overlap between the exposure and outcome GWAS used.

Finally, the authors will need to modify the title as it should include "in European-ancestry populations". Also "human genetics" is too vague and does not provide details on the methodology used.

We thank the reviewer for this feedback. In response, we have performed a meta-regression analysis to estimate potential effect modification by sex across the identified genetic loci and described the method in the main text (**line 712-727**). In summary, we found nominal evidence of potential sex-modified genetic effect in 3 out of 66 sentinel variants across identified HF loci (**line 322-326, Supplementary Table 6**). We are unable to perform a formal sex-stratified genetic association analysis across participating cohorts at this stage due to unavailability of sex-specific results in the current data freeze. The findings of the study-level meta-regression inform a *de novo* sex-stratified analysis, with updated case counts, which we will embark on

after publication of the present manuscript. We have also discussed potential limitations of this analysis due to lack of statistical power and potential confounding by sex-related factors in the main text (**line 584-588**).

Whilst we agree that findings from the present study warrant functional validation, providing such evidence is a substantial undertaking and beyond the scope of the current manuscript. Indeed, an important aspect of the current work is for the findings to be released as soon as possible to allow multiple laboratories, not just ours, to embark on functional studies in parallel to the benefit of the wider community.

Regarding the MR analysis, we have added a discussion on caveats and potential limitations of two-sample MR due to sample overlap. We also note that the scope of univariable MR analysis in the current study is limited to estimating causal effects of individual risk factors, and we highlight multivariable MR and mediation analysis as a possible follow-up analysis to model the relationships between individual risk factors (**line 588-592**). Finally, we keep the current manuscript title as we feel it succinctly describes the scope of the study, and that details of study participants and methods used are already outlined in the abstract.

We are happy to liaise further with the reviewer and editorial team if there is any remaining concern regarding our manuscript.

Reviewer #3:

Remarks to the Author:

Thanks to the authors for carefully considering and responding to the points made at first review. No further comments.